# Asymptotics of the Bootstrap via Stability with Applications to Inference with Model Selection

**Morgane Austern**
Microsoft Research
morgane.austern@gmail.com

**Vasilis Syrgkanis**
Microsoft Research
vasy@microsoft.com

## Abstract

One of the most commonly used methods for forming confidence intervals is the empirical bootstrap, which is especially expedient when the limiting distribution of the estimator is unknown. However, despite its ubiquitous role in machine learning, its theoretical properties are still not well understood. Recent developments in probability have provided new tools to study the bootstrap method. However, they have been applied only to specific applications and contexts, and it is unclear whether these techniques are applicable to the understanding of the consistency of the bootstrap in machine learning pipelines. In this paper, we derive general stability conditions under which the empirical bootstrap estimator is consistent and quantify the speed of convergence. Moreover, we propose alternative ways to use the bootstrap method to build confidence intervals with coverage guarantees. Finally, we illustrate the generality and tightness of our results by examples of interest for machine learning including for two-sample kernel tests after kernel selection and the empirical risk of stacked estimators.

## 1 Introduction

Bootstrap resampling [26], has been one of the most popular techniques for measuring the uncertainty of a statistic, primarily due to its simple algorithmic definition and its conveniency with dealing with opaque statistical procedures that produce a test statistic. For this reason, uncertainty quantification based on bootstrap resampling has been a staple in the machine learning community, starting from the early days of the field [40] and continuing into the deep learning and SVM era [39, 30, 32, 17]. Despite it's widespread use, general conditions for the consistency of the bootstrap for complex non-linear statistics is generally not fully explored and hence, it is not clear when the bootstrap method will accurately capture uncertainty in machine learning pipelines, especially when model selection procedures are involved.

Our goal is to understand the distribution of a large class of non-linear statistic $\hat{\theta}_n := g_n(X_1, \ldots, X_n)$, as the samples $X_i$ are drawn from their unknown distribution. Examples of such statistics could be the out-of-sample risk of a machine learning predictor, or the maximum-mean-discrepancy of a two-sample kernel test, or the prediction of a machine learning model at a given point. One approach to approximating this distribution is the empirical bootstrap: sample new observations $Z_1, \ldots, Z_n$ independently and uniformly from $\{X_1, \ldots, X_n\}$ (with replacement), and define $\hat{\theta}_n^{\text{boot}}$ as the value of the estimator taken at the bootstrap sample $\hat{\theta}_n^{\text{boot}} := g_n(Z_1, \ldots, Z_n)$. This procedure can be repeated many times in order to estimate the conditional distribution of $\hat{\theta}_n^{\text{boot}}$. If this distribution is approximately the same as the distribution of $\hat{\theta}_n$, as the sample size $n$ grows, we say that the bootstrap method is consistent. Consistency of the bootstrap can be subsequently utilized to construct confidence intervals for the quantity of interest.

35th Conference on Neural Information Processing Systems (NeurIPS 2021).

Classical consistency proofs for the bootstrap, either require that the limiting distribution of the statistic be Gaussian, or even more stringently that the statistic is asymptotically linear. Notably, when $\hat{\theta}_n$ is asymptotically normal those intervals are known to be consistent under general conditions; see e.g. [33, 5, 18]. Moreover, classical conditions for such statements to hold are that the statistic is Hadamard differentiable and smooth with respect to the underlying distribution of the random variables. However, these properties can be violated by test statistics that are implicitly defined via machine learning data analysis pipelines and especially when the model selection is involved. Thus it is crucial to provide more generally applicable sufficient conditions for the validity of the bootstrap.

Notably, recent breakthrough results in statstics by Chatterjee [13], have shown that the Lindenberg technique that has been long used to establish Central Limit Theorems, is more widely applicable and can be leveraged to show limit distributional consistency of two random variables more broadly. Notably, this intuition has already been exploited to show that the bootstrap method is consistent in particular applications in the econometric literature, such as the construction of uniform confidence bands. However, these proofs are tailored to the particular application of interest and do not provide general characterizations.

Our main contribution is to provide a set of sufficient stability conditions that imply the consistency of the bootstrap and which go well beyond the existing general conditions via the use of Chatterjee's generalized Lindenberg approach and a smart path interpolation technique. We then apply these general results to derive inference in two machine learning applications: i) bootstrapping the test statistic in two-sample kernel tests, after model selection on the kernel and sample re-use, ii) bootstrapping the risk of a stacked estimator with sample re-use.

Roughly, our sufficient conditions impose unilateral stability properties on the statistic, reminiscent of the stability conditions required for classical concentration inequalities, such as McDiarmid's inequality. In particular, we assume that the functions $(g_n)$ are approximable by three-times differentiable functions whose first, second and third order partial derivatives, with respect to a single observation, taken at $(X_1, \ldots, X_n)$, are of respective order $o(n^{-1/3})$, $o(n^{-1/2})$ and $o(n^{-1})$. These conditions assure that the value of $g_n(X_1, \ldots, X_n)$ is not oversensitive to the value of a single observation (see Section 3 for a formal exposition). Exploiting these assumptions, we exactly characterize the limiting distribution of the bootstrap estimator $\hat{\theta}_n^{\text{boot}}$, compare it to the distribution of the original statistic $\hat{\theta}_n$ and study how fast the distribution of $\hat{\theta}_n^{\text{boot}}$ converges. Notably, we also discover that when the mean of the observations $X_1$ is unknown, then the bootstrap method is in general not consistent and we propose corrections to bootstrap based intervals with guaranteed minimum coverage.

**Related litterature**   The empirical bootstrap method was first introduced in a breakthrough paper by Efron [27]. Other bootstraps methods have since been proposed including the multiplier bootstrap [54], the residual bootstrap [21] or the non-remplacement bootstrap method [47]. A vast literature studies the theoretical properties of those techniques with some of the main results synthesized in the following books [33, 21, 36, 3]. Most relevant to us are studies of the asymptotics of the bootstrap method. The consistency of the bootstrap method for linear statistics, t-statistics, Von-Mises functionals and quantiles has been established in [5, 46, 48] and for U-statistics in [1, 55]. Those results, among others, have been extended to high-dimensional regression and M-estimation [6, 45, 10, 2, 24], misspecified models [49], solutions of estimating equations [11] and to robust estimators [15]. In contrast, other works established the poor performance of the bootstrap method for non smooth statistics [25, 4, 5], or for non-sparse high-dimensional regressions [28].

Several recent breakthrough papers studied the consistency of the bootstrap method, both empirical and wild, for the maximum of high-dimensional centered averages with the dimension taken to be growing exponentially fast with the sample size. Notably [18, 19] established the consistency of the bootstrap and Gaussian approximation method when respectively $\log(p_n n)^{7/8} = o(n^{1/8})$ and $\log(p_n n)^{7/6} = o(n^{1/6})$ hold. A series of works have strengthen those results: [20, 23, 41] established the consistency of the multiplier and empirical bootstrap when $\log(pn)^{5/4} = o(n^{1/4})$, [43] established a quasi $\sqrt{n}^{-1}$ rate for the wild bootstrap, [22] built slightly conservative confidence sets with guaranteed coverage under the conditions that $\log(p) = o(n)$ and [16] proved that similar results hold for high-dimensional U-statistics. Those works use a combination of the Stein method, Edgeworth expansions, Lindeberg's method [13] and the Slepian smart interpolation path. We note that the limiting distributions of those statistics are in general not Gaussian [23]. In contrast, our results apply more broadly and are not limited to the study of maximums of centered empirical

averages. We notably apply our results to machine learning estimators that are smoothed arg-minima of an objective function. Other works have studied the accuracy of the bootstrap method for specific statistics whose distributions are known to be asymptotically not Gaussian such as: the operator norm in high dimensions [44, 34, 37], sampled eigenvalues of random matrices in high and moderate dimensions [29] or M-estimators having cube root convergence [9]. The main contrast between this series of work and ours is that, rather than studying the bootstrap method for one specific statistic or application, we seek to establish the asymptotics of the bootstrap method under universal conditions on the estimators $(g_n)$. Our proof builds on a breakthrough method proposed by Chatterjee [13] that generalized the Lindeberg method to a general technique for comparing the expectations of $f(X_{1:n})$ and $f(Y_{1:n})$ of a large class of functions $f$.

## 2 Problem Statement

Let $(X_i^n)$ be a triangular array of independent and identically distributed (i.i.d) processes with observations $X_i^n$ taking value in $\mathbb{R}^{d_n}$. Moreover, let $X^n = (X_1^n, \ldots, X_n^n)$ denote its $n$-th row. Consider an estimator $\hat{\theta}_n := g_n(X^n)$, where $g_n : \times_{l=1}^n \mathbb{R}^{d_n} \to \mathbb{R}$ is a measurable function, that we will typically refer to as a *statistic*, and let $(g_n)$ denote the sequence of measurable functions as $n$ grows. To evaluate the performance of this estimator and build confidence intervals, we need to approximate its distribution. In this work, we will analyze the empirical bootstrap method.

**Empirical bootstrap**  Bootstrap samples $Z^n = (Z_1^n, \ldots, Z_n^n)$ are sampled with replacement from the observations $\{X_1^n, \ldots, X_n^n\}$. This implies that conditionally on $X^n$ the coordinates of $Z^n$ are distributed i.i.d, with $Z_i^n \mid X^n \sim \text{unif}(\{X_1^n, \ldots, X_n^n\})$, for all $i \in [n]$.

**Consistency metric and bootstrap consistency**  Throughout the paper we denote with $Y^n = (Y_1^n, \ldots, Y_n^n)$ an independent copy of $X^n$. The bootstrap method is said to be consistent for $(g_n)$ if conditionally on $X^n$ the distribution of $g_n(Z^n)$ well-approximates the distribution of $g_n(Y^n)$, as $n \to \infty$. To make this statement rigorous we introduce a metric on the space of probability distributions. First, we define the class of three times continuously differentiable measurable functions with bounded third-order derivatives:

$$\mathcal{F} := \left\{ h \in C^3(\mathbb{R}) \mid \sup_{x \in \mathbb{R}} \left| h^{(i)}(x) \right| \leq 1, \ \forall 1 \leq i \leq 3 \right\};$$

Given this, we define the distance on the space of probability measures, as the maximum mean discrepancy, where test functions range over the class $\mathcal{F}$:

$$d_{\mathcal{F}}(\mu, \nu) := \sup_{h \in \mathcal{F}} \mathbb{E}_{X \sim \mu, Y \sim \nu}[h(X) - h(Y)].$$

We remark that $d_{\mathcal{F}}$ is a metric on the space of probability measures of real-valued random variables. Notably for two probability distributions $\mu$ and $\nu$ if $d_{\mathcal{F}}(\mu, \nu) = 0$ then those distributions are the same $\nu = \mu$. Moreover, the topology defined by $d_{\mathcal{F}}$ is finer than the weak convergence topology. Indeed, for a sequence of distributions $(\nu_n)$ if we have $d_{\mathcal{F}}(\nu_n, \mu) \to 0$ then $(\mu_n)$ converges weakly to $\mu$: $\nu_n \xrightarrow{d} \mu$. Finally we remark that this metric is related to the classical Levy-Prokhorov distance on probability spaces [7].

Moreover, we use the shorthand notation:

$$d_{\mathcal{F}}(\mu, \nu \mid \mathcal{E}) := \sup_{h \in \mathcal{F}} \mathbb{E}_{X \sim \mu, Y \sim \nu}[h(X) - h(Y) \mid \mathcal{E}].$$

We say that *the empirical bootstrap method is consistent for* $(g_n)$ if:

$$d_{\mathcal{F}}(g_n(Z^n), g_n(Y^n) \mid X^n) \xrightarrow{p} 0.$$

**Centering discrepancy and centered bootstrap consistency**  Notably, an individual bootstrap sample $Z_1^n \mid X^n$, has a slightly different mean $\mathbb{E}[Z_1^n \mid X^n] = \bar{X}^n := \frac{1}{n} \sum_{i \leq n} X_i^n$, than the one of $X_1^n$. As we will see this small difference plays a crucial role in determining the consistency of the bootstrap and for this reason it will be useful to define artificially centered versions of the random variables $(Z_i^n)$ and $(Y_i^n)$. A centered bootstrap sample

$$\tilde{Z}_i^n := Z_i^n - (\bar{X}^n - \mathbb{E}[X_1^n])$$

is a bootstrap sample that has been re-centered to artificially have the same mean as $X_1^n$. Moreover, denote with $\tilde{Y}_i^n$ a corrected version of $Y_i^n$, artificially re-centered to have the same mean as $Z_1^n$, i.e.:

$$\tilde{Y}_i^n := Y_i^n + \bar{X}^n - \mathbb{E}\left[X_1^n\right].$$

We say that *the centered bootstrap is consistent for* $(g_n)$ if:

$$d_{\mathcal{F}}\left(g_n(\tilde{Z}^n), g_n(Y^n) \mid X^n\right) \xrightarrow{p} 0.$$

**From metric consistency to confidence intervals with nominal coverage**  We can compare the confidence intervals of two random variables $X$ and $Y$ in terms of their mutual distance $d_{\mathcal{F}}(X, Y)$ (proof in Appendix M.1).

**Proposition 1** *Let $X$ and $Y$ be two real-valued random variables and $\mathcal{E}$ any random event. Let $\epsilon > 0$ be a constant then for any Borel set $A \in \mathcal{B}(\mathbb{R})$ the following holds:*

$$P(X \in A_{6\epsilon} \mid \mathcal{E}) \geq P(Y \in A \mid \mathcal{E}) - \frac{d_{\mathcal{F}}(X, Y \mid \mathcal{E})}{\epsilon^3},$$

*where we wrote $A_\epsilon := \{x \in \mathbb{R} \mid \exists y \in A \text{ s.t } |x - y| \leq \epsilon\}$. Moreover, if $[a, b]$ is a confidence interval at level $1 - \alpha$ for $Y - \mathbb{E}\left[Y \mid \mathcal{E}\right]$, conditional on $\mathcal{E}$, then we have:*

$$P\left(X - \mathbb{E}\left[X \mid \mathcal{E}\right] \in [a - 6\epsilon, b + 6\epsilon] \mid \mathcal{E}\right) \geq 1 - \alpha - \frac{2d_{\mathcal{F}}(X, Y \mid \mathcal{E})}{\epsilon^3}.$$

For instance, suppose that we care about estimating $\theta_n := \mathbb{E}\left[g_n(Y^n)\right]$. Then the bootstrap method, if consistent, can be used to build consistent confidence intervals for $\theta_n$. Indeed since we can estimate the conditional distribution of $\theta_n^{\text{bootstrap}} := g_n(\tilde{Z}^n)$, by drawing sufficiently many bootstrap sub-samples, we can find $C^{\alpha,n}$ such that

$$P\left(\hat{\theta}_n^{\text{bootstrap}} - \mathbb{E}\left[\hat{\theta}_n^{\text{bootstrap}} \mid X^n\right] \in C^{\alpha,n} \mid X^n\right) = 1 - \alpha.$$

Then, if we write $\hat{\theta}_n := g_n(Y^n)$, using the consistency of the bootstrap method we obtain that:

$$\liminf_{\epsilon \downarrow 0} \liminf_{n \to \infty} P\left(\hat{\theta}_n - \theta_n \in C_\epsilon^{\alpha,n}\right) \geq 1 - \alpha.$$

Therefore, confidence intervals built using the bootstrap method achieve asymptotically nominal level of confidence. We note that prior works (e.g. [18, 19]), typically provide a slightly stronger statement that $\liminf_{n \to \infty} P(\hat{\theta}_n - \theta_n \in C^{\alpha,n}) \geq 1 - \alpha$, by proving anti-concentration results on the limit distribution of $\hat{\theta}_n - \theta_n$. Such anti-concentration, allows one to argue that the mass of the random variable $\hat{\theta}_n - \theta_n$ contained in $C_\epsilon^{\alpha,n}$ converges to the mass contained in $C^{\alpha,n}$ as $\epsilon \downarrow 0$ and thereby, $\liminf_{\epsilon \downarrow 0} \liminf_{n \to \infty} P(\hat{\theta}_n - \theta_n \in C_\epsilon^{\alpha,n}) = \liminf_{n \to \infty} P(\hat{\theta}_n - \theta_n \in C_\epsilon^{\alpha,n})$. Given that these results typically require stronger conditions on the statistic and many times Gaussian limits, we omit this step in this work and note that a slightly weaker, albeit still practically useful, statement on coverage is achievable in a more general setup.

**From metric consistency to p-values**  Alternatively, suppose that we want to test if a specific null hypothesis $(H_0^n)$ holds against the alternative $(H_1^n)$. To do so we compute a test statistic $T_n(X_{1:n})$ and determine a rejection region $\mathcal{R}^n$. A crucial quantity to estimate is the p-value $P\left(T_n(X_{1:n}^n) \in \mathcal{R}^n | H_0\right)$. The bootstrap method, if consistent, allows us to upper-bound the p-value by enlarging the rejection region with an infinitesimally small quantity. Indeed according to Proposition 1 if the bootstrap method is consistent then we have

$$\lim_{\epsilon \downarrow 0} \liminf_{n \to \infty} P\left(T_n(Z_{1:n}^n) \in \mathcal{R}_\epsilon^n | X^n\right) \geq \limsup_{n \to \infty} P\left(T_n(X_{1:n}^n) \in \mathcal{R}^n\right).$$

## 2.1  Notations and definitions

For a scalar random variable $X$ we denote with $\|X\|_{L_p}$, the $L_p$-norm: $\|X\|_{L_p} := \mathbb{E}[X^p]^{1/p}$. Moreover, for vector $x \in \mathbb{R}^d$, we denote with $\|x\|_p$, the $\ell_p$ vector norm: $\|x\|_p = \left(\sum_{i=1}^d x_i^p\right)^{1/p}$. For simplicity, given a sequence $(x_i)$, with $x_i \in \mathbb{R}^d$ and a constant $c \in \mathbb{R}^d$, we shorthand

$$x_{1:n} := (x_1, \ldots, x_n), \quad x_{1:n} + c := (x_1 + c, \ldots, x_n + c), \quad cx_{2:n} := (c, x_2, \ldots, x_n).$$

We denote the $k$-th coordinate of $x_i \in \mathbb{R}^d$ as $x_{i,k}$. For a function $f : \times_{l=1}^n \mathbb{R}^{d_n} \to \mathbb{R}$ and a random variable $X$ taking values in $\mathbb{R}^{d_n}$, we designate $f(\cdot + X)$ the random function: $x_{1:n} \to f(x_{1:n} + X)$.

**Lindenberg path interpolation** Let $Z^{n,i}$ and $Z^{n,i,x}$ be the following interpolating processes between $Z^n$ and $\tilde{Y}^n$:

$$Z^{n,i} := \left(\tilde{Y}_1^n, \ldots, \tilde{Y}_i^n, Z_{i+1}^n, \ldots, Z_n^n\right)$$

$$Z^{n,i,x} := \left(\tilde{Y}_1^n, \ldots, \tilde{Y}_{i-1}^n, x, Z_{i+1}^n, \ldots, Z_n^n\right)$$

**Higher-order derivatives and bounds** If a function $f$ is three-times differentiable then we let:

$$\partial_{i,k} f(x_{1:n}) := \partial_{x_{i,k}} f(x_{1:n})$$

$$\partial_{i,k_{1:2}}^2 f(x_{1:n}) := \partial_{x_{i,k_1}} \partial_{x_{i,k_2}} f(x_{1:n})$$

$$\partial_{i,k_{1:3}}^3 f(x_{1:n}) := \partial_{x_{i,k_1}} \partial_{x_{i,k_2}} \partial_{x_{i,k_3}} f(x_{1:n})$$

Moreover, for a potentially random function $f$ we define the constants:

$$M_k^n := 2 \left\| X_{1,k}^n \right\|_{L_{12}},$$

$$D_{k_1}^n(f) := M_{k_1}^n \max_{i \leq n} \left\| \partial_{i,k_1} f(Z^{n,i,\bar{X}^n}) \right\|_{L_{12}}$$

$$D_{k_{1:2}}^n(f) := M_{k_1}^n M_{k_2}^n \max_{i \leq n} \left\| \partial_{i,k_{1:2}}^2 f(Z^{n,i,\bar{X}^n}) \right\|_{L_{12}}$$

$$D_{k_{1:3}}^n(f) := M_{k_1}^n M_{k_2}^n M_{k_3}^n \max_{i \leq n} \left\| \max_{x \in \left[\bar{X}^n, \tilde{Y}_1^n\right] \cup \left[\bar{X}^n, Z_1^n\right]} \partial_{i,k_{1:3}}^3 f(Z^{n,i,x}) \right\|_{L_{12}}$$

where for any two vectors $a, b \in \mathbb{R}^d$, we denote with $[a, b]$ their convex closure, i.e.

$$[a, b] := \{t\, a + (1 - t)\, b : t \in [0, 1]\}$$

## 3   Main Results

If the statistics $(g_n)$ were linear, i.e. $g_n(x_{1:n}) = \sum_{i \leq n} x_i$, then the influence of a single observation $X_1^n$, on the estimate $\hat{\theta}_n$, would depend uniquely on the value of the random variable itself, i.e. $g_n(X^n) - g(0 X_{2:n}^n) = X_1^n$. This is not the case for non-linear statistics. For instance, if $g_n(x_{1:n}) = \max\left(\sum_{i \leq n} x_{i,1}, \sum_{i \leq n} x_{i,2}\right)$, then the influence of observation $x_1$ depends on the relative size of $\sum_{i>2} x_{i,1}$ and $\sum_{i>2} x_{i,2}$. In this paper, we want to study the asymptotics of the bootstrap method for such non-linear statistics, with complex influence functions. To control the degree of non-linearity, we assume that the statistics $(g_n)$ can be approximated by three times differentiable functions.

**Assumption 1 ($\mathbb{C}^3$-Approximability)** *There exists a sequence of functions $(f_n)$ with $f_n \in \mathbb{C}^3$ s.t.:*

  *1. The functions $(f_n)$ approximate the estimators $(g_n)$:*

$$\left\| f_n(Z^n) - g_n(Z^n) \right\|_{L_1} + \left\| f_n(\tilde{Y}^n) - g_n(\tilde{Y}^n) \right\|_{L_1} \xrightarrow{n \to \infty} 0. \tag{$H_0$}$$

  *2. The first, second and third order derivatives are respectively of size $o(n^{-1/3})$, $o(n^{-1/2})$, $o(n^{-1})$:*

$$R_{n,1} := n^{1/3} \sum_{k_1 \leq d_n} D_{k_1}^n(f_n) = o(1) \qquad R_{n,2} := \sqrt{n} \sum_{k_1, k_2 \leq d_n} D_{k_{1:2}}^n(f_n) = o(1)$$

$$R_{n,3} := n \sum_{k_1, k_2, k_3 \leq d_n} D_{k_{1:3}}^n(f_n) = o(1).$$

$$\tag{$H_1$}$$

To motivate Assumption 1, we present in Appendix C two illustrating examples of simple estimators which fail to satisfy conditions $(H_0)$, $(H_1)$ and for which the bootstrap method is not consistent.

Under Assumption 1 we study the limiting distribution of the bootstrap statistic and establish that it is asymptotically the same as $g_n(\tilde{Y}^n)$ (proof in Appendix L).

**Theorem 1** *Let* $(g_n : \times_{l=1}^{n} \mathbb{R}^{d_n} \to \mathbb{R})$ *be a sequence of measurable functions. Let* $(X_i^n)$ *be a triangular array of i.i.d processes such that* $X_1^n \in L_{12}$. *Under Assumption 1, there is a constant* $K$ *independent of* $n$ *such that:*

$$\left\| d_{\mathcal{F}}\left(g_n(Z^n), g_n(\tilde{Y}^n) \mid X^n\right)\right\|_{L_1} \leq \epsilon_n := \left\{ \begin{array}{l} \left\| g_n(\tilde{Y}^n) - f_n(\tilde{Y}^n)\right\|_{L_1} + \left\| g_n(Z^n) - f_n(Z^n)\right\|_{L_1} \\ + K\left(R_{n,1}^2 \max\left\{\frac{1}{n^{1/6}}, R_{n,1}\right\} + R_{n,3} + R_{n,2}\right) \end{array} \right\} \to 0.$$

**Remark 1** *We remark that the theorem also holds under slightly modified stability conditions. See Theorem 8 in the appendix for more details. Moreover, the hypothesis that* $(X_i^n)$ *is an i.i.d process can also be relaxed to assuming that the process* $(X_i^n)$ *is exchangeable. See Theorem 10 for more details in the appendix. Finally, note that Theorem 1 can also be extended to random estimators* $(g_n)$, *such as ones obtained by stochastic optimization methods (e.g SGD). See Theorem 10 for more details in the appendix.*

**Remark 2** *We note that Assumption 1 controls how stable the function* $g_n$ *is to the change of one random variable for example* $X_1$. *Many concentration inequalities, such as the Mcdiarmid inequality, impose conditions on similar quantities. To be able to derive a central limit theorem one would need to make additional assumptions regulating how much this change* $g_n(X_1, X_2, \ldots, X_n) - g_n(0, X_2, \ldots, X_n)$ *depends on the other random variables* $X_2, \ldots, X_n$ *[14]. The function* $h_n \to \max(0, \frac{1}{\sqrt{n}}\sum_{i \leq n} X_i - \mathbb{E}(X_1))$ *is an example of a statistic satisfying Assumption 1 but is not asymptotically normal.*

**Remark 3** *When the mean of the observations* $\mathbb{E}(X_1^n)$ *is known, we propose in Appendix E an alternative bootstrap method that exploits this information, the centered-bootstrap method, and prove that it is consistent for* $g_n(Y_{1:n}^n)$. *This is useful for example for estimating p-values for hypothesis testing.*

Theorem 1 guarantees that we can use the bootstrap method to estimate the distribution of $g_n(\tilde{Y}^n)$, which implies that it can also be used to build confidence intervals for $\mathbb{E}\left[g_n(\tilde{Y}^n) \mid X^n\right]$.

**Corollary 1** *Let* $(g_n : \times_{l=1}^{n} \mathbb{R}^{d_n} \to \mathbb{R})$ *be a sequence of measurable symmetric functions. Let* $(X_i^n)$ *be a triangular array of i.i.d processes such that* $X_1^n \in L_{12}$. *Assume that* $(g_n)$ *and* $(X_i^n)$ *satisfy all the conditions of Theorem 1. Then there is a constant* $K$ *independent of* $n$ *such that for* $\epsilon_n$ *as defined in Theorem 1:*

$$\left\| d_{\mathcal{F}}\left(g_n(Z^n) - \mathbb{E}\left[g_n(Z^n) \mid X^n\right], g_n(\tilde{Y}^n) - \mathbb{E}\left[g_n(\tilde{Y}^n) \mid X^n\right] \mid X^n\right)\right\|_{L_1} \leq 2\epsilon_n \to 0.$$

The distribution we are interested in is that of $g_n(Y^n)$ rather than $g_n(\tilde{Y}^n)$. Moreover, the shape of the confidence intervals of $g_n(\tilde{Y}^n)$ can be arbitrary compared to the ones of $g_n(Y^n)$, i.e. they are not systematically larger or smaller. This is illustrated in Appendix D by a series of examples. Therefore in Appendix F we propose conditions that guarantee that the two distributions are asymptotically identical and we prove in Appendix G that those conditions are tight.

When those conditions are not met we propose the use of the bootstrap method to build adjusted confidence intervals that are guaranteed to have at least (but not necessarily equal to) some minimum asymptotic coverage. In Appendix H we propose an alternative method to do so by exploiting the bootstrap method for slightly shifted observations. Moreover, in Appendix I we assume that the mean $\mathbb{E}(X_1^n)$ belongs to a certain subset $A_n$ and build robust confidence intervals with a guaranteed coverage level for all potential values of the mean.

According to Corollary 1 the bootstrap method can be used to build consistent confidence intervals for $\mathbb{E}(g_n(\tilde{Y}^n) \mid X^n)$. Therefore if we can bound the distance from $\mathbb{E}(g_n(\tilde{Y}^n) \mid X^n)$ to $\mathbb{E}(g_n(Y^n))$ we can use the bootstrap method to build confidence intervals on the latter. To do so we exploit the fact that under mild conditions $\sqrt{n}\left[\bar{X}^n - \mathbb{E}(X_1^n)\right]$ is approximately normal. We assume that the function $x \to \mathbb{E}\left[g_n(Y^n + x)\right]$ is $\alpha$-Holder and that the moments of $X_1^n$ are bounded. More formally,

suppose that there is a sequence $(C_n)$ and a constant $b$ such that

$$\left| \mathbb{E}\left[ g_n\left( Y^n + \frac{x}{\sqrt{n}} \right) - g_n(Y^n) \right] \right| \leq C_n \max_{k \leq d_n} |x_k|^\alpha, \quad \forall x \in \mathbb{R}^{d_n} \qquad (H_3)$$

$$\min_{j \leq d_n} \|X_{1,j}^n\|_{L_3} \geq b, \qquad \frac{\log(d_n)^{7/6} \| \sup_{k \leq d_n} |X_{1,k}^n| \|_{L_4}^4}{n^{1/6}} = o(1)$$

**Theorem 2** *Let $(g_n : \times_{l=1}^n \mathbb{R}^{d_n} \to \mathbb{R})$ be a sequence of measurable functions satisfying Assumption 1 and $(H_3)$. Denote $\Sigma_n$ the variance-covariance matrix of $X_1^n$ and $(N^n)$ to be a sequence of Gaussian vectors distributed as $N^n \sim N(0, \Sigma_n)$. Let $\beta > 0$ be a real; write $t_{g,n}^{\beta/2}$, and $t_{b,n}^{\beta/2}(X^n)$ as quantities satisfying*

$$P\left( \left| g_n(Z^n) - \mathbb{E}(g_n(Z^n)|X^n) \right| \geq t_{b,n}^{\beta/2}(X^n) \mid X^n \right) \leq \beta/2;$$

$$P\left( \max_k |N_k^n| \geq (t_{g,n}^{\beta/2})^{\frac{1}{\alpha}} C_n^{-\frac{1}{\alpha}} \right) \leq \beta/2.$$

*Then the following holds:*

$$\limsup_{\delta \downarrow 0} \limsup_n P\left( \mathbb{E}(g_n(Y^n)) \notin \left[ g_n(Z^n) - t_{b,n}^{\beta/2} - t_{g,n}^{\beta/2} - \delta, \; g_n(Z^n) + t_{b,n}^{\beta/2} + t_{g,n}^{\beta/2} + \delta \right] \right) \leq \beta.$$

See Appendix N.6 for proof of Theorem 2. We present in Appendix H an illustrative example.

## 4 P-value of a Two-Sample Kernel Test

In this subsection, we show how the bootstrap method can be used to obtain consistent p-values for kernel two sample tests. Given two independent i.i.d processes $(X_{i,1}^n)$ and $(X_{i,2}^n)$ taking value in $\mathcal{X}_n \subset \mathbb{R}^{d_n}$, the goal of two-sample tests is to determine if the two set of observations $(X_{i,1}^n)$ and $(X_{i,2}^n)$ are sampled from the same distribution. For ease of notations, we designate by $\mu_{n,1}$ and $\mu_{n,2}$ respectively the distribution of the first sample $(X_{i,1}^n) \overset{i.i.d}{\sim} \mu_{n,1}$ and of the second sample $(X_{i,2}^n) \overset{i.i.d}{\sim} \mu_{n,2}$; and we want test if the null hypothesis holds

$$(H_0^n) : \mu_{n,1} = \mu_{n,2}$$

against the alternative

$$(H_1^n) : \mu_{n,1} \neq \mu_{n,2}.$$

A popular method to do so are non-parametric kernel two samples tests [42, 31, 52, 51].

Let $\mathcal{F}_n$ be a class of functions from $\mathcal{X}_n$ into $\mathbb{R}$. If the two distributions are the same $\mu_{n,1} = \mu_{n,2}$ then we have:

$$\sup_{f \in \mathcal{F}_n} \left| \mathbb{E}(f(X_{1,1}^n)) - \mathbb{E}(f(X_{1,2}^n)) \right| = 0.$$

Moreover if $\mathcal{F}_n$ is dense in the space of bounded continuous functions then the converse also holds. The main difficulty therefore consists of choosing the set $\mathcal{F}_n$ to be big enough to differentiate between the distributions $\mu_{n,1}$ and $\mu_{n,2}$ but structured enough that we can estimate of $\sup_{f \in \mathcal{F}_n} \left| \mathbb{E}(f(X_{1,1}^n)) - \mathbb{E}(f(X_{1,2}^n)) \right|$. To do so, we choose a reproducing kernel space $\mathcal{H}_n$ with kernel $K_n : \mathcal{X}_n \times \mathcal{X}_n \to \mathbb{R}$ and set the class of functions $\mathcal{F}_n$ to be the unit ball of $\mathcal{H}_n$. Different choices of kernels will lead to various level of power for our test especially for structured or high dimensional data. The goal is to choose the kernel that is the most likely to maximize the power of the test.

Let $(K_{\theta_k}(\cdot, \cdot))_{k \leq p_n}$ be a finite set of potential Kernel candidates. We write for all $i, j \leq n$ and for all $k \leq p_n$

$$H_{i,j}^{\theta_k} := K_{\theta_k}(X_{j,1}^n, X_{i,1}^n) + K_{\theta_k}(X_{j,2}^n, X_{i,2}^n) - K_{\theta_k}(X_{j,1}^n, X_{i,2}^n) - K_{\theta_k}(X_{j,2}^n, X_{i,1}^n);$$

and for all subsets $B \subset [\![n]\!]$ we denote $\hat{M}_{\theta_k}(X_B^n) := \frac{1}{|B|^2} \sum_{i,j \in B} H_{i,j}^{\theta_k}$. The idea proposed in [42] is to select the kernel that gives rise to a test with the highest (estimated) power. This is done by selecting

a subset $B_n \subset [\![n]\!]$ and maximizing the following quantity $\hat{\theta}_n^{B_n} := \text{argmax}_{\theta \in \{\theta_1, \ldots, \theta_{p_n}\}} \, p_\theta(X_{B_n}^n)$ where we have set

$$p_\theta(X_{B_n}^n) := \frac{\hat{M}_\theta(X_{B_n}^n)}{\frac{4}{|B_n|^3} \sum_{i \in B_n} \left[ \sum_{j \in B_n} H_{i,j}^\theta \right]^2 - \frac{4}{|B_n|^4} \left[ \sum_{i,j \leq B_n} H_{i,j}^\theta \right]^2 + \lambda_n}$$

where $(\lambda_n)$ are tuning parameters. Once the kernel is chosen the test statistic is computed on $[\![n]\!] \setminus B_n$ the remaining data: $\frac{1}{n^2} \sum_{i,j \in [\![n]\!] \setminus B_n} H_{i,j}^{\hat{\theta}_n}$. The fact that the kernel is chosen on a different sample than the test statistic is computed on, means that the conditional limiting distribution of the test statistic, under $H_0$, is known to be a chi-square [42]. Hence one can compute a consistent estimate of the p-value. However under this approach only a portion of the data is used to select the kernel. This could be problematic when dealing with high-dimensional kernels.

We propose a different method that does not require data splitting and uses the bootstrap method to estimate the p-value. The test statistic that we propose is a softmax:

$$\hat{T}_n(X^n) := \sum_{k \leq p_n} \frac{1}{n^2} \sum_{i,j \leq n} H_{i,j}^{\theta_k} \, \omega_k(X_{1:n}), \qquad \text{where } \omega_k(X_{1:n}) := \frac{e^{\beta_n p_{\theta_k}(X^n)}}{\sum_{k' \leq p_n} e^{\beta_n p_{\theta'_k}(X^n)}} \, ;$$

and where $(\beta_n)$ are hyper-parameters. The bigger $\beta_n$ is the more weight we give to the kernel maximizing $p_\theta(X^n)$.

We note that the distribution of $\hat{T}_n$ is unknown and depends in an intricate fashion on the set of kernels $\{K_{\theta_k}, \, k \leq p_n\}$ as well as on $p_n$. Therefore to be able to compute the p-value we want to estimate its distribution under $H_0$.

For technical reasons it is convenient to apply the statistic to a random vector whose coordinates are identically distributed. We achieve this by randomly permuting the two samples before passing it to the kernel test statistic (a transformation also conducted in the prior work of [42]). We remark that under the null hypothesis the distribution of $X_{i,1}^n$ and $X_{i,2}^n$ are the same which implies that the samples are interchangeable $\left( X_{i,1}^n, X_{i,2}^n \right) \overset{d}{=} \left( X_{i,2}^n, X_{i,1}^n \right)$. It is therefore natural to compare the distribution of $(X_i^n)$ to the corresponding randomly permuted process. This is the idea behind permutation tests [42]. In general, for an i.i.d random process $(\tilde{X}_i)$ taking value in $\mathbb{R}^2$ we define the process $(\tilde{X}_i^M)$ obtained by randomly permuting the observations $\tilde{X}_{i,1}$ and $\tilde{X}_{i,2}$:

$$\tilde{X}_i^M := \begin{cases} \tilde{X}_i & \text{with probability } 0.5 \\ (\tilde{X}_{i,2}, \tilde{X}_{i,1})^T & \text{with probability } 0.5. \end{cases}$$

We note that this permuted process has identically distributed coordinates, i.e. $\tilde{X}_{1,1}^M \overset{d}{=} \tilde{X}_{1,2}^M$. As a side fact, we note that $d_W \left( \tilde{X}_1, \tilde{X}_1^M \right) \leq d_W \left( \tilde{X}_{1,1}, \tilde{X}_{1,2} \right)$, where $d_W$ is the Wasserstein distance, but, more importantly, if the distribution of $(\tilde{X}_i)$ are already in $H_0$ then its distribution is left invariant by those permutations. We show that the bootstrap gives consistent and asymptotically tight upper-bounds to the p-value even when $p_n$ grows exponentially fast (proof in Appendix Q.1).

**Proposition 2** *Let* $(X_i^n) := \left( (X_{i,1}^n, X_{i,2}^n) \right)$ *be a triangular array of i.i.d processes. Let* $\{K_{\theta_k}, \, k \leq p_n\}$ *be a sequence of positive definite continuous kernels. We suppose that*

$$\max_{k \leq p_n} \text{tr}(K_{\theta_k}) < \infty; \qquad \frac{\beta_n \log(p_n) D_n^4}{\lambda_n^2} = o(n^{1/6});$$

*where we shorthanded* $D_n := \max \left( \left\| \sup_{k \leq p_n} K_{\theta_k}(X_{1,1}^M, X_{1,1}^M) \right\|_{L_{120}}, 1 \right)$. *Let* $(Y_i^n)$ *be an independent copy of* $(X_i^n)$ *and* $(Z_i^n)$ *be bootstrap samples of* $(X_i^n)$. *We have:*

$$\left\| d_{\mathcal{F}} \left( n\hat{T}_n(Z_{1:n}^M), n\hat{T}_n(Y_{1:n}^M) \mid X^n \right) \right\|_{L_1} \to 0.$$

## 5 Empirical Risk of Smooth Stacked Ensemble Estimator

A ubiquitous and popular approach for model selection and ensembling in machine learning practice is known as stacking [53, 8, 50]. Given a set of trained *base estimators* $\{\hat{\theta}^1, \ldots, \hat{\theta}^{p_n}\}$, for example

representing a fitted neural network, a random forest and a nearest-neighbour estimator, we call the *smooth-stacked estimator* the linear ensemble of those estimators $\{\hat{\theta}^k\}$ weighted by coefficients that are related to the out-of-sample risk of each estimator. An important question: if we use all the samples to estimate the weights of the ensemble, then can we construct confidence intervals on the risk of the ensemble estimator?

The most straightforward version of stacking is to put all the weight on the model with the smallest out-of-sample risk. Other approaches proposed in practice are to fit a linear regression model using the outputs of each model as an input co-variate to the linear model and using the learned coefficients as coefficients on the ensemble [50].

In this subsection, we analyze a smooth version of stacking, proposed and analyzed experimentally, for instance, in [38], that adds stability to the chosen ensemble, while putting most weight on the best performing model. This ensemble can be viewed as a regularized instance of the linear regression stacking approach where an entropic regularizer is added to the square loss objective. This regularization adds smoothness and stability to the chosen ensemble and allows us to show that the distribution of the ensemble's risk can be estimated with the bootstrap, even if the all the data are used to estimate the weights or fit the base models.

Let $(X_i^n)$ be a triangular array of i.i.d observations taking value in $\mathbb{R}^{d_n}$; and let $(m_n)$ be an increasing sequence. Define $\mathcal{F}_n$ as the space of measurable functions from $\times_{n=1}^{\infty}\mathbb{R}^{d_n}$ to $\mathbb{R}^{d'_n}$. We estimate $p_n$ different estimators $\Omega_n := \left\{\hat{\theta}_n^k(X_{1:m_n}^n),\ k \leq p_n\right\}$ built on the first $m_n$ data-points. Each estimator $\hat{\theta}_n^k$ is a training algorithm that takes as input $m_n$ samples $X_{1:m_n}^n$ and returns a model, which itself is a function from $\mathbb{R}^{d_n}$ to $\mathbb{R}^{d'_n}$. We denote with $\hat{\theta}_n^k(X_{1:m_n}^n)$ the returned model and with $\hat{\theta}_n^k(X_{1:m_n}^n)(x_u) \in \mathbb{R}^{d'_n}$ the evaluation of the model at a point $x_u \in \mathbb{R}^{d_n}$.

The loss of a model at a sample is measured by a common loss function $\mathcal{L}_n : \mathbb{R}^{d_n} \times \mathbb{R}^{d'_n} \to \mathbb{R}$ and the empirical risk of the $k$-th estimator is computed on all the remaining $n - m_n$ data points as:

$$\mathcal{R}_n^k(x_{1:n}) := \frac{1}{n - m_n} \sum_{u=m_n+1}^{n} \mathcal{L}_n(x_u, \hat{\theta}_n^k(x_{1:m_n})(x_u)).$$

The smooth-stacked estimator is defined as the following ensemble learner

$$\hat{\Theta}_n(x_{1:n})(\cdot) = \sum_{k \leq p_n} \frac{e^{-\beta_n \mathcal{R}_n^k(x_{1:n})}}{\sum_{k' \leq p_n} e^{-\beta_n \mathcal{R}_n^{k'}(x_{1:n})}} \hat{\theta}_n^k(x_{1:m_n})(\cdot).$$

The hyperparameter $\beta_n$ controls how concentrated the stacked estimator is around the estimator(s) $\hat{\theta}_n^k$ with the lowest empirical error. We denote the empirical risk of an ensemble model $\Theta \in (\mathbb{R}^{d_n} \to \mathbb{R}^{d'_n})$ as

$$\mathcal{R}_\Theta^s(x_{1:n}) := \frac{1}{\sqrt{n - m_n}} \sum_{i=m_n+1}^{n} \mathcal{L}_n(x_i, \Theta(x_i)).$$

Let $(Z^n)$ be a bootstrap sample of $(X_i^n)_{i \geq m_n}$. We show that the bootstrap method is systematically consistent if and only if $\beta_n = o(\sqrt{n - m_n})$. For simplicity we suppose that the estimators $\hat{\theta}_n^k$ return models that when evaluated at any point $x_u \in \mathbb{R}^{d_n}$ have bounded coordinates; and that the loss function $\mathcal{L}_n$ is smooth, and have bounded partial derivatives in its second argument.

We write the set of all convex combinations of the estimators: $\Omega(\{\theta_p, p \leq p_n\}) := \left\{\sum_{p \leq p_n} \omega_p \theta_p \mid \omega_p \geq 0 \text{ and } \sum_{p \leq p_n} \omega_p = 1\right\}$ and introduce the following notations:

$$T_n := \sup_{\ell \leq d'_n} \left\|\sup_{p \leq p_n} \left|\hat{\theta}_n^p(X_{1:m_n}^n)(X_n^n)_\ell\right|\right\|_{L_\infty} \vee 1,$$

$$L_n := \left\|\sup_{p \leq p_n} \left|\mathcal{L}_n\left(X_n^n, \hat{\theta}_n^p(X_{1:m_n}^n)(X_n^n)\right)\right|\right\|_{L_\infty} \vee \sup_{\ell \leq d'_n} \left\|\sup_{\theta \in \Omega(\{\hat{\theta}_p(X_{1:m_n}^n),\ p \leq p_n\})} \partial_{2,\ell} \left|\mathcal{L}_n(X_n^n, \theta(X_n^n))\right|\right\|_{L_\infty} \vee 1,$$

where by $\partial_{2,\ell}\mathcal{L}_n(x,y)$ we designate $\partial_{y_i}\mathcal{L}_n(x,y)$ We show that if the following hypothesis ($H_1^{\text{stacked}}$) holds then the bootstrap method is asymptotically consistent (proof in Appendix R).

$$\frac{\beta_n d'_n}{\sqrt{n-m_n}} L_n T_n \, e^{\frac{\beta_n}{n-m_n}L_n} \longrightarrow 0. \qquad (H_1^{\text{stacked}})$$

**Proposition 3** *Choose* $(m_n)$, $(\beta_n)$ *and* $(p_n)$ *be increasing sequences. Let* $(X_i^n)$ *be a triangular array of i.i.d observations taking value in* $\mathbb{R}^{d_n}$. *Set* $(\mathcal{L}_n : \mathbb{R}^{d_n} \times \mathbb{R}^{d'_n} \to \mathbb{R})$ *to be a sequence of smooth loss functions. Let* $(Z_i^n)$ *and* $(Y_i^n)$ *be respectively a bootstrap sample and an independent copy of* $(X_{m_n+1}^n, \ldots, X_n^n)$. *Suppose that the hypothesis* ($H_1^{\text{stacked}}$) *holds then we have:*

$$\left\| d_\mathcal{F}\left( \mathcal{R}^{\text{s}}_{\hat{\Theta}_n}(Z^n_{m_n+1:n}) - \mathbb{E}\big[\mathcal{R}^{\text{s}}_{\hat{\Theta}_n}(Z^n_{m_n+1:n})\big|\hat{\Theta}_n\big], \ \mathcal{R}^{\text{s}}_{\hat{\Theta}_n}(Y^n_{m_n+1:n}) - \mathbb{E}\big[\mathcal{R}^{\text{s}}_{\hat{\Theta}_n}(Y^n_{m_n+1:n})\big|\hat{\Theta}_n\big] \mid X^n \right) \right\|_{L_1} \to 0;$$

*where we have shorthanded* $\hat{\Theta}_n := \hat{\Theta}_n(X^n)$. *Therefore if we choose* $t_{n,\alpha}(X^n)$ *to be such that:*

$$P\left( \left| \mathcal{R}^{\text{s}}_{\hat{\Theta}_n}(Z^n_{m_n+1:n}) - \mathbb{E}\big[\mathcal{R}^{\text{s}}_{\hat{\Theta}_n}(Z^n_{m_n+1:n})\big|\hat{\Theta}_n\big] \right| \geq t_{n,\alpha}(X^n) \mid X^n \right) \leq \alpha$$

*then the following holds*

$$\limsup_{n\to\infty} P\left( \left| \mathcal{R}^{\text{s}}_{\hat{\Theta}_n}(Y^n_{m_n+1:n}) - \mathbb{E}\big[\mathcal{R}^{\text{s}}_{\hat{\Theta}_n}(Y^n_{m_n+1:n})\big|\hat{\Theta}_n\big] \right| \geq t_{n,\alpha}(X^n) \mid \hat{\Theta}_n \right) \leq \alpha$$

If $\beta_n$ grows proportionally to $\beta_n \propto \sqrt{n-m_n}$ then the bootstrap method is not a systematically consistent estimator of the risk of the smooth stacked estimator. We present a simple example illustrating this in Appendix K and establish the asymptotic distribution of the bootstrap. However, we show that using Theorem 2 we can still propose a corrected confidence interval with guaranteed asymptotic coverage.

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
