# Contents of Technical Appendix

## A   Preliminary Lemmas and Notation

If a function $f$ is three-times differentiable then we let:

$$\partial_{i,k} f(x_{1:n}) := \partial_{x_{i,k}} f(x_{1:n}) \qquad \partial_i f(x_{1:n}) := (\partial_{i,1} f(x_{1:n}), \dots, \partial_{i,d} f(x_{1:n}))^\top$$

$$\partial^2_{i,k_{1:2}} f(x_{1:n}) := \partial_{x_{i,k_1}} \partial_{x_{i,k_2}} f(x_{1:n}) \qquad \partial^2_i f(x_{1:n}) := \left(\partial^2_{i,k_{1:2}} f(x_{1:n})\right)_{k_1,k_2 \le d}$$

$$\partial^3_{i,k_{1:3}} f(x_{1:n}) := \partial_{x_{i,k_1}} \partial_{x_{i,k_2}} \partial_{x_{i,k_3}} f(x_{1:n}) \quad \partial^3_i f(x_{1:n}) := \left(\partial^3_{i,k_{1:3}} f(x_{1:n})\right)_{k_1,k_2,k_3 \le d}$$

### A.1   Preliminary results

**Lemma 3** *Let $(\tilde{X}_i^n)$ be an array of martingale differences taking value in $\mathbb{R}^{p_n}$. Suppose that $\left\| \max_{k \le p_n} |\tilde{X}_{1,k}^n| \right\|_{L_p} < \infty$ where $p \ge 3$. Then there exists a constant $C_p$, that does not depend on the distribution of $(\tilde{X}_i^n)$, such that*

$$\left\| \max_{k \le p_n} \left| \frac{1}{\sqrt{n}} \sum_{i \le n} \tilde{X}_{i,k}^n \right| \right\|_{L_p} \le \left\| \max_{k \le p_n} |\tilde{X}_{i,k}^n| \right\|_{L_p} \left( 1 + C_p \left( \log(p_n) + \frac{\log(p_n)^2}{\sqrt{n}} \right) \right)$$

*Thus if $\log(p_n) = o(n^{1/4})$ then*

$$\left\| \max_{k \le p_n} \left| \frac{1}{\sqrt{n}} \sum_{i \le n} \tilde{X}_{i,k}^n \right| \right\|_{L_p} = O\left( \log(p_n) \left\| \sup_{k \le p_n} \tilde{X}_{i,k}^n \right\|_{L_p} \right).$$

*Moreover let $(X_i^n)$ be a triangular array of i.i.d process and $(g_{k,n})$ be sequences of measurable functions, for each $k \in [p]$. Then:*

$$\left\| \max_{k \le p_n} g_{k,n}(X^n) \right\|_{L_p} = O\left( \log(p_n) \sqrt{n} \sup_{i \le n} \left\| \sup_{k \le p_n} \left| g_{k,n}(X^n) - g_{k,n}(X^{n,i}) \right| \right\|_{L_p} \right),$$

*where we have defined* $X_j^{n,i} := \begin{cases} X_j^n, & \text{if } j \neq i \\ X_i', & \text{if } i = j \end{cases}$ *with* $(X_i')$ *being an independent copy of* $(X_i^n)$.

See Appendix T.1 for proof of Theorem 3.

**Lemma 4** *For any set of random variables* $U_1, \ldots, U_m$:

$$\mathbb{E}\left[\left(\sum_{t \leq m} U_t\right)^d\right] \leq \left(\sum_{t \leq m} \|U_t\|_{L_d}\right)^d$$

**Proof:** By expanding the polynomial, applying a repeated version of Cauchy-Schwarz inequality and collapsing the polynomial again, we get:

$$\mathbb{E}\left[\left(\sum_{t \leq m} U_t\right)^d\right] = \mathbb{E}\left[\sum_{t_{1:d} \leq m} \prod_{\ell=1}^d U_{t_\ell}\right] \leq \sum_{t_{1:d} \leq m} \prod_{\ell=1}^d \|U_{t_\ell}\|_{L_d} \leq \left(\sum_{t \leq m} \|U_t\|_{L_d}\right)^d$$

$\square$

**Lemma 5** *The distribution distance* $d_{\mathcal{F}}$ *satisfies the triangle inequality.*

**Proof:** For any three random variables $X, Y, Z$:

$$
\begin{aligned}
d_{\mathcal{F}}(X, Z) &:= \sup_{h \in \mathcal{F}} \mathbb{E}[h(X)] - \mathbb{E}[h(Z)] \\
&= \sup_{h \in \mathcal{F}} \mathbb{E}[h(X)] - \mathbb{E}[h(Y)] + \mathbb{E}[h(Y)] - \mathbb{E}[h(Z)] \\
&\leq \sup_{h \in \mathcal{F}} \mathbb{E}[h(X)] - \mathbb{E}[h(Y)] + \sup_{h \in \mathcal{F}} \mathbb{E}[h(Y)] - \mathbb{E}[h(Z)] =: d_{\mathcal{F}}(X, Y) + d_{\mathcal{F}}(Y, Z)
\end{aligned}
$$

$\square$

**Lemma 6** *The distribution distance* $d_{\mathcal{F}}$ *is translation invariant: For all random variables* $X$ *and* $Y$ *and all constant* $z$ *we have*

$$d_{\mathcal{F}}(X, Y) = d_{\mathcal{F}}(X - z, Y - z)$$

**Proof:** For all $h \in \mathcal{F}$ define $h_z x \to h(x - z)$. We have:

$$\mathbb{E}h(X - z) - h(Y - z) = \mathbb{E}h_z(X) - h_z(Y) \leq d_{\mathcal{F}}(X, Y).$$

As this holds for all $h \in \mathcal{F}$ it implies that

$$d_{\mathcal{F}}(X - z, Y - z) \leq d_{\mathcal{F}}(X, Y).$$

The reverse inequality is proved in exactly the same fashion. $\square$

**Lemma 7** *Let* $p_1, p_2$ *be two distributions that are uniformly continuous with respect to a measure* $\mu$. *Then the following holds:*

$$1 - \frac{1}{2}e^{-KL(p_2, p_1)} \geq \|p_1(\cdot) - p_2(\cdot)\|_{TV}$$

**Proof:** We denote $f_1, f_2$ the Radon-Nikodym densities of respectively $p_1$ and $p_2$ with respect to $\mu$. By the Cauchy-Swartz inequality we have:

$$
\begin{aligned}
&\left(\int \sqrt{f_1(x)f_2(X)}d\mu(x)\right)^2 \\
&\leq \left(\int \sqrt{\min(f_1(x), f_2(x))\max(f_1(x), f_2(X))}d\mu(x)\right)^2 \\
&\leq \int \min(f_1(x), f_2(x))d\mu(x) \int \max(f_1(x), f_2(X))d\mu(x) \\
&\leq \int \min(f_1(x), f_2(x))d\mu(x) \left(2 - \int \min(f_1(x), f_2(X))\right)d\mu(x) \\
&\leq 2(1 - \|p_1(\cdot) - p_2(\cdot)\|_{TV}).
\end{aligned}
$$

Moreover by another application of Cauchy-Swartz we know that

$$\left( \int \sqrt{f_1(x)f_2(X)}d\mu(x) \right)^2$$

$$= e^{\log\left( \left( \int \sqrt{f_1(x)f_2(X)}d\mu(x) \right)^2 \right)}$$

$$= e^{2\log\left( \int \frac{\sqrt{f_1(x)}}{\sqrt{f_2(x)}} f_2(X)d\mu(x) \right)}$$

$$\leq e^{-KL(p_2,p_1)}$$

Therefore by combining those two inequalities we obtain that:

$$1 - \frac{1}{2}e^{-KL(p_2,p_1)} \geq \|p_1(\cdot) - p_2(\cdot)\|_{TV}$$

$\square$

# B    Extensions and Variations of Main Theorem

In this section we present some supplementary results that have been motivated in the main body of the article.

## B.1    Alternative Condition to $(H_1)$

As mentioned in the main body of the text the results also hold under a slightly modified condition $(H_1)$. We denote $\|\cdot\|_{v,d_n}$, $\|\cdot\|_{m,d_n}$ and $\|\cdot\|_{t,m_n}$ respectively the $L_1$ norm for $d_n$ dimensional vectors, $d_n \times d_n$ dimensional matrices and $d_n \times d_n \times d_n$ dimensional tensors. We define the following quantities:

$$R_{1,n}^* := 2n^{\frac{1}{3}} \sup_{i \leq n} \left\| \left\| \partial_i f_n(Z_{1:n}^{n,i,\bar{X}^n}) \right\|_{v,d_n} \right\|_{L_{12}} \left\| \sup_{i \leq d_n} |X_{1,i}^n| \right\|_{L_{12}}$$

$$R_{2,n}^* := 4n^{1/2} \sup_{i \leq n} \left\| \left\| \partial_i^2 f_n(Z^{n,i,\bar{X}^n}) \right\|_{m,d_n} \right\|_{L_{12}} \left\| \sup_{i \leq d_n} |X_{1,i}^n| \right\|_{L_{12}}^2$$

$$R_{3,n}^* := 8n \sup_{i \leq n} \left\| \max_{x \in [\bar{X}^n, \tilde{Y}_i^n] \cup [\bar{X}^n, Z_1^n]} \left\| \partial_i^3 f_n(Z^{n,i,x}) \right\|_{t,d_n} \right\|_{L_{12}} \left\| \sup_{i \leq d_n} |X_{1,i}^n| \right\|_{L_{12}}^4.$$

We assume that the functions $(f_n)$ satisfy the following conditions:

$$\log(d_n)\left( (R_{n,1}^*)^2 \max\left\{ \frac{1}{n^{1/6}}, R_{n,1}^* \right\} + R_{n,2}^* \right) + R_{n,3}^* \to 0. \qquad (H_1^*)$$

**Theorem 8** *Let $(g_n : \times_{l=1}^n \mathbb{R}^{d_n} \to \mathbb{R})$ be a sequence of symmetric measurable symmetric functions. Let $(X_i^n)$ be a triangular array of i.i.d processes such that $X_1^n \in L_{12}$. Assume that there is a sequence $(f_n)$ of measurable functions satisfying condition $(H_0)$ and $(H_1^*)$. Then there exists a universal constant $K$ such that:*

$$\left\| d_{\mathcal{F}}\left( g_n(Z^n), g_n(\tilde{Y}^n) \mid X^n \right) \right\|_{L_1} \leq \left\| g_n(\tilde{Y}^n) - f_n(\tilde{Y}^n) \right\|_{L_1} + \left\| g_n(Z^n) - f_n(Z^n) \right\|_{L_1}$$

$$+ K\left( \log(d_n)\left( (R_{n,1}^*)^2 \max\left\{ \frac{1}{n^{1/6}}, R_{n,1}^* \right\} + R_{n,2}^* \right) + R_{n,3}^* \right)$$

Similarly Theorem 2 also holds under the hypothesis $H_1^*$. In addition we generalize it to the setting where the variance-covariance matrix of $(X_i^n)$ is unknown but can be estimated.

**Theorem 9** *Let $(g_n : \times_{l=1}^n \mathbb{R}^{d_n} \to \mathbb{R})$ be a sequence of measurable functions satisfying hypothesis $(H_0)$, $(H_1^*)$ and $(H_3)$. Denote $\Sigma_n^2$ the variance-covariance matrix of $X_1^n$ and by $\hat{\Sigma}_n^2$ an estimator of $\Sigma_n^2$, and $(N^n)$ to be a sequence of Gaussian vectors distributed as $N^n \sim N(0, \mathrm{Id})$. Suppose that $\hat{\Sigma}_n$ is independent from $(X_i^n)$ and that it verifies:*

$$\sup_t \left| P\left( \max_k |(\hat{\Sigma}_n N^n)_k| \geq t \right) - P\left( \max_k |(\Sigma_n N^n)_k| \geq t \right) \right| \to 0.$$

*Let $\beta > 0$ be a real; write $\hat{t}_{g,n}^{\beta/2}$, and $t_{b,n}^{\beta/2}(X^n)$ as quantities satisfying*

$$P\left(\left|g_n(Z^n) - \mathbb{E}(g_n(Z^n)|X^n)\right| \geq t_{b,n}^{\beta/2}(X^n) \mid X^n\right) \leq \beta/2;$$

$$P\left(\max_k |(\hat{\Sigma}_n N^n)_k| \geq (\hat{t}_{g,n}^{\beta/2})^{\frac{1}{\alpha}} C_n^{-\frac{1}{\alpha}}\right) \leq \beta/2.$$

*Then the following holds:*

$$\limsup_{\delta \downarrow 0} \limsup_n P\left(\mathbb{E}(g_n(Y^n)) \leq \left[g_n(Z^n) - t_{b,n}^{\beta/2} - \hat{t}_{g,n}^{\beta/2} - \delta, \ g_n(Z^n) + t_{b,n}^{\beta/2} + \hat{t}_{g,n}^{\beta/2} + \delta\right]\right) \leq \beta.$$

## B.2 Extension to Exchangeable Sequences and Random Estimators $(g_n)$

In this subsection we generalize theorem 1 to random estimators $(g_n)$ and to exchangeable processes $(X_i^n)$. We say that a process $(X_i^n)$ is exchangeable if and only if for all permutations $\pi \in \mathbb{S}(\mathbb{N})$ and all indexes $i_1, \ldots, i_k \in \mathbb{N}$ we have:

$$(X_{i_1}^n, \ldots, X_{i_k}^n) \overset{d}{=} (X_{\pi(i_1)}^n, \ldots, X_{\pi(i_k)}^n).$$

We designate by $\tau(X^n)$ the tail $\sigma-$algebra of $X^n$ which is defined as $\tau(X^n) := \bigcap_{i=1}^{\infty} \sigma\left(X_k^n, \ k \geq i\right)$. By the De Finitti theorem we know that $(X_i^n)$ is exchangeable if and only if conditionally on $\tau(X^n)$ the process $(X_i^n)$ is an i.i.d process.

We assume that the sequence of (potentially random) functions $(g_n)$ is such that there is a net of (potentially random) three-times differentiable functions $(f_n)$ respecting conditions $(H_0)$ and $(H_1)$. We establish under those conditions the limiting distribution of the bootstrap estimator.

**Theorem 10** *Let $(g_n : \times_{l=1}^n \mathbb{R}^{d_n} \to \mathbb{R})$ be a sequence of measurable functions. Let $(X_i^n)$ be a triangular array of exchangeable processes such that $X_1^n \in L_{12}$. Assume that there is a net $(f_n)$ of (potentially random) functions satisfying Assumption 1. Let $Y^n := (Y_i^n)$ be a process that is, conditionally on $\tau(X^n)$, an independent copy of $X^n$ that is also independent from $(g_n)$. Define $(Z^n)$ to be a boostrap sample of $X^n$ that is independent of $(g_n)$ conditionally on $X^n$. Then there exists a universal constant $K$ such that:*

$$\left\|d_{\mathcal{F}}\left(g_n(Z^n), g_n(\tilde{Y}^n) \mid X^n\right)\right\|_{L_1} \leq \left\|g_n(\tilde{Y}^n) - f_n(\tilde{Y}^n)\right\|_{L_1} + \|g_n(Z^n) - f_n(Z^n)\|_{L_1}$$
$$+ K\left(R_{n,1}^2 \max\left(\frac{1}{n^{1/6}}, R_{n,1}\right) + R_{n,3} + R_{n,2}\right)$$
$$\longrightarrow 0.$$

**Remark 4** *We note that Theorem 12, Theorem 11, Theorem 15 and Theorem 8, can be generalized in the exact same fashion.*

## C  Counter-examples when Assumption 1 does not hold

Firstly we note that if $(H_0)$ and $(H_1)$ hold we have $\|g_n(X^n) - g_n(0X_{2:n}^n)\|_{L_3} = o(n^{-1/3})$. This is a first-order stability property, i.e. that each sample $i$'s influence on the estimate has to decay at rate $n^{-1/3}$. Our first example is chosen to violate this.

**Example 1** *Let $(X_i^n)$ be a sequence of i.i.d observations distributed as $X_i \sim \text{unif}(0,1)$. Let $(g_n)$ be the following sequence of functions: $g_n(x_{1:n}) := n \min_{i \leq n} x_i$. Then neither the bootstrap method nor the centered bootstrap method are consistent. Moreover, we note that: $\|g_n(X_{1:n}) - g_n(0X_{2:n})\|_{L_3} \propto n^{-1/3}$. In this example, the bootstrap estimator $g_n(Z^n) \geq g_n(X)$ is systematically larger than the original statistic, which leads to inconsistency of the bootstrap distribution.*

Another consequence of having the second and third order derivative of respective order $o(n^{-1/2})$ and $o(n^{-1})$ is that the following two conditional expectations are very similar:

$$\left\|\mathbb{E}\left[g_n(Z^n) \mid X^n\right] - \mathbb{E}\left[g_n(\tilde{Y}^n) \mid X^n\right]\right\|_{L_1} = o(1). \tag{1}$$

Our second example is chosen to satisfy the main implication of the first order stability conditions: i.e. $\|g_n(X^n) - g_n(0X_{2:n}^n)\|_{L_3} = o(n^{1/3})$; but to fail to respect this new property.

**Example 2** *Let $(X_i^n)$ be a sequence of i.i.d observations distributed as $X_i^n \sim \text{unif}(0,1)$. Let $(g_n)$ be the following sequence of functions:*

$$g_n(x_{1:n}) := \frac{1}{\sqrt{n}} \sum_{i \le n} 1\left\{\min_{i \ne j} |x_i - x_j| > 1/n\right\} - P\left(\min_{i \ne 1} |X_1 - X_i| > 1/n\right).$$

*Then neither the bootstrap nor the centered bootstrap are consistent. Moreover we note that the first order stability result holds, but $\|g_n(Y_{1:n}) - g_n(0Y_{2:n})\|_{L_3} \propto n^{-1/2}$ and Condition (1) is violated. The main driving force of inconsistency in this example is that contrary to the original sample, it is likely that the bootstrap sample will contain repeats. Hence we expect $P\left(\min_{i \ne 1} |Z_i^n - Z_j^n| > 1/n \mid X^n\right)$ to be smaller than $P\left(\min_{i \ne 1} |X_1 - X_i| > 1/n\right)$. See Appendix N.1 for formal proof.*

# D Illustrative Examples and Counterexamples

We present a sequence of simple examples illustrating that our theorems hold even if the estimator is not asymptotically normal. Moreover, we provide negative examples where the shape of the confidence intervals obtained by the bootstrap method can be arbitrary compared to the ones of the original statistics $g_n(Y_{1:n})$. The first example we consider are polynomials of the empirical average. Their limiting distribution is in not Gaussian for $p > 1$.

**Example 3** *Let $p \in \mathbb{N}$ be an integer and let $X := (X_i)$ be an i.i.d sequence taking value in $\mathbb{R}$ with mean $0$ and admitting a $12p$-th moment $\mathbb{E}\left[|X_i|^{12p}\right] < \infty$. We define the functions $(g_n)$ as $g_n : x_{1:n} \to \left(\frac{1}{\sqrt{n}}\sum_{i \le n} x_i\right)^p$. We write $(Z_i^n)$ a bootstrap sample and $(Y_i^n)$ an independent copy of $X$. Then the following holds:*

$$\left\|d_{\mathcal{F}}\left(g_n(Z^n), \left(\sqrt{n}\,\bar{X}^n + \sqrt{n}\,\bar{Y}^n\right)^p \mid X\right)\right\|_{L_1} = O\left(\frac{1}{\sqrt{n}}\right).$$

*Moreover, let $A$ be an $1 - \alpha$ confidence-interval for $g_n(\tilde{Y}_{1:n})$ meaning $P(g_n(\tilde{Y}_{1:n}) \in A) \ge 1 - \alpha$. Write: $A_{\bar{X}^n} := \left\{x \in \mathbb{R} \mid \exists y \in A \text{ s.t } x = \text{sign}(x)\right] \left(|y|^{1/p} - \bar{X}^n\right)^p\right\}$ then*

$$P(g_n(Y_{1:n}) \in A_{\bar{X}^n}) \ge 1 - \alpha.$$

See Appendix O.1 for the proof.

**Example 4** *Let $X := (X_i)$ be an i.i.d sequence of bounded real valued random variables satisfying $\mathbb{E}(X_1) = 0$. We define $g_n : \times_{l=1}^n \mathbb{R} \to \mathbb{R}$ to be the following function: $g_n(x_1, \ldots, x_n) := \sqrt{n}\left[\prod_{i=1}^n\left(1 + \frac{x_i}{n}\right) - 1\right]$. Write $(\tilde{Z}_i^n)$ to be a centered bootstrap sample and let $Y := (Y_i)$ be an independent copy of $X$. Then the following holds:*

$$\left\|d_{\mathcal{F}}\left(g_n(\tilde{Z}^n) - \mathbb{E}(g_n(\tilde{Z}^n)|X), g_n(Y_{1:n}) - \mathbb{E}(g_n(Y^n))\Big|X\right)\right\|_{L_1} \to 0.$$

*See Appendix O.2 for a formal proof.*

The next example demonstrates that the confidence intervals obtained by the bootstrap method are neither systematically bigger or smaller than the ones of original statistics.

**Example 5** *Let $(X_i)$ be a sequence of i.i.d standard normal observations $X_i \sim N(0,1)$. Define $(g_n)$ to be the following sequence of functions: $g_n(x_{1:n}) := \left[\frac{1}{\sqrt{n}}\sum_{i \le n} x_i\right]^+$. Let $(Z^n)$ be a bootstrap sample. The following holds:*

$$\left\|d_{\mathcal{F}}\left(g_n(Z^n), \sqrt{n}\left[\bar{Y} + \bar{X}\right]^+ \mid X\right)\right\|_{L_1} \to 0.$$

*Moreover given $\alpha < 0.5$ and a sequence $(t_n)$ such that: $P(g_n(Z^n) \le t_n \mid X^n) = 1 - \alpha$ then:*

$$P(g_n(Y^n) \le t_n - \sqrt{n}\bar{X}^n) \sim 1 - \alpha.$$

*We notice that the segment $[0, t]$ is smaller than $[0, t - \sqrt{n}\bar{X}^n]$ only if $\sqrt{n}\bar{X}^n > 0$ which asymptotically happens with a probability of $1/2$.*

See Appendix O.3 for a formal proof.

In the next example we show that our results apply to classical quantities in mathematical physics. We consider the entropy of spin glasses configurations.

**Example 6** *Let $X := (X_{i,j})$ be an array of i.i.d observations satisfying $X_{i,j} \overset{i.i.d}{\sim} N(0,1)$. We denote $X^n := (X_{i,j})_{i,j \le n}$ the induced matrix and define $g_n : M_n(\mathbb{R}) \to \mathbb{R}$ to be the following function: $g_n(X) := \frac{1}{n} \log \left( \sum_{m \in \{-1,1\}^n} e^{\frac{1}{\sqrt{n}} m^\top X^n m} \right)$. Write $(Z_{i,j}^n)$ and $(\tilde{Z}_{i,j}^n)$ respectively a bootstrap and centered bootstrap sample and $Y^n$ an independent copy of $X^n$. Then the following holds:*

$$\left\| d_\mathcal{F} \left( g_n(Z^n), \frac{1}{n} \log \left( \sum_{m \in \{-1,1\}^n} e^{\frac{1}{\sqrt{n}} m^\top Y^n m} e^{\bar{X}^n (\sum_i m_i)^2 / \sqrt{n}} \right) \mid X^n \right) \right\|_{L_1} \to 0;$$

*and*

$$\left\| d_\mathcal{F} \left( g_n(\tilde{Z}^n), \frac{1}{n} \log \left( \sum_{m \in \{-1,1\}^n} e^{\frac{1}{\sqrt{n}} m^\top Y^n m} \right) \mid X^n \right) \right\|_{L_1} \to 0;$$

*where we have denoted $\bar{X}^n := \frac{1}{n^2} \sum_{i,j \le n} X_{i,j}^n$.*

See Appendix O.4 for a formal proof.

## E   Consistency of the Centered-Bootstrap

First we explore the case when we know the mean of the observations $\mathbb{E}[X_1^n]$. In this case, we can leverage this knowledge to build centered bootstrap samples $\tilde{Z}_i^n := Z_i^n + \mathbb{E}[X_1^n] - \bar{X}^n$. Observe that these centered samples satisfy the crucial property that $\mathbb{E}\left[ \tilde{Z}_1^n \mid X^n \right] = \mathbb{E}(X_1^n)$. We prove that, under mild conditions, the centered bootstrap estimator is asymptotically consistent. The conditions needed for the centered bootstrap to be consistent are hypothesis $(H_0)$ and $(H_1)$ formulated instead for $(Y_i^n)$ and $(\tilde{Z}_i^n)$ rather than for $(\tilde{Y}_i^n)$ and $(Z_i^n)$.

**Assumption 2 (Approximation by $\mathbb{C}^3$ of $g_n(\cdot + \mathbb{E}(X_1^n) - \bar{X}^n)$.)** *There exists a sequence of functions $(f_n)$ with $f_n \in \mathbb{C}^3$ s.t.:*

1. *The functions $(f_n)$ approximate the estimators $(g_n)$:*

$$\sup_n \left\| f_n(\tilde{Z}^n) - g_n(\tilde{Z}^n) \right\|_{L_1} + \left\| f_n(Y^n) - g_n(Y^n) \right\|_{L_1} \xrightarrow{\beta \to \infty} 0. \qquad (H_0^c)$$

2. *The first, second and third order derivatives are respectively of size $o(n^{-1/3})$, $o(n^{-1/2})$, $o(n^{-1})$:*

$$R_{n,1}^c := n^{1/3} \sum_{k_1 \le d_n} D_{1,k_1}^n \left( f_n(\cdot + \mathbb{E}(X_1^n) - \bar{X}^n) \right) = o(1);$$

$$R_{n,2}^c := \sqrt{n} \sum_{k_1,k_2 \le d_n} D_{2,k_{1:2}}^n \left( f_n(\cdot + \mathbb{E}(X_1^n) - \bar{X}^n) \right) = o(1); \qquad (H_1^c)$$

$$R_{n,3}^c := n \sum_{k_1,k_2,k_3 \le d_n} D_{3,k_{1:3}}^n \left( f_n(\cdot + \mathbb{E}(X_1^n) - \bar{X}^n)) \right) = o(1).$$

We show that under those conditions the centered bootstrap is asymptotically consistent and thereby can be used to build confidence intervals with asymptotically nominal coverage.

**Theorem 11** *Let $(g_n : \times_{l=1}^n \mathbb{R}^{d_n} \to \mathbb{R})$ be a sequence of measurable functions. Let $(X_i^n)$ be a triangular array of i.i.d processes such that $X_1^n \in L_{12}$. Assume that there is a sequence $(f_n :$*

$\times_{l=1}^{n} \mathbb{R}^{d_n} \to \mathbb{R}$) *of measurable functions satisfying conditions* $(H_0^c)$ *and* $(H_1^c)$*. Then there is a universal constant* $K$ *such that:*

$$\left\| d_{\mathcal{F}} \left( g_n(\tilde{Z}^n), g_n(Y^n) \mid X^n \right) \right\|_{L_1} \leq \left\{ \begin{array}{l} \|g_n(Y^n) - f_n(Y^n)\|_{L_1} + \left\| g_n(\tilde{Z}^n) - f_n(\tilde{Z}^n) \right\|_{L_1} \\ + K \left( (R_{n,1}^c)^2 \max \left\{ \dfrac{1}{n^{1/6}}, R_{n,1}^c \right\} + R_{n,3}^c + R_{n,2}^c \right) \end{array} \right\} \to 0.$$

**Example 7 (Application to hypothesis testing)** *An important application is hypothesis testing. Suppose we want to test* $(H_0):$ $\mathbb{E}[X_1^n] = \theta$ *against an alternative* $(H_1)$*. In this goal, we build a test statistic* $\hat{T}_n(X^n)$ *for which we want to compute a p-value. Let* $(Z_i^n)$ *be a bootstrap sample of* $\{X_1^n, \ldots, X_n^n\}$*; define* $(Z_i^\theta)$ *as the following process:*

$$Z_i^\theta := Z_i^n - \bar{X}^n + \theta.$$

*We remark that under the null,* $(Z_i^\theta)$ *is a centered bootstrap sample of* $X^n$*. Using Theorem 11 we know, under stability conditions on* $(\hat{T}_n)$ *(i.e. Assumption 2), that we can use* $\hat{T}^n(Z_{1:n}^\theta)$ *to estimate the p-value of* $\hat{T}_n$*.*

**Proposition 4** *Let* $(X_i^n)$ *be a triangular array of i.i.d processes taking value in* $\mathbb{R}^{d_n}$ *and* $\hat{T}_n :$ $\times_{i=1}^{n} \mathbb{R}^{d_n} \to \mathbb{R}$ *be a sequence of measurable functions that satisfies Assumption 2. Then:*

$$\left\| d_{\mathcal{F}} \left( \hat{T}_n(Z_{1:n}^\theta), \hat{T}_n(Y_{1:n}^{n,\theta}) \mid X^n \right) \right\|_{L_1} \to 0.$$

# F    Stable Estimators to Uniform Perturbations

In this section we explore conditions guaranteeing that the distribution of $g_n(Y^n) - \mathbb{E}[g_n(Y^n)]$ is asymptotically the same as the distribution of $g_n(\tilde{Y}^n) - \mathbb{E}\left[g_n(\tilde{Y}^n) \mid X^n\right]$, conditional on $X^n$, as this would imply that the bootstrap method provides consistent confidence intervals for $\mathbb{E}[g_n(Y^n)]$. We start by noting that if $(g_n)$ are linear then it automatically holds as we have

$$\sum_{i \leq n} \tilde{Y}_i^n - \mathbb{E}\left[ \sum_{i \leq n} \tilde{Y}_i^n \mid X^n \right] = \sum_{i \leq n} Y_i^n - \mathbb{E}\left[ \sum_{i \leq n} Y_i^n \mid X^n \right].$$

Observe that the random variables $\tilde{Y}_i^n$ differ from $Y_i^n$ in a benign manner: a random offset $\bar{X}^n - \mathbb{E}[X_1^n]$, which is independent of $Y^n$, is added to all the random variables. Moreover, this offset is with high probability $O(n^{-1/2})$, since it is the difference of a sample and a population mean. We will refer to such perturbations of a sample $Y^n$ as a *uniform perturbation*. To study general statistics, we introduce the following assumption which guarantees that small uniform perturbations do not drastically change the distribution of $g_n(Y^n)$:

**Assumption 3 (Stability to Uniform Perturbation)** *A statistic sequence* $(g_n)$ *is stable to small uniform perturbations if for all* $B > 0$*:*

$$r^{n,B} := \left\| \sup_{x \in B_{d_n}(0,B)} \left| g_n(X^n + x/\sqrt{n}) - g_n(X^n) - \mathbb{E}\left[ g_n(X^n + x/\sqrt{n}) - g_n(X^n) \right] \right| \right\|_{L_1} \overset{n \to \infty}{\to} 0$$

$$(H_2)$$

*where we define* $B_{d_n}(0, B) := \{x \in \mathbb{R}^{d_n} \mid \|x\|_2 \leq B\}$*.*

Note that the perturbations considered in hypothesis $(H_2)$ are uniform on all the coordinates $i \in [n]$. This notably implies that if $g_n$ depends only the relative distance between the observations then hypothesis $(H_2)$ holds. We prove, under hypothesis $(H_2)$, that the bootstrap method is consistent and hence by Proposition 1 can be used to build asymptotically consistent confidence intervals for $\mathbb{E}[g_n(X^n)]$ (proof in Appendix N.2).

**Theorem 12** *Let $(g_n : \times_{l=1}^n \mathbb{R}^{d_n} \to \mathbb{R})$ be a sequence of measurable functions. Let $(X_i^n)$ be a triangular array of i.i.d processes such that $X_1^n \in L_{12}$. Assume that $(g_n)$ satisfies Assumption 1 and Assumption 3. Then there exists a universal constant $K$ such that:*

$$\left\| d_{\mathcal{F}} \left( g_n(Z^n) - \mathbb{E}\big[g_n(Z^n)|X^n\big], \, g_n(Y^n) - \mathbb{E}(g_n(Y^n)) \mid X^n \right) \right\|_{L_1}$$

$$\leq \inf_{B_n \in \mathbb{R}} \left\{ \begin{array}{l} \left\| g_n(\tilde{Y}^n) - f_n(\tilde{Y}^n) \right\|_{L_1} + \left\| g_n(Z^n) - f_n(Z^n) \right\|_{L_1} + K (R_{n,1})^2 \max\left\{ \dfrac{1}{n^{1/6}}, R_{n,1} \right\} \\[3mm] + K \left( R_{n,3} + R_{n,2} \right) + \dfrac{2\sqrt{\sum_{k \leq k_n} \|X_{1,k}^n\|_{L_2}^2}}{B_n} \left[ \left\| g_n(Y^n) \right\|_{L_2} + \left\| g_n(\tilde{Y}^n) \right\|_{L_2} \right] + r^{n,B_n} \end{array} \right\} \to 0.$$

Condition $(H_2)$ holds beyond linear statistics. We present two simple illustrative examples of such non-linear estimators, for which $(H_2)$ is satisfied.

**Example 8** *Let $(X_i)$ be an i.i.d sequence of random variables taking value in $\mathbb{R}$. We suppose that they are bounded: $\|X_1\|_{L_\infty} < \infty$. We define the functions $(g_{n,1})$ and $(g_{n,2})$ as satisfying:*

$$g_{n,1} : x_{1:n} \to \left( \frac{1}{\sqrt{n}} \sum_{i \leq \lfloor n/2 \rfloor} x_i - x_{i+\lfloor n/2 \rfloor} \right)^2, \qquad g_{n,2} : x_{1:n} \to \sqrt{n} \Big[ \prod_{i=1}^n \left( 1 + \frac{x_i - \bar{x}^n}{n} \right) - 1 \Big].$$

*Then the functions $(g_{n,1}, g_{n,2})$ satisfy conditions $(H_0)$, $(H_1)$ and $(H_2)$. Hence the bootstrap is consistent, i.e.:*

$$d_{\mathcal{F}} \left( g_{n,1}(Z^n) - \mathbb{E}(g_{n,1}(Z^n)|X), g_{n,1}(Y_{1:n}) - \mathbb{E}(g_{n,1}(Y_{1:n})) \mid X \right) \to 0;$$
$$d_{\mathcal{F}} \left( g_{n,2}(Z^n) - \mathbb{E}(g_{n,2}(Z^n)|X), g_{n,2}(Y_{1:n}) - \mathbb{E}(g_{n,2}(Y_{1:n})) \mid X \right) \to 0.$$

However hypothesis $(H_2)$ can be easily violated by simple examples. We prove in the next subsection, under mild conditions, that *violation of $(H_2)$ implies that no re-sampling method can provide asymptotically consistent confidence intervals*. We present here a simple example of this phenomenon.

**Example 9** *Let $(X_i)$ be an i.i.d sequence of scalar-valued, bounded observations with mean $0$. Write $(Y_i)$ an independent copy of $(X_i)$. Define the following functions $g_n : x_{1:n} \to \left( \frac{1}{\sqrt{n}} \sum_{i \leq n} x_i \right)^2$. Then hypothesis $(H_2)$ does not hold and the centered distributions of $(g_n(Y_{1:n}))$ and $(g_n(\tilde{Y}_{1:n}))$ are not asymptotically identical*

$$d_{\mathcal{F}} \left( g_n(\tilde{Y}_{1:n}) - \mathbb{E}\left[ g_n(\tilde{Y}_{1:n}) \mid X \right], \, g_n(Y_{1:n}) - \mathbb{E}\left[ g_n(Y_{1:n}) \right] \mid X \right) \not\to 0.$$

## G   Impossibility for Unstable Estimators to Uniform Perturbations

In this section we prove that if the estimators are sensitive to small uniform perturbations then the bootstrap method is not consistent. Then we offer three solutions on how to use the bootstrap to build confidence intervals with a guaranteed minimum coverage.

**Non-consistency of the bootstrap if the estimators are unstable.**   Let $\mathcal{P}_n'$ be a class of probability distributions on $\mathbb{R}^{d_n}$. Write $\mathcal{P}_M(\mathbb{R})$ the set of probability measures on $\mathbb{R}$. We say that the centered distribution of $g_n(\cdot)$ can be estimated over the class of distributions $\mathcal{P}_n'$ if there is a measurable function $\mathcal{Q}_n : x_1, \ldots, x_n \to \mathcal{P}_M(\mathbb{R})$ such that for all sequences of distributions $(\nu_n) \in \prod_{i=1}^\infty \mathcal{P}_n'$ we have

$$\mathbb{E}_{X^n \sim \nu_n} \left[ |d_{\mathcal{F}} \left( \mathcal{Q}_n(X^n), \, g_n(Y^n) - \mathbb{E}\left[ g_n(Y^n) \right] \mid X^n \right)| \right] \xrightarrow{n \to \infty} 0;$$

where $(Y_i^n)$ is taken to be to be an independent copy of $(X_i^n)$. We prove that the centered distribution of $g_n(\cdot)$ cannot be estimated if a hypothesis similar to $(H_2)$ is not respected (proof in Appendix N.4).

**Theorem 13** *Let $(g_n : \times_{l=1}^n \mathbb{R}^{d_n} \to \mathbb{R})$ be a sequence of measurable functions. Define $\Omega_n \subset \mathbb{R}^{d_n}$ to be a non empty open subset of $\mathbb{R}^{d_n}$ and let $\mathcal{P}_n' := \{p_\theta^n, \theta \in \Omega_n\}$ be a parametric subset of $\mathcal{P}_n$ such that $\mathbb{E}_{X \sim p_\theta^n}(X) = \theta$. Denote $(\mathcal{I}_n(\theta))$ the Fisher information matrix of $(p_\theta^n)$. Suppose that there is a sequence of measures $(p_{\theta_n}^n) \in \prod_{n=1}^\infty \mathcal{P}_n'$, a sequence $(z_n) \in \prod_{n=1}^\infty \mathbb{R}^{d_n}$ and a real $\epsilon > 0$ such that*

(i). $\limsup \sup_{\tilde{\theta}_n \in \left[\theta_n, \, \theta_n + \frac{z_n}{\sqrt{n}}\right]} \left\| \mathcal{I}_n(\tilde{\theta}_n)^{1/2} \frac{z_n}{\sqrt{n}} \right\|_2 < \infty.$

(ii). *The following holds if* $(X_i^n) \overset{i.i.d}{\sim} p_{\theta_n}^n$

$$\liminf_{n \to \infty} d_{\mathcal{F}} \left( g_n \left( X^n + \frac{z_n}{\sqrt{n}} \right) - \mathbb{E} \left[ g_n \left( X^n + \frac{z_n}{\sqrt{n}} \right) \right], \, g_n(X^n) - \mathbb{E}\left[ g_n(X^n) \right] \right) > \epsilon.$$

(iii). $\theta_n + \frac{z_n}{\sqrt{n}} \in \Omega_n$

*Then for all measurable functions* $\mathcal{Q}_n : x_1, \ldots, x_n \to \mathcal{P}_M(\mathbb{R})$ *there is a sequence* $(\nu_n) \in \prod_{n=1}^{\infty} \mathcal{P}'_n$ *such that:*

$$\left\| d_{\mathcal{F}} \left( \mathcal{Q}_n(X^n), g_n(Y^n) - \mathbb{E}\left[ g_n(Y^n) \right] \mid X^n \right) \right\|_{L_1} \not\to 0$$

*where* $(X_i^n), (Y_i^n) \overset{i.i.d}{\sim} \nu_n.$

Theorem 13 implies that if the means of the observations are unknown then no re-sampling method will in general be consistent. We propose in Appendix E, **??** and Appendix I three alternative ways to build confidence intervals, that bypass this impossibility result and have asymptotically a guaranteed coverage of at least $1 - \alpha$. Albeit, some of these intervals will potentially have larger size than needed.

## H   Corrected confidence intervals

We start by presenting an illustrative example of Theorem 2.

**Example 10** *Let* $(X_i)$ *be an i.i.d sequence with mean* 0 *and variance* 1. *Suppose that* $X_i \in L_{12}$ *and let* $c_\alpha(X^n)$ *be such that:*

$$P \left( \left| \left[ \frac{1}{\sqrt{n}} \sum_{i \leq n} Z_i^n \right]^2 - \mathbb{E}\left( \left[ \frac{1}{\sqrt{n}} \sum_{i \leq n} Z_i^n \mid X \right]^2 \right) \right| \geq c_\alpha(X) \mid X \right) \leq \alpha.$$

*Denote* $z_\alpha$ *the* $1 - \alpha$ *quantile of a standard normal:* $P(Z \geq z_\alpha) \leq \alpha$ *where* $Z \geq N(0, 1)$. *Then the following holds:*

$$\limsup_{n \to 0} P \left( \left| \left[ \frac{1}{\sqrt{n}} \sum_{i \leq n} Z_i^n \right]^2 - \mathbb{E}\left( \left[ \frac{1}{\sqrt{n}} \sum_{i \leq n} X_i \right]^2 \right) \right| \geq c_{\alpha/2}(X) + z_{\alpha/4}^2 \right) \leq \alpha$$

The second method exploits the bootstrap method for slightly shifted observations. The goal is to use the fact that under moderate conditions we know that $\left\| \bar{X}^n - \mathbb{E}(X_1^n) \right\|$ is of size $O(1/\sqrt{n})$. In this goal, we denote $B_{d_n}(\gamma)$ the ball in $\mathbb{R}^{d_n}$ of radius $\gamma$ for the Euclidean-norm.

**Theorem 14** *Let* $(g_n)$ *be a sequence of measurable functions. Suppose that for all sequence* $(\mu_n) \in B_{d_n}(\gamma_n)$ *Assumption 1 is satisfied by* $(X^n + \mu_n)$ *and* $(g_n)$. *Define* $(\gamma_n)$ *to be a sequence such that* $\frac{\sqrt{n}\gamma_n}{\log(d_n)} \to \infty$. *Set* $t_*^\alpha(X^n)$ *to be satisfying*

$$\sup_{\mu \in B_{d_n}(\gamma_n)} P \left( |g_n(Z^n + \mu) - \mathbb{E}(g_n(Z^n + \mu) \mid X^n)| \geq t_*^\alpha(X^n) \mid X^n \right) \leq \alpha$$

*Then the following holds:*

$$\limsup_{\delta \downarrow 0} \limsup_{n \to 0} P \left( |g_n(Y^n) - \mathbb{E}(g_n(Y^n))| \geq t_*^\alpha(X^n) + \delta \mid X^n \right) \leq \alpha.$$

See Appendix N.9 for a proof. We apply this new result to the previous illustrative example.

**Example 11** *Let* $(X_i)$ *be an i.i.d sequence with mean* $0$ *and variance* $1$. *Suppose that* $X_i \in L_{12}$ *and let* $c_\alpha(X^n)$ *be such that:*

$$\sup_{x,|x|\leq \log(n)/\sqrt{n}} P\left(\left|\left[\frac{1}{\sqrt{n}}\sum_{i\leq n} Z_i + x\right]^2 - \mathbb{E}\left(\left[\frac{1}{\sqrt{n}}\sum_{i\leq n} Z_i + x\right]^2\right)\right| \geq c_\alpha(X^n)\Big|X^n\right) \leq \alpha.$$

*Then the following holds:*

$$\limsup_{n\to 0} P\left(\left|\left[\frac{1}{\sqrt{n}}\sum_{i\leq n} Y_i\right]^2 - \mathbb{E}\left(\left[\frac{1}{\sqrt{n}}\sum_{i\leq n} Y_i\right]^2\right)\right| \geq c_\alpha(X^n)\right) \leq \alpha$$

# I   Robust Confidence Interval

Theorem 11 states that if the mean $\mathbb{E}[X_1^n]$ of the observations is known then we can instead study the centered bootstrap estimator, which under technical conditions, is asymptotically consistent. However assuming that the mean is known can be unrealistic. In this section, we instead assume that we know that it belongs to a certain subset $A_n$ and seek to find a confidence interval with a guaranteed coverage level for all potential values of the mean. To make this more precise, we consider an adversary that can see the draw of the random samples and translate them by any offset in the translation set $B_n := \{x - \mathbb{E}[X_1^n] : x \in A_n\}$. Our goal is to guarantee that no-matter what perturbation the adversary chooses, we produce a confidence interval with guaranteed coverage. Let $\mathcal{P}_n$ a set of probability distributions on $\mathbb{R}^{d_n}$ such that there exists a sequence of functions $(f_n)$ with $f_n \in \mathbb{C}^3$ such that:

1. The functions $(f_n)$ approximate the estimators $(g_n)$:

$$\sup_{\nu\in\mathcal{P}_n} \mathbb{E}_{X^n\sim\nu^{\otimes\infty}}\left(\left|f_n(\tilde{Z}^n) - g_n(\tilde{Z}^n)\right|\right) + \mathbb{E}_{X^n\sim\nu^{\otimes\infty}}\left(|f_n(X^n) - g_n(X^n)|\right) \xrightarrow{n\to\infty} 0.$$

$$(H_0^{rob})$$

2. The first, second and third order derivatives are such that:

$$\sup_{\nu\in\mathcal{P}_n} \max\left(R_{n,1}^{c,\nu}, R_{n,2}^{c,\nu}, R_{n,3}^{c,\nu}\right) \to 0. \qquad (H_1^{rob})$$

   where for each distribution $\nu \in \mathcal{P}_n$ we denoted by $R_{n,1}^{c,\nu}$, $R_{n,2}^{c,\nu}$ and $R_{n,3}^{c,\nu}$ the coefficients $R_{n,1}^c$, $R_{n,2}^c$ and $R_{n,3}^c$ computed for $(X_i^n) \overset{i.i.d}{\sim} \nu$.

Our goal we is to use the bootstrap method to find $(t_n^\alpha(X^n))$ such that the following holds:

$$\limsup_{n\to\infty} \sup_{\substack{\nu\in\mathcal{P}_n \\ \mathbb{E}_{X\sim\nu}(X)\in A_n}} P_{X^n,Y^n \overset{i.i.d}{\sim}\nu}\left(|g_n(Y^n) - \mathbb{E}[g_n(Y^n)]| \geq t_n^\alpha(X^n)\right) \leq \alpha. \qquad (2)$$

If conditions $(H_0^{\mathrm{rob}})$ and $(H_1^{\mathrm{rob}})$ hold then the bootstrap method can be used to find a sequence $(t_n^\alpha)$ such that (3) holds (proof in Appendix N.10).

**Theorem 15** *Let* $(g_n)$ *be a sequence of measurable functions, let* $(\mathcal{P}_n)$ *be sets of probability measures chosen such that* $(H_0^{\mathrm{rob}})$ *and* $(H_1^{\mathrm{rob}})$ *hold. For all* $\nu \in \mathcal{P}_n$ *and given a sample* $X^n \sim \nu$ *define* $t_n^\alpha(X^n)$ *to be such that:*

$$\sup_{\mu\in A_n} P\left(\left|g_n(Z^n + \mu - \bar{X}^n) - \mathbb{E}\left[g_n(Z^n + \mu - \bar{X}^n) \mid X^n\right]\right| \geq t_n^\alpha(X^n) \mid X^n\right) \leq \alpha. \qquad (3)$$

*Then if we write* $\mathcal{Q}_n := \{\nu \in \mathcal{P}_n \mid \mathbb{E}_{X\sim\nu}(X) \in A_n\}$ *then the following holds*

$$\liminf_{\delta\downarrow 0} \limsup_{n\to\infty} \sup_{\nu\in\mathcal{Q}_n} P_{X^n,Y^n\overset{i.i.d}{\sim}\nu}\left(|g_n(Y^n) - \mathbb{E}[g_n(Y^n)]| \geq t_n^\alpha(X^n) + \delta\right) \leq \alpha.$$

## J  Uniform Confidence Bands

In this section, we study the maximum of centered empirical processes. This is motivated by its application to uniform confidence bounds (see e.g. [18, 19]). Let $(X_i^n)$ be a triangular array of i.i.d process with $X_1^n$ taking value in $\mathbb{R}^{p_n}$ where $(p_n)$ is an increasing sequence. We want to estimate the distribution of

$$\max_{j \leq p_n} \frac{1}{\sqrt{n}} \sum_{i \leq n} X_{i,j}^n - \mathbb{E}\big[X_{1,j}^n\big].$$

For fast growing sequences of $(p_n)$ this statistics is not asymptotically Gaussian [23]. Therefore to study its distribution one might want to use the bootstrap method. Using our results we recover the results of [23] and establish conditions under which the bootstrap is asymptotically consistent (proof in Appendix P.1).

**Proposition 5** *Let $(p_n)$ be a sequence of integers satisfying $\log(p_n) = o(n^{1/4})$. Define $(X_i^n)$ to be a triangular array of sequences of i.i.d random variables taking value in $\mathbb{R}^{p_n}$. We suppose that $\left\| \sup_{k \leq p_n} \left| X_{1,k}^n \right| \right\|_{L_{12}} < \infty$. We denote $\mathcal{M}_n(x_{1:n}) = \max_{j \leq p_n} \frac{1}{\sqrt{n}} \sum_{i \leq n} x_{i,j}$. Then the following holds*

$$\left\| d_{\mathcal{F}}\left( \mathcal{M}_n(Z^n - \bar{X}^n),\ \mathcal{M}_n(Y^n - \mathbb{E}[Y_1^n]) \mid X^n \right) \right\|_{L_1} = o(1).$$

## K  Additional results for the stacked estimator

If $\beta_n$ grows proportionally to $\beta_n \propto \sqrt{n - m_n}$ then the bootstrap method is not a systematically consistent estimator of the risk of the smooth stacked estimator. We present a simple example demonstrating this.

**Example 12** *Let $(X_i)$ be a process of i.i.d random variables taking value in $\mathbb{R}$. Suppose that $X_1 \sim N(0,1)$. We choose $m_n = \lfloor n/2 \rfloor$ and $\beta_n = \sqrt{n}$ and define the estimators $(\hat{\theta}_n^1, \hat{\theta}_n^2)$ as constantly equal to*

$$\hat{\theta}_n^1(X_{1:n}) := 1 \quad \text{and} \quad \hat{\theta}_n^2(X_{1:n}) := -1.$$

*We shorthand by $\hat{\Theta}_n$ the corresponding stacked estimator. We choose the loss function $\mathcal{L}$ to be the square loss $\mathcal{L}(x, \theta) := (x - \theta)^2$. Let $(Z_1, Z_2) \sim N(0, \begin{bmatrix} 4 & 0 \\ 0 & 1 \end{bmatrix})$ be a Gaussian vector. Then the asymptotic centered distribution of the empirical loss is $Z_1 + Z_2 \tanh(4Z_2)$. However the asymptotic distribution of the bootstrap empirical loss is*

$$\mathcal{R}_{\hat{\Theta}_n}^{\mathrm{s}}(Z_{m_n+1:n}^n) - \mathbb{E}(\mathcal{R}_{\hat{\Theta}_n}^{\mathrm{s}}(Z_{m_n+1:n}^n) \big| \hat{\Theta}_n) \xrightarrow{d} Z_1 + Z_2 \tanh\left(4Z_2 + 4\sqrt{n}\bar{X}_{m_n+1:n}\right).$$

*Therefore the bootstrap method is not asymptotically consistent.*

For ease of notations, we denote

$$L_n^* := \sup_{\substack{i \leq 4 \\ l_{1:i} \leq d_n'}} \left\| \max_{p \leq p_n} \left| \partial_{2, l_{1:i}}^i \mathcal{L}_n\left( X_n^n, \hat{\theta}_n^p(X_{1:m_n}^n)(X_n^n) \right) \right| \right\|_{L_\infty}^{1/i} \vee 1.$$

We establish the limiting distribution of our bootstrap estimate under the following hypothesis:

$$(d_n' \max(L_n^*, 1))^2 \sqrt{\log(n) \max(T_n, 1)^3} = o(n^{1/6}) \qquad (H_1^{\mathrm{st\ bis}})$$

**Proposition 6** *Choose $(m_n)$, $(\beta_n)$ and $(p_n)$ be increasing sequences. Let $(X_i^n)$ be a triangular array of i.i.d observations taking value in $\mathbb{R}^{d_n}$. Set $(\mathcal{L}_n : \mathbb{R}^{d_n} \times \mathbb{R}^{d_n'} \to \mathbb{R})$ to be a sequence of smooth loss functions verifying condition $(H_1^{\mathrm{st\ bis}})$. Let $(Z_i^{n,1})$ and $(Z_i^{n,2})$ be independent bootstrap samples; and $(Y_i^{n,1})$ and $(Y_i^{n,2})$ be independent copies of $(X_{m_n+1:n}^n)$.*

*Then we have:*

$$\left\| d_{\mathcal{F}}\left( \mathcal{R}^{\mathrm{s}}_{\hat{\Theta}^{Z^{n,2}}_n}(Z^{n,1}_{1:n-m_n}) - \mathbb{E}\left( \mathcal{R}^{\mathrm{s}}_{\hat{\Theta}^{Z^{n,2}}_n}(Z^{n,1}_{1:n-m_n}) \mid \hat{\Theta}^{Z^{n,2}}_n, X^n \right), \right.$$

$$\left. \mathcal{R}^{\mathrm{s}}_{\hat{\Theta}'_n}(Y^{n,1}_{1:n-m_n}) - \mathbb{E}\left( \mathcal{R}^{\mathrm{s}}_{\hat{\Theta}'_n}(Y^{n,1}_{1:n-m_n}) \Big| \hat{\Theta}'_n \right) \mid X^n \right) \right\|_{L_1} \to 0;$$

*where we have set $\hat{\Theta}^{Z^{n,2}}_n := \hat{\Theta}_n(X^n_{1:m_n} Z^{n,2}_{1:n-m_n})$ and defined*

$$\hat{\Theta}'_n := \sum_{p \leq p_n} \hat{\theta}^p_n(X^n_{1:m_n}) \frac{e^{-\beta_n \left[ \mathcal{R}^p_n(X^n_{1:m_n} Y^{n,2}_{1:n-m_n}) + \tilde{\mathcal{R}}^p_n(X^n) \right]}}{\sum_{p' \leq p_n} e^{-\beta_n \left[ \mathcal{R}^{p'}_n(X^n_{1:m_n} Y^{n,2}_{1:n-m_n}) + \tilde{\mathcal{R}}^{p'}_n(X^n) \right]}};$$

*where we wrote $\tilde{\mathcal{R}}^p_n(X^n) := \mathcal{R}^p_n(X^n) - \sqrt{n - m_n}\, \mathbb{E}\big[ \mathcal{L}_n(X^n_n, \theta^n_p(X^n_{1:m_n})) \big| \theta^n_p(X^n_{1:m_n}) \big].$*

We notice that this implies that the bootstrap method is in general not consistent if $\beta_n \propto \sqrt{n - m_n}$, as it is illustrated in example 12.

However using Theorem 2 we can still use the bootstrap method to obtain confidence intervals that asymptotically have a minimum guaranteed coverage. To do so we need to estimate the (random) conditional variance-covariance matrix of $\left( \mathcal{L}_n(X^n_n, \hat{\theta}_p(X^n_{1:m_n})(X^n_n)) \big| X^n_{1:m_n} \right)_{p \leq p_n}$. In this goal for all $i \leq m_n$ we write $\theta^{\backslash i}_p := \hat{\theta}_p(X^n_1, \ldots, X^n_{i-1}, X^n_{i+1}, \ldots, X^n_{m_n})$ the leave-one-out estimator obtained by omitting the random variable $X^n_i$ from the training set. Moreover we write $S_{i,p} := \mathcal{L}(X^n_i, \hat{\theta}^{\backslash i}_p(X^n_i))$ and $S_i := (S_{i,p})$. We denote by $\hat{\Sigma}^2_n$ the empirical variance-covariance of the leave-one-out cross validated risks:

$$\hat{\Sigma}^2_n := \frac{1}{m_n - 1} \sum_{i \leq m_n} (S_i - \bar{S})(S_i - \bar{S})^T.$$

We make the following additional hypothesis: We assume that there is $b > 0$ such that

*Hypothesis* $(H^{\mathrm{st\ bis*}}_1)$

$$\inf_{\theta \in \mathbb{R}^{d'_n}} \|\mathcal{L}(X^n_n, \theta)\|_{L_3} \geq b, \qquad \frac{\log(p_n)^{7/6}}{n^{1/6}} = o(1)$$

$$\delta_{m_n} := \left\| \max_{p \leq p_n} \left| \mathcal{L}(X^n_n, \hat{\theta}^{\backslash 1}_p(X^n_n)) - \mathcal{L}(X^n_n, \hat{\theta}_p(X^n_{1:m_n})(X^n_n)) \right| \right\|_{L_2} = o(1)$$

$$\epsilon_{m_n} := \left\| \max_{p \leq p_n} \left| \mathcal{L}(X^n_n, \hat{\theta}^{\backslash 1}_p(X^n_n)) - \mathcal{L}(X^n_n, \hat{\theta}_p(X' X^n_{3:m_n})(X^n_n)) \right| \right\|_{L_2} = o((\sqrt{n} \log(p_n)^2)^{-1})$$

$$L'n := \left\| \max_{p \leq p_n} \left| \mathcal{L}(X^n_n, \hat{\theta}^{\backslash 1}_p(X^n_n)) \right| \right\|_{L_\infty} < \infty,$$

where $X'$ is an independent copy of $X^n_2$.

**Proposition 7** *Choose $(m_n)$, $(\beta_n)$ and $(p_n)$ be increasing sequences. Let $(X^n_i)$ be a triangular array of i.i.d observations taking value in $\mathbb{R}^{d_n}$. Set $(\mathcal{L}_n : \mathbb{R}^{d_n} \times \mathbb{R}^{d'_n} \to \mathbb{R})$ to be a sequence of smooth bounded loss functions satisfying conditions $(H^{\mathrm{st\ bis}}_1)$ and $(H^{\mathrm{st\ bis*}}_1)$. Suppose that the estimators $(\hat{\theta}_p)$ are symmetric in their coordinates: $\hat{\theta}_p(x_{1:m_n}) := \hat{\theta}_p(x_{\pi(1)}, \ldots, x_{\pi(m_n)})$ for all permutations $\pi$.*
*Let $(Z^{n,1}_i)$ be a bootstrap sample; and $(Y^{n,1}_i)$ and $(Y^{n,2}_i)$ be independent copies of $(X^n_{m_n+1:n})$. Let $N^n \sim N(0, \mathrm{Id}_{p_n})$. Let $\alpha > 0$ be a real; write $t^{\alpha/2}_{g,n}$, and $\hat{t}^{\alpha/2}_{b,n}(X^n)$ as quantities satisfying*

$$P\left( \left| \mathbb{E}\left( \mathcal{R}^{\mathrm{s}}_{\hat{\Theta}^{Z^{n,2}}_n}(Z^{n,1}_{1:n-m_n}) \mid X^n, Z^{n,1} \right) - \mathbb{E}\left( \mathcal{R}^{\mathrm{s}}_{\hat{\Theta}^{Z^{n,2}}_n}(Z^{n,1}_{1:n-m_n}) \mid X^n \right) \right| \geq t^{\alpha/2}_{b,n}(X^n) \mid X^n \right) \leq \alpha/2;$$

$$P\left( \sup_p \left| (\hat{\Sigma} N^n)_p \right| \geq \frac{\hat{t}^{\alpha/2}_{g,n}}{2d'_n T_n L^*_n} \Big| X^n \right) \leq \alpha/2;$$

*where we have set $\hat{\Theta}^{Z^{n,2}}_n := \hat{\Theta}_n(X^n_{1:m_n} Z^{n,2}_{1:n-m_n})$. Then for all $\delta > 0$ we have:*

$$\liminf_n P\left( \mathbb{E}\big[ \mathcal{R}^{\mathrm{s}}_{\hat{\Theta}^{Y^{n,2}}_n}(Y^{n,1}_{m_n+1:n}) \big| X^n \big] \in \mathbb{E}\left( \mathcal{R}^{\mathrm{s}}_{\hat{\Theta}^{Z^{n,2}}_n}(Z^{n,1}_{1:n-m_n}) \mid X^n, Z^{n,1} \right) \pm (\hat{t}^{\alpha/2}_{g,n} + t^{\alpha/2}_{b,n}(X^n) + \delta) \right) \geq 1 - \alpha;$$

*where we have set $\hat{\Theta}^{Y^{n,2}}_n := \hat{\Theta}_n(X^n_{1:m_n} Y^{n,2}_{1:n-m_n}).$*

# L  Proof of Theorem 1, Theorem 8, Theorem 15 and Theorem 10

As the proof of Theorem 1 and Theorem 8 are very similar, we present the proof for Theorem 1 and highlight the differences with the proof of theorem 8. The proof of Theorem 1 and Theorem 10 are identical.

Throughout the proofs we will use the following notations. We write $(X_i^c)$ and $(\tilde{Y}_i^c)$ the re-centered processes, around the empirical mean:

$$X_i^c := X_i^n - \bar{X}^n, \qquad \tilde{Y}_i^c := \tilde{Y}_i^n - \bar{X}^n, \qquad Y_i^c := Y_i^n - \bar{X}^n, \qquad Z_i^c := Z_i^n - \bar{X}^n.$$

## L.1  Main Lemmas

**Lemma 16 (Approximation Error)** *Let $(f_n)$ be a sequence of $\mathbb{C}^3$ functions that approximates $(g_n)$ as designated by $(H_0)$. Then:*

$$\left\| d_{\mathcal{F}} \left( g_n(Z^n), g_n(\tilde{Y}^n) \mid X^n \right) \right\|_{L_1} \leq \left\| d_{\mathcal{F}} \left( f_n(Z^n), f_n(\tilde{Y}^n) \mid X^n \right) \right\|_{L_1}$$
$$+ \left\| g_n(\tilde{Y}^n) - f_n(\tilde{Y}^n) \right\|_{L_1} + \left\| g_n(Z^n) - f_n(Z^n) \right\|_{L_1}$$

**Proof:**  Let $(f_n)$ be a sequence of $\mathbb{C}^3$ functions that approximate approximate $(g_n)$ as designated by Assumption 1. By Condition $(H_0)$ and the fact that for all $h \in \mathcal{F}$, $\sup_{x \in \mathbb{R}} |h'(x)| \leq 1$, we have:

$$\forall h \in \mathcal{F} : \left\| \mathbb{E} \left[ h \left( g_n(\tilde{Y}^n) \right) - h \left( f_n(\tilde{Y}^n) \right) \mid X^n \right] \right\|_{L_1} \leq \left\| \mathbb{E} \left[ \left| g_n(\tilde{Y}^n) - f_n(\tilde{Y}^n) \right| \mid X^n \right] \right\|_{L_1}$$
$$\leq \left\| g_n(\tilde{Y}^n) - f_n(\tilde{Y}^n) \right\|_{L_1}$$

$$\forall h \in \mathcal{F} : \left\| \mathbb{E} \left[ h \left( g_n(Z^n) \right) - h \left( f_n(Z^n) \right) \mid X^n \right] \right\|_{L_1} \leq \left\| \mathbb{E} \left[ g_n(Z^n) - f_n(Z^n) \mid X^n \right] \right\|_{L_1}$$
$$\leq \left\| g_n(Z^n) - f_n(Z^n) \right\|_{L_1}$$

Thus we can conclude that:

$$\left\| d_{\mathcal{F}} \left( g_n(Z^n), g_n(\tilde{Y}^n) \mid X^n \right) \right\|_{L_1} \leq \left\| d_{\mathcal{F}} \left( f_n(Z^n), f_n(\tilde{Y}^n) \mid X^n \right) \right\|_{L_1}$$
$$+ \left\| g_n(\tilde{Y}^n) - f_n(\tilde{Y}^n) \right\|_{L_1} + \left\| g_n(Z^n) - f_n(Z^n) \right\|_{L_1}$$

Therefore it is enough to study the metric distance between the distributions of $f_n(Z^n)$ and $f_n(\tilde{Y}^n)$. $\square$

**Lemma 17 (Lindenberg Path Decomposition)** *For any statistic $f_n$ and $i \in [n]$, let:*

$$A_i := \left\| \sup_{h \in \mathcal{F}} \mathbb{E} \left[ h \left( f_n \left( Z^{n,i} \right) \right) - h \left( f_n \left( Z^{n,i-1} \right) \right) \mid X^n \right] \right\|_{L_1}$$

*Then:*

$$\left\| d_{\mathcal{F}} \left( f_n(Z^n), f_n(\tilde{Y}^n) \mid X^n \right) \right\|_{L_1} \leq \sum_{i=1}^n A_i$$

**Proof:**  By the triangle inequality and writing the difference between $h(f_n(\tilde{Y}^n))$ and $h(f_n(Z^n))$ as a "Lindenberg" telescoping sum of interpolating differences, we have for all $h \in \mathcal{F}$:

$$\left\| d_{\mathcal{F}} \left( f_n(Z^n), f_n(\tilde{Y}^n) \mid X^n \right) \right\|_{L_1} = \left\| \sup_{h \in \mathcal{F}} \mathbb{E} \left[ h \left( f_n \left( \tilde{Y}^n \right) \right) - h \left( f_n \left( Z^n \right) \right) \mid X^n \right] \right\|_{L_1}$$
$$= \left\| \sup_{h \in \mathcal{F}} \mathbb{E} \left[ h \left( f_n \left( Z^{n,n} \right) \right) - h \left( f_n \left( Z^{n,0} \right) \right) \mid X^n \right] \right\|_{L_1}$$
$$= \left\| \sup_{h \in \mathcal{F}} \sum_{i=1}^n \mathbb{E} \left[ h \left( f_n \left( Z^{n,i} \right) \right) - h \left( f_n \left( Z^{n,i-1} \right) \right) \mid X^n \right] \right\|_{L_1}$$
$$\leq \sum_{i=1}^n \left\| \sup_{h \in \mathcal{F}} \mathbb{E} \left[ h \left( f_n \left( Z^{n,i} \right) \right) - h \left( f_n \left( Z^{n,i-1} \right) \right) \mid X^n \right] \right\|_{L_1}$$

$\square$

**Lemma 18 (Third-Order Approximation of Test Function)** *For any statistic $f_n$, let:*

$$\bar{f}_n^i := f_n\left(Z^{n,i,\bar{X}^n}\right) = f_n(\tilde{Y}_1^n, \ldots, \tilde{Y}_{i-1}^n, \bar{X}^n, Z_{i+1}^n, \ldots, Z_n^n)$$
$$\Delta_i(f_n, x) := f_n(Z^{n,i,x}) - \bar{f}_n^i.$$

*Then each quantity $A_i$ as defined in Theorem 17 can be bounded as:*

$$A_i \leq \mathbb{Q}_{1i} + \mathbb{Q}_{2i} + \mathbb{Q}_{3i} \tag{4}$$

*with:*

$$\mathbb{Q}_{1i} := \left\| \mathbb{E}\left[\Delta_i(f_n, \tilde{Y}_i^n) \mid Z^{n,i,\bar{X}^n}, X^n\right] - \frac{1}{n}\sum_{\ell=1}^n \Delta_i(f_n, X_\ell^n) \right\|_{L_1}$$

$$\mathbb{Q}_{2i} := \frac{1}{2}\left\| \mathbb{E}\left[\Delta_i(f_n, \tilde{Y}_i^n)^2 \mid Z^{n,i,\bar{X}^n}, X^n\right] - \frac{1}{n}\sum_{\ell=1}^n \Delta_i(f_n, X_\ell^n)^2 \right\|_{L_1}$$

$$\mathbb{Q}_{3i} := \frac{1}{6}\left(\left\| \Delta_i(f_n, \tilde{Y}_i^n)\right\|_{L_3}^3 + \|\Delta_i(f_n, Z_i^n)\|_{L_3}^3\right)$$

**Proof:**   By centering around $h(\bar{f}_n^i)$ we can re-write $A_i$ as:

$$A_i = \left\| \sup_{h\in\mathcal{F}} \mathbb{E}\left[h\left(f_n\left(Z^{n,i}\right)\right) - h\left(\bar{f}_n^i\right) - h\left(f_n\left(Z^{n,i-1}\right)\right) + h\left(\bar{f}_n^i\right) \mid X^n\right] \right\|_{L_1}$$

Applying a third-order Taylor expansion of each difference around $\bar{f}_n^i$ and using the fact that $h \in \mathcal{F}$ has third order derivatives, uniformly bounded by 1:

$$A_i \leq \left\| \sup_{h\in\mathcal{F}} \mathbb{E}\left[h'\left(\bar{f}_n^i\right)\left(f_n\left(Z^{n,i}\right) - \bar{f}_n^i - \left(f_n\left(Z^{n,i-1}\right) - \bar{f}_n^i\right)\right) \mid X^n\right] \right\|_{L_1}$$
$$+ \frac{1}{2}\left\| \sup_{h\in\mathcal{F}} \mathbb{E}\left[h''\left(\bar{f}_n^i\right)\left(\left(f_n\left(Z^{n,i}\right) - \bar{f}_n^i\right)^2 - \left(f_n\left(Z^{n,i-1}\right) - \bar{f}_n^i\right)^2\right) \mid X^n\right] \right\|_{L_1}$$
$$+ \frac{1}{6}\left(\left\| f_n\left(Z^{n,i}\right) - \bar{f}_n^i\right\|_{L_3}^3 + \left\| f_n\left(Z^{n,i-1}\right) - \bar{f}_n^i\right\|_{L_3}^3\right) =: \mathbb{I}_1 + \mathbb{I}_2 + \mathbb{I}_3$$

**Bounding $\mathbb{I}_1$.**   We now upper bound the $\mathbb{I}_1$ term. Observe that:

$$\mathbb{I}_1 := \left\| \sup_{h\in\mathcal{F}} \left| \mathbb{E}\left[h'\left(\bar{f}_n^i\right)\left(f_n\left(Z^{n,i}\right) - \bar{f}_n^i - \left(f_n\left(Z^{n,i-1}\right) - \bar{f}_n^i\right)\right) \mid X^n\right] \right| \right\|_{L_1}$$
$$= \left\| \sup_{h\in\mathcal{F}} \left| \mathbb{E}\left[h'\left(\bar{f}_n^i\right)\left(\Delta_i(f_n, \tilde{Y}_i^n) - \Delta_i(f_n, Z_i^n)\right) \mid X^n\right] \right| \right\|_{L_1}$$

By a tower law of expectations and the fact that $|h'(\bar{f}_n^i)| \leq 1$, for all $h \in \mathcal{F}$:

$$\mathbb{I}_1 = \left\| \sup_{h\in\mathcal{F}} \mathbb{E}\left[\mathbb{E}\left[h'\left(\bar{f}_n^i\right)\left(\Delta_i(f_n, \tilde{Y}_i^n) - \Delta_i(f_n, Z_i^n)\right) \mid Z^{n,i,\bar{X}^n}, X^n\right] \mid X^n\right] \right\|_{L_1}$$
$$= \left\| \sup_{h\in\mathcal{F}} \mathbb{E}\left[h'\left(\bar{f}_n^i\right)\mathbb{E}\left[\Delta_i(f_n, \tilde{Y}_i^n) - \Delta_i(f_n, Z_i^n) \mid Z^{n,i,\bar{X}^n}, X^n\right] \mid X^n\right] \right\|_{L_1}$$
$$\leq \left\| \sup_{h\in\mathcal{F}} \mathbb{E}\left[\left|h'\left(\bar{f}_n^i\right)\right|\left|\mathbb{E}\left[\Delta_i(f_n, \tilde{Y}_i^n) - \Delta_i(f_n, Z_i^n) \mid Z^{n,i,\bar{X}^n}, X^n\right]\right| \mid X^n\right] \right\|_{L_1}$$
$$\leq \left\| \mathbb{E}\left[\left|\mathbb{E}\left[\Delta_i(f_n, \tilde{Y}_i^n) - \Delta_i(f_n, Z_i^n) \mid Z^{n,i,\bar{X}^n}, X^n\right]\right| \mid X^n\right] \right\|_{L_1}$$
$$= \left\| \mathbb{E}\left[\Delta_i(f_n, \tilde{Y}_i^n) - \Delta_i(f_n, Z_i^n) \mid Z^{n,i,\bar{X}^n}, X^n\right] \right\|_{L_1}$$

Moreover, observe that conditional on $X^n$ and $Z^{n,i,\bar{X}^n}$, the only thing that varies in the random variable $\Delta_i(f_n, Z_i^n)$ is $Z_i^n$. Moreover, $Z_i^n$ is distributed uniformly over $\{X_1^n, \ldots, X_n^n\}$, conditional on $Z^{n,i,\bar{X}^n}, X^n$ (since conditional on $X^n$, $Z_i^n$ is independent of $\tilde{Y}^n$):

$$\mathbb{E}\left[\Delta_i(f_n, Z_i^n) \mid Z^{n,i,\bar{X}^n}, X^n\right] = \frac{1}{n}\sum_{\ell=1}^{n}\Delta_i(f_n, X_\ell^n)$$

We can then conclude that:

$$\mathbb{I}_1 \leq \left\|\mathbb{E}\left[\Delta_i(f_n, \tilde{Y}_i^n) \mid Z^{n,i,\bar{X}^n}, X^n\right] - \frac{1}{n}\sum_{\ell=1}^{n}\Delta_i(f_n, X_\ell^n)\right\|_{L_1}$$

**Bounding $\mathbb{I}_2$.** Observe that:

$$\mathbb{I}_2 := \frac{1}{2}\left\|\sup_{h\in\mathcal{F}}\mathbb{E}\left[h''\left(\bar{f}_n^i\right)\left(\left(f_n\left(Z^{n,i}\right) - \bar{f}_n^i\right)^2 - \left(f_n\left(Z^{n,i-1}\right) - \bar{f}_n^i\right)^2\right) \mid X^n\right]\right\|_{L_1}$$

$$= \frac{1}{2}\left\|\sup_{h\in\mathcal{F}}\mathbb{E}\left[h''\left(\bar{f}_n^i\right)\left(\Delta_i(f_n, \tilde{Y}_i^n)^2 - \Delta_i(f_n, Z_i^n)^2\right) \mid X^n\right]\right\|_{L_1}$$

By a tower law of expectations and the fact that $|h''(\bar{f}_n^i)| \leq 1$, for all $h \in \mathcal{F}$:

$$\mathbb{I}_2 = \frac{1}{2}\left\|\sup_{h\in\mathcal{F}}\mathbb{E}\left[\mathbb{E}\left[h''\left(\bar{f}_n^i\right)\left(\Delta_i(f_n, \tilde{Y}_i^n)^2 - \Delta_i(f_n, Z_i^n)^2\right) \mid Z^{n,i,\bar{X}^n}, X^n\right] \mid X^n\right]\right\|_{L_1}$$

$$= \frac{1}{2}\left\|\sup_{h\in\mathcal{F}}\mathbb{E}\left[h''\left(\bar{f}_n^i\right)\mathbb{E}\left[\Delta_i(f_n, \tilde{Y}_i^n)^2 - \Delta_i(f_n, Z_i^n)^2 \mid Z^{n,i,\bar{X}^n}, X^n\right] \mid X^n\right]\right\|_{L_1}$$

$$\leq \frac{1}{2}\left\|\sup_{h\in\mathcal{F}}\mathbb{E}\left[\left|h''\left(\bar{f}_n^i\right)\right|\left|\mathbb{E}\left[\Delta_i(f_n, \tilde{Y}_i^n)^2 - \Delta_i(f_n, Z_i^n)^2 \mid Z^{n,i,\bar{X}^n}, X^n\right]\right| \mid X^n\right]\right\|_{L_1}$$

$$\leq \frac{1}{2}\left\|\mathbb{E}\left[\left|\mathbb{E}\left[\Delta_i(f_n, \tilde{Y}_i^n)^2 - \Delta_i(f_n, Z_i^n)^2 \mid Z^{n,i,\bar{X}^n}, X^n\right]\right| \mid X^n\right]\right\|_{L_1}$$

$$= \frac{1}{2}\left\|\mathbb{E}\left[\Delta_i(f_n, \tilde{Y}_i^n)^2 - \Delta_i(f_n, Z_i^n)^2 \mid Z^{n,i,\bar{X}^n}, X^n\right]\right\|_{L_1}$$

By the conditional independence reasoning we presented in the bound for $\mathbb{I}_1$, the latter can further be written as:

$$\mathbb{I}_2 \leq \frac{1}{2}\left\|\mathbb{E}\left[\Delta_i(f_n, \tilde{Y}_i^n)^2 \mid Z^{n,i,\bar{X}^n}, X^n\right] - \frac{1}{n}\sum_{\ell=1}^{n}\Delta_i(f_n, X_\ell^n)^2\right\|_{L_1}$$

**Bounding $\mathbb{I}_3$.** We simply observe that $\mathbb{I}_3$ can be re-written as:

$$\mathbb{I}_3 := \frac{1}{6}\left(\left\|f_n\left(Z^{n,i}\right) - \bar{f}_n^i\right\|_{L_3}^3 + \left\|f_n\left(Z^{n,i-1}\right) - \bar{f}_n^i\right\|_{L_3}^3\right)$$

$$= \frac{1}{6}\left(\left\|\Delta_i(f_n, \tilde{Y}_i^n)\right\|_{L_3}^3 + \left\|\Delta_i(f_n, Z_i^n)\right\|_{L_3}^3\right)$$

$\square$

**Lemma 19 (Third Order Approximation of Smooth Statistic)** *Consider any statistic $f_n \in \mathbb{C}^3$ and any random vector $V \in \mathbb{R}^d$. Consider the random variables:*

$$\bar{\mathcal{J}}_i := \partial_i f_n\left(Z^{n,i,\bar{X}^n}\right)$$

$$\bar{\mathcal{H}}_i := \partial_i^2 f_n\left(Z^{n,i,\bar{X}^n}\right)$$

*If $\|V_k\|_{L_{12}} \leq \|X_k^n\|_{L_{12}}$ and $(H_1)$ holds, then:*

$$\left\|\Delta_i(f_n, V) - \bar{\mathcal{J}}_i^\top V^c - \frac{1}{2}\left(V^c\right)^\top \bar{\mathcal{H}}_i V^c\right\|_{L_3} \leq \frac{R_{n,3}}{6n}$$

*If $\|\sup_{k\leq d} V_k\|_{L_{12}} \leq \|\sup_{k\leq d} X_k^n\|_{L_{12}}$ and $(H_1^*)$ holds, then:*

$$\left\|\Delta_i(f_n, V) - \bar{\mathcal{J}}_i^\top V^c - \frac{1}{2}\left(V^c\right)^\top \bar{\mathcal{H}}_i V^c\right\|_{L_3} \leq \frac{R_{n,3}^*}{6n}$$

**Proof:** Since $f_n$ is three-times differentiable, for any random vector $V \in \mathbb{R}^d$ with $\|V_k\|_{L_{12}} \leq \|X_k\|_{L_{12}}$, for all $k \in [d]$, if we let $V^c = V - \bar{X}^n$, then by a Taylor expansion and Theorem 4:

$$\left\| \Delta_i(f_n, V) - \bar{\mathcal{J}}_i^\top V^c - \frac{1}{2} (V^c)^\top \bar{\mathcal{H}}_i V^c \right\|_{L_3}$$

$$\leq \frac{1}{6} \left\| \sup_{x \in [\bar{X}^n, V]} \sum_{k_1, k_2, k_3 \leq d_n} \partial^3_{i, k_{1:3}} f_n(Z^{n,i,x}) V^c_{k_1} V^c_{k_2} V^c_{k_3} \right\|_{L_3} \leq \frac{1}{6} \sum_{k_1, k_2, k_3 \leq d_n} D^n_{k_{1:3}}(f_n) \leq \frac{R_{n,3}}{6n}.$$

where we used the fact that $\|V^c_k\|_{L_{12}} \leq \|V_k\|_{L_{12}} + \|\bar{X}^n_k\|_{L_{12}} \leq 2\|X^n_k\|_{L_{12}} = M^n_k$. The second part of the lemma follows along identical lines, but in the second-to-last inequality we instead bound by:

$$\frac{1}{6} \left\| \sup_{x \in [\bar{X}^n, V]} \sum_{k_1, k_2, k_3 \leq d_n} |\partial^3_{i, k_{1:3}} f_n(Z^{n,i,x})| \left( \sup_{k \leq d} V^c_k \right)^3 \right\|_{L_3}$$

By a Cauchy-Schwarz inequality the latter is upper bounded by:

$$\frac{1}{6} \left\| \sup_{x \in [\bar{X}^n, V]} \sum_{k_1, k_2, k_3 \leq d_n} |\partial^3_{i, k_{1:3}} f_n(Z^{n,i,x})| \right\|_{L_{12}} \left\| \sup_{k \leq d} V^c_k \right\|^4_{L_{12}}$$

Since $\| \sup_{k \leq d} V_k \|_{L_{12}} \leq \| \sup_{k \leq d} X^n_k \|_{L_{12}}$, we also have that: $\| \sup_{k \leq d} V^c_k \|_{L_{12}} \leq 2 \| \sup_{k \leq d} X^n_k \|_{L_{12}}$. By the definition of $R^*_{n,3}$, we get the result. $\square$

## L.2 Proof of Theorem 1

**Lemma 20 (Bounding $\mathbb{Q}_{3i}$ under $(H_1)$)** *For any statistic sequence $(f_n)$, with $f_n \in \mathbb{C}^3$ that satisfies $(H_1)$, we have for all $i \in [n]$:*

$$\max \left\{ \|\Delta_i(f_n, Z^n_i)\|^3_{L_3}, \|\Delta_i(f_n, \tilde{Y}^n_i)\|^3_{L_3} \right\} \leq \frac{9}{n} \left( R^3_{n,1} + \frac{1}{2\sqrt{n}} R^3_{n,2} + \frac{1}{6n^2} R^3_{n,3} \right)$$

*Therefore:*

$$\mathbb{Q}_{3i} \leq \frac{18}{n} \left( R^3_{n,1} + \frac{1}{2\sqrt{n}} R^3_{n,2} + \frac{1}{6n^2} R^3_{n,3} \right)$$

**Proof:** By Theorem 19 and the fact that for any $a, b \in \mathbb{R}$: $|a+b|^3 \leq 3 \left(|a|^3 + |b|^3\right)$, we have that:

$$\left\| \Delta_i(f_n, \tilde{Y}^n_i) \right\|^3_{L_3} \leq 3 \left\| \bar{\mathcal{J}}_i^\top \tilde{Y}^c_i + \frac{1}{2} \left( \tilde{Y}^c_i \right)^\top \bar{\mathcal{H}}_i \tilde{Y}^c_i \right\|^3_{L_3} + \frac{R^3_{n,3}}{72\,n^3}$$

$$\leq 9 \left\| \bar{\mathcal{J}}_i^\top \tilde{Y}^c_i \right\|^3_{L_3} + \frac{9}{8} \left\| \left( \tilde{Y}^c_i \right)^\top \bar{\mathcal{H}}_i \tilde{Y}^c_i \right\|^3_{L_3} + \frac{R^3_{n,3}}{72\,n^3}$$

Moreover, by Theorem 4 we have:

$$\mathbb{E} \left[ \left| \bar{\mathcal{J}}_i^\top \tilde{Y}^c_i \right|^3 \right] \leq \left( \sum_{k \leq d_n} \|\tilde{Y}^c_{i,k}\|_{L_6} \|\bar{\mathcal{J}}_{i,k}\|_{L_6} \right)^3 \leq \frac{(R_{n,1})^3}{n}$$

$$\mathbb{E} \left[ \left| \left( \tilde{Y}^c_i \right)^\top \bar{\mathcal{H}}_i \tilde{Y}^c_i \right|^3 \right] \leq \left( \sum_{k_1, k_2 \leq d_n} \|\tilde{Y}^c_{i,k_1}\|_{L_9} \|\tilde{Y}^c_{i,k_2}\|_{L_9} \|\bar{\mathcal{H}}_{i,k_1,k_2}\|_{L_9} \right)^3 \leq \frac{(R_{n,2})^3}{n^{3/2}}$$

Combining the above yields the bound on $\|\Delta_i(f_n, \tilde{Y}^n_i)\|^3_{L_3}$. The same bound on $\|\Delta_i(f_n, Z^n_i)\|^3_{L_3}$ can be obtained in an identical manner. The lemma then follows. $\square$

**Lemma 21 (Bounding $\mathbb{Q}_{1i}$ under $(H_1)$)** *For any statistic sequence $(f_n)$, with $f_n \in \mathbb{C}^3$, which satisfies $(H_1)$, we have for all $i \in [n]$:*

$$\mathbb{Q}_{1i} \leq \frac{R_{n,3}}{3\,n} + \frac{2\,R_{n,2}}{n}$$

**Proof:** Applying Theorem 19 for $V = X_\ell^n$ and $V = \tilde{Y}_i^n$ and replacing the terms $\Delta_i(f_n, X_\ell^n)$ and $\Delta_i(f_n, \tilde{Y}_i^n)$ in $\mathbb{Q}_{1i}$, with their corresponding second degree approximations, we have:

$$\mathbb{Q}_{1i} \leq \frac{R_{n,3}}{3n} + \left\| \bar{\mathcal{J}}_i^\top \left( \mathbb{E}\left[ \tilde{Y}_i^n \mid X^n \right] - \bar{X}^n \right) \right\|_{L_1}$$
$$+ \frac{1}{2} \left\| \frac{1}{n} \sum_{\ell=1}^n (X_\ell^c)^\top \bar{\mathcal{H}}_i X_\ell^c - \mathbb{E}\left[ \left( \tilde{Y}_i^c \right)^\top \bar{\mathcal{H}}_i \tilde{Y}_i^c \mid Z^{n,i,\bar{X}^n}, X^n \right] \right\|_{L_1}$$

*Importantly, observe that by the definition of $\tilde{Y}_i^n$, we have that $\mathbb{E}\left[ \tilde{Y}_i^n \mid X^n \right] = \bar{X}^n$. Thus the first order term in this expansion vanishes.* Hence:

$$\mathbb{Q}_{1i} \leq \frac{R_{n,3}}{3n} + \frac{1}{2} \left\| \frac{1}{n} \sum_{\ell=1}^n (X_\ell^c)^\top \bar{\mathcal{H}}_i X_\ell^c - \mathbb{E}\left[ \left( \tilde{Y}_i^c \right)^\top \bar{\mathcal{H}}_i \tilde{Y}_i^c \mid Z^{n,i,\bar{X}^n}, X^n \right] \right\|_{L_1}$$

We can further split the second term on the right hand side as:

$$\mathbb{Q}_{1i} \leq \frac{R_{n,3}}{3n} + \frac{1}{2} \left\| \frac{1}{n} \sum_{\ell=1}^n (X_\ell^c)^\top \bar{\mathcal{H}}_i X_\ell^c - \mathbb{E}\left[ (Y_i^c)^\top \bar{\mathcal{H}}_i Y_i^c \mid Z^{n,i,\bar{X}^n}, X^n \right] \right\|_{L_1}$$
$$+ \frac{1}{2} \left\| \mathbb{E}\left[ (Y_i^c)^\top \bar{\mathcal{H}}_i Y_i^c - \left( \tilde{Y}_i^c \right)^\top \bar{\mathcal{H}}_i \tilde{Y}_i^c \mid Z^{n,i,\bar{X}^n}, X^n \right] \right\|_{L_1} \quad (5)$$

Moreover by exploiting the independence of the observations $(X_i^n)$ we remark that

$$\mathbb{Q}_{1i}^{(a)} := \left\| \frac{1}{n} \sum_{\ell=1}^n (X_\ell^c)^\top \bar{\mathcal{H}}_i X_\ell^c - \mathbb{E}\left[ (Y_i^c)^\top \bar{\mathcal{H}}_i Y_i^c \mid Z^{n,i,\bar{X}^n}, X^n \right] \right\|_{L_1}$$
$$\leq \sum_{k_1,k_2 \leq d_n} \left\| \frac{1}{n} \sum_{\ell=1}^n X_{\ell,k_1}^c X_{\ell,k_2}^c - \mathbb{E}\left[ Y_{i,k_1}^c Y_{i,k_2}^c \mid X^n \right] \right\|_{L_2} \left\| \bar{\mathcal{H}}_{i,k_1,k_2} \right\|_{L_2}$$
$$\leq \sum_{k_1,k_2 \leq d_n} \sqrt{\mathrm{Var}\left[ \frac{1}{n} \sum_{\ell \leq n} X_{\ell,k_1} X_{\ell,k_2} \right]} \left\| \bar{\mathcal{H}}_{i,k_1,k_2} \right\|_{L_2}$$
$$+ \sum_{k_1,k_2 \leq d_n} \left\| \bar{X}_{k_2}^n \left( \bar{X}_{k_1}^n - \mathbb{E}\left[ \bar{X}_{k_1}^n \right] \right) + \bar{X}_{k_1}^n \left( \bar{X}_{k_2}^n - \mathbb{E}\left[ \bar{X}_{k_2}^n \right] \right) \right\|_{L_2} \left\| \bar{\mathcal{H}}_{i,k_1,k_2} \right\|_{L_2}$$
$$\leq \frac{3}{\sqrt{n}} \sum_{k_1,k_2 \leq d_n} M_{k_1}^n M_{k_2}^n \left\| \bar{\mathcal{H}}_{i,k_1,k_2} \right\|_{L_2}$$
$$\leq \frac{3}{\sqrt{n}} \sum_{k_1,k_2 \leq d_n} D_{k_{1:2}}(f_n) \leq \frac{3}{n} R_{n,2} \quad (6)$$

Moreover, since for any two vectors $a, b$ and symmetric matrix $M$, we have that: $a^\top M a - b^\top M b = (a - b)^\top M a + b^\top M (b - a)$ and since $Y_i^c - \tilde{Y}_i^c = \mathbb{E}[X_1^n] - \bar{X}^n$, we have:

$$\mathbb{Q}_{1i}^{(b)} := \left\| \mathbb{E}\left[ (Y_i^c)^\top \bar{\mathcal{H}}_i Y_i^c - \left( \tilde{Y}_i^c \right)^\top \bar{\mathcal{H}}_i \tilde{Y}_i^c \mid Z^{n,i,\bar{X}^n}, X^n \right] \right\|_{L_1}$$
$$= \left\| \mathbb{E}\left[ \left( \bar{X}^n - \mathbb{E}\left[ X_1^n \right] \right)^\top \bar{\mathcal{H}}_i Y_i^c + \left( \tilde{Y}_i^c \right)^\top \bar{\mathcal{H}}_i \left( \bar{X}^n - \mathbb{E}\left[ X_1^n \right] \right) \mid Z^{n,i,\bar{X}^n}, X^n \right] \right\|_{L_1}$$

Moreover, since $\mathbb{E}\left[ Y_i^c \mid Z^{n,i,\bar{X}^n}, X^n \right] = \mathbb{E}[X_1^n] - \bar{X}^n$ and $\mathbb{E}\left[ \tilde{Y}_i^c \mid Z^{n,i,\bar{X}^n}, X^n \right] = 0$, we have:

$$\mathbb{Q}_{1i}^{(b)} = \left\| \left( \bar{X}^n - \mathbb{E}\left[ X_1^n \right] \right)^\top \bar{\mathcal{H}}_i \left( \mathbb{E}\left[ X_1^n \right] - \bar{X}^n \right) \right\|_{L_1}$$
$$\leq \sum_{k_1,k_2 \leq d_n} \left\| \bar{X}_{k_1}^n - \mathbb{E}\left[ X_{1,k_1}^n \right] \right\|_4 \left\| \bar{X}_{k_2}^n - \mathbb{E}\left[ X_{1,k_2}^n \right] \right\|_4 \left\| \bar{\mathcal{H}}_{i,k_1,k_2} \right\|_{L_2}$$
$$\leq \frac{1}{n} \sum_{k_1,k_2 \leq d_n} M_{k_1}^n M_{k_2}^n \left\| \bar{\mathcal{H}}_{i,k_1,k_2} \right\|_{L_2} \leq \frac{1}{n^{3/2}} R_{n,2} \quad (7)$$

Combining (5), (6) and (7) we obtain the result. $\qquad\square$

**Lemma 22 (Bounding $\mathbb{Q}_{2i}$ under $(H_1)$)** *For any statistic sequence $(f_n)$ that satisfies Assumption 1, we have for all $i \in [n]$:*

$$\mathbb{Q}_{2i} \leq \frac{R_{n,3}}{3n^{4/3}} \left( 4R_{n,1} + \frac{2}{n^{1/6}} R_{n,2} + \frac{1}{n^{2/3}} R_{n,3} \right) + \frac{3}{n^{7/6}} \left( (R_{n,1})^2 + \frac{1}{n^{1/3}} (R_{n,2})^2 \right)$$

**Proof:** For shorthand notation, let:

$$U(x) = \Delta_i(f_n, x) \qquad\qquad V(x) = \bar{\mathcal{J}}_i^\top x + \frac{1}{2} x^\top \bar{\mathcal{H}}_i^\top x$$

We will use the fact that for any two random variables $U, V$:

$$\|U^2 - V^2\|_{L_1} = \|(U - V)(U + V)\|_{L_1} \leq \|U - V\|_{L_2} (\|U\|_{L_2} + \|V\|_{L_2}).$$

We instantiate the latter with $U = U(X_\ell^n)$ and $V = V(X_\ell^c)$. Then by Theorem 19, we then have that: $\|U - V\|_{L_2} \leq \frac{R_{n,3}}{6n}$. By Theorem 20, we have that $\|U\|_{L_2} \leq \frac{3}{n^{1/3}} \left( R_{n,1} + \frac{1}{2n^{1/6}} R_{n,2} + \frac{1}{2n^{2/3}} R_{n,3} \right)$. Moreover, by a sequence of triangle and Cauchy-Schwarz inequalities, we also have that: $\|V\|_{L_2} \leq \frac{R_{n,1}}{n^{1/3}} + \frac{R_{n,2}}{\sqrt{n}}$. We can thus measure the approximation error of a second degree Taylor approximation:

$$\|U(X_\ell^n)^2 - V(X_\ell^c)^2\|_{L_1} \leq \frac{R_{n,3}}{6n} \left( \frac{3}{n^{1/3}} \left( R_{n,1} + \frac{1}{2n^{1/6}} R_{n,2} + \frac{1}{2n^{2/3}} R_{n,3} \right) + \frac{R_{n,1}}{n^{1/3}} + \frac{R_{n,2}}{\sqrt{n}} \right)$$

$$\leq \frac{R_{n,3}}{6n^{4/3}} \left( 4R_{n,1} + \frac{2}{n^{1/6}} R_{n,2} + \frac{1}{n^{2/3}} R_{n,3} \right) =: \epsilon_n$$

With identical steps the same bound holds for the analogous quantities $U(\tilde{Y}_i^n), V(\tilde{Y}_i^c)$. Therefore we have

$$\mathbb{Q}_{2i} \leq 2\epsilon_n + \frac{1}{2} \left\| \frac{1}{n} \sum_{\ell \leq n} V(X_\ell^c)^2 - \mathbb{E}\left[ V(\tilde{Y}_i^c)^2 \mid Z^{n,i,\bar{X}^n}, X^n \right] \right\|_{L_1}$$

Moreover, if we denote $\tilde{X}_\ell^n := X_\ell^n - \mathbb{E}[X_1^n]$ and $\zeta := \bar{X}^n - \mathbb{E}[X_1^n]$, then we have:

$$\left\| V(\tilde{X}_\ell^n)^2 - V(X_\ell^c)^2 \right\|_{L_1} \leq \left\| \left[ \bar{\mathcal{J}}_i^T \zeta + \frac{1}{2}(X_\ell^c)^T \bar{\mathcal{H}}_i \zeta + \frac{1}{2}\zeta^T \bar{\mathcal{H}}_i \tilde{X}_\ell^n \right\| \right\|_{L_2} \left( \|V(X_\ell^c)\|_{L_2} + [\|V(\tilde{X}_\ell^n)\|_{L_2} \right)$$

Observe that the first term in the product on the right-hand side is at most $\frac{1}{\sqrt{n}} \left( \frac{R_{1,n}}{n^{1/3}} + \frac{R_{2,n}}{\sqrt{n}} \right)$; by applying a series of Cauchy–Schwarz and triangle inequalities, and invoking concentration of the vector $\zeta$, i.e. $\|\zeta_k\|_{L_2} \leq \frac{M_k^n}{\sqrt{n}}$. Moreover, each of the summands in the second term is at most $\left( \frac{R_{1,n}}{n^{1/3}} + \frac{R_{2,n}}{\sqrt{n}} \right)$; by Cauchy–Schwarz and traingle inequality. Thus we get:

$$\left\| V(\tilde{X}_\ell^n)^2 - V(X_\ell^c)^2 \right\|_{L_1} \leq \frac{2}{n^{7/6}} \left( R_{1,n} + \frac{1}{n^{1/6}} R_{2,n} \right)^2$$

Thus it suffices to upper bound the term:

$$\left\| \frac{1}{n} \sum_{\ell \leq n} V(\tilde{X}_\ell^n)^2 - \mathbb{E}\left[ V(\tilde{Y}_i^c)^2 \mid Z^{n,i,\bar{X}^n}, X^n \right] \right\|_{L_1}$$

Moreover, note that $\tilde{X}_\ell^n \overset{d}{=} \tilde{Y}_\ell^c$. Noting that by the form of $V$, we can expand the latter as:

$$\sum_{k_{1:2} \leq d_n} \left\| \bar{\mathcal{J}}_{i,k_1} \bar{\mathcal{J}}_{i,k_2} \left( \frac{1}{n} \sum_{\ell \leq n} \tilde{X}_{\ell,k_1}^n \tilde{X}_{\ell,k_2}^n - \mathbb{E}\left[ \tilde{X}_{\ell,k_1}^n \tilde{X}_{\ell,k_2}^n \right] \right) \right\|_{L_1}$$

$$+ 2 \sum_{k_{1:3} \leq d_n} \left\| \bar{\mathcal{J}}_{i,k_1} \bar{\mathcal{H}}_{i,k_2,k_3} \left( \frac{1}{n} \sum_{\ell \leq n} \tilde{X}_{\ell,k_1}^n \tilde{X}_{\ell,k_2}^n \tilde{X}_{\ell,k_3}^n - \mathbb{E}\left[ \tilde{X}_{\ell,k_1}^n \tilde{X}_{\ell,k_2}^n \tilde{X}_{\ell,k_3}^n \right] \right) \right\|_{L_1}$$

$$+ \sum_{k_{1:4} \leq d_n} \left\| \bar{\mathcal{H}}_{i,k_1,k_2} \bar{\mathcal{H}}_{i,k_3,k_4} \left( \frac{1}{n} \sum_{\ell \leq n} \tilde{X}_{\ell,k_1}^n \tilde{X}_{\ell,k_2}^n \tilde{X}_{\ell,k_3}^n \tilde{X}_{\ell,k_4}^n - \mathbb{E}\left[ \tilde{X}_{\ell,k_1}^n \tilde{X}_{\ell,k_2}^n \tilde{X}_{\ell,k_3}^n \tilde{X}_{\ell,k_4}^n \right] \right) \right\|_{L_1}$$

By invoking Cauchy–Schwarz inequality and the concentration of each of the centered empirical averages, we have that the latter is bounded by:

$$\frac{1}{\sqrt{n}} \sum_{k_{1:2}\leq d_n} \left\|\bar{\mathcal{J}}_{i,k_1}\right\|_{L_4} \left\|\bar{\mathcal{J}}_{i,k_2}\right\|_{L_4} M^n_{k_1} M^n_{k_2} + \frac{2}{\sqrt{n}} \sum_{k_{1:3}\leq d_n} \left\|\bar{\mathcal{J}}_{i,k_1}\right\|_{L_4} \left\|\bar{\mathcal{H}}_{i,k_2,k_3}\right\|_{L_4} M^n_{k_1} M^n_{k_2} M^n_{k_3}$$

$$+ \frac{1}{\sqrt{n}} \sum_{k_{1:4}\leq d_n} \left\|\bar{\mathcal{H}}_{i,k_1,k_2}\right\|_{L_4} \left\|\bar{\mathcal{H}}_{i,k_3,k_4}\right\|_{L_4} M^n_{k_1} M^n_{k_2} M^n_{k_3} M^n_{k_4}$$

which in turn is upper bounded by:

$$\frac{1}{\sqrt{n}} \left( \sum_{k_1\leq d_n} \left\|\bar{\mathcal{J}}_{i,k_1}\right\|_{L_4} M^n_{k_1} + \sum_{k_{1:2}\leq d_n} \left\|\bar{\mathcal{H}}_{i,k_1,k_2}\right\|_{L_4} M^n_{k_1} M^n_{k_2} \right)^2 \leq \frac{1}{\sqrt{n}} \left( \frac{R_{n,1}}{n^{1/3}} + \frac{R_{n,2}}{\sqrt{n}} \right)^2$$

We can then conclude that:

$$\mathbb{Q}_{2i} \leq 2\epsilon_n + \frac{3}{n^{7/6}} \left( (R_{n,1})^2 + \frac{1}{n^{1/3}} (R_{n,2})^2 \right)$$

□

Therefore by combining Theorem 16, Theorem 17, Theorem 18, with the three lemmas in this section we obtain that for some sufficiently large universal constant $K$:

$$\left\| d_{\mathcal{F}} \left( g_n(Z^n), g_n(\tilde{Y}^n) \mid X^n \right) \right\|_{L_1} \leq \left\| g_n(\tilde{Y}^n) - f_n(\tilde{Y}^n) \right\|_{L_1} + \left\| g_n(Z^n) - f_n(Z^n) \right\|_{L_1}$$

$$+ K \left( R_{n,3} + R_{n,2} + (R_{n,1})^2 \max\left\{ \frac{1}{n^{1/6}}, R_{n,1} \right\} \right).$$

### L.3   Proof of Theorem 8

**Lemma 23 (Bounding $\mathbb{Q}_{3i}$ under $(H_1^*)$)** *For any statistic sequence $(f_n)$ that satisfies $(H_1^*)$, we have for all $i \in [n]$:*

$$\max\left\{ \|\Delta_i(f_n, Z^n_i)\|^3_{L_3}, \|\Delta_i(f_n, \tilde{Y}^n_i)\|^3_{L_3} \right\} \leq \frac{9}{n} \left( R^3_{n,1^*} + \frac{1}{2\sqrt{n}} R^3_{n,2^*} + \frac{1}{6n^2} R^3_{n,3^*} \right)$$

*Therefore:*

$$\mathbb{Q}_{3i} \leq \frac{18}{n} \left( (R^*_{n,1})^3 + \frac{1}{2\sqrt{n}} (R^*_{n,2})^3 + \frac{1}{6n^2} (R^*_{n,3})^3 \right)$$

**Proof:**   By Theorem 19 and the fact that for any $a, b \in \mathbb{R}$: $|a + b|^3 \leq 3 \left( |a|^3 + |b|^3 \right)$, we have that:

$$\left\| \Delta_i(f_n, \tilde{Y}^n_i) \right\|^3_{L_3} \leq 3 \left\| \bar{\mathcal{J}}_i^\top \tilde{Y}_i^c + \frac{1}{2} \left( \tilde{Y}_i^c \right)^\top \bar{\mathcal{H}}_i \tilde{Y}_i^c \right\|^3_{L_3} + \frac{(R^*_{n,3})^3}{72\, n^3}$$

$$\leq 9 \left\| \bar{\mathcal{J}}_i^\top \tilde{Y}_i^c \right\|^3_{L_3} + \frac{9}{8} \left\| \left( \tilde{Y}_i^c \right)^\top \bar{\mathcal{H}}_i \tilde{Y}_i^c \right\|^3_{L_3} + \frac{(R^*_{n,3})^3}{72\, n^3}$$

Moreover, by Cauchy-Schwarz inequality we have:

$$\mathbb{E}\left[ \left| \bar{\mathcal{J}}_i^\top \tilde{Y}_i^c \right|^3 \right] \leq \left\| \sup_{k\leq d_n} |\tilde{Y}^c_{i,k}|^3 \left( \sum_{k\leq d_n} |\bar{\mathcal{J}}_{i,k}| \right)^3 \right\|_{L_1} \leq \left\| \sup_{k\leq d_n} |\tilde{Y}^c_{i,k}| \right\|^3_{L_6} \left\| \sum_{k\leq d_n} |\bar{\mathcal{J}}_{i,k}| \right\|^3_{L_6} \leq \frac{(R^*_{n,1})^3}{n}$$

By similar applications of the Cauchy-Schwarz inequality we obtain that:

$$\mathbb{E}\left[ \left| \left( \tilde{Y}_i^c \right)^\top \bar{\mathcal{H}}_i \tilde{Y}_i^c \right|^3 \right] \leq \left( \left\| \sup_{k\leq d_n} \left| \tilde{Y}^c_{i,k} \right| \right\|^2_{L_9} \left\| \sum_{k_1,k_2\leq d_n} |\bar{\mathcal{H}}_{i,k_1,k_2}| \right\|_{L_9} \right)^3 \leq \frac{(R^*_{n,2})^3}{n^{3/2}}$$

Combining the above yields the bound on $\|\Delta_i(f_n, \tilde{Y}^n_i)\|^3_{L_3}$. The same bound on $\|\Delta_i(f_n, Z^n_i)\|^3_{L_3}$ can be obtained in an identical manner. The lemma then follows.

□

**Lemma 24 (Bounding $\mathbb{Q}_{1i}$ under $(H_1^*)$)** *For any statistic sequence $(f_n)$ that satisfies $(H_1^*)$, we have that there is a constant $C$ that does not depend on $n$ such that for all $i \in [n]$:*

$$\mathbb{Q}_{1i} \leq \frac{R_{n,3}^*}{3n} + \frac{C \max(\log(d_n), 1) R_{2,n}^*}{n}$$

**Proof:** Applying Theorem 19 for $V = X_\ell^n$ and $V = \tilde{Y}_i^n$ and replacing the terms $\Delta_i(f_n, X_\ell^n)$ and $\Delta_i(f_n, \tilde{Y}_i^n)$ in $\mathbb{Q}_{1i}$, with their corresponding second degree approximations, we have:

$$\mathbb{Q}_{1i} \leq \frac{R_{n,3}^*}{3n} + \left\| \bar{\mathcal{J}}_i^\top \left( \mathbb{E}\left[\tilde{Y}_i^n \mid X^n\right] - \bar{X}^n \right) \right\|_{L_1}$$
$$+ \frac{1}{2} \left\| \frac{1}{n} \sum_{\ell=1}^n (X_\ell^c)^\top \bar{\mathcal{H}}_i X_\ell^c - \mathbb{E}\left[ \left(\tilde{Y}_i^c\right)^\top \bar{\mathcal{H}}_i \tilde{Y}_i^c \mid Z^{n,i,\bar{X}^n}, X^n \right] \right\|_{L_1}$$

*Importantly, observe that by the definition of $\tilde{Y}_i^n$, we have that $\mathbb{E}\left[\tilde{Y}_i^n \mid X^n\right] = \bar{X}^n$. Thus the first order term in this expansion vanishes.* Hence:

$$\mathbb{Q}_{1i} \leq \frac{R_{n,3}^*}{3n} + \frac{1}{2} \left\| \frac{1}{n} \sum_{\ell=1}^n (X_\ell^c)^\top \bar{\mathcal{H}}_i X_\ell^c - \mathbb{E}\left[ \left(\tilde{Y}_i^c\right)^\top \bar{\mathcal{H}}_i \tilde{Y}_i^c \mid Z^{n,i,\bar{X}^n}, X^n \right] \right\|_{L_1}$$

We can further split the second term on the right hand side as:

$$\mathbb{Q}_{1i} \leq \frac{R_{n,3}^*}{3n} + \frac{1}{2} \left\| \frac{1}{n} \sum_{\ell=1}^n (X_\ell^c)^\top \bar{\mathcal{H}}_i X_\ell^c - \mathbb{E}\left[ (Y_i^c)^\top \bar{\mathcal{H}}_i Y_i^c \mid Z^{n,i,\bar{X}^n}, X^n \right] \right\|_{L_1}$$
$$+ \frac{1}{2} \left\| \mathbb{E}\left[ (Y_i^c)^\top \bar{\mathcal{H}}_i Y_i^c - \left(\tilde{Y}_i^c\right)^\top \bar{\mathcal{H}}_i \tilde{Y}_i^c \mid Z^{n,i,\bar{X}^n}, X^n \right] \right\|_{L_1} \quad (8)$$

Moreover by the triangular inequality we remark that

$$\mathbb{Q}_{1i}^{(a)} := \left\| \frac{1}{n} \sum_{\ell=1}^n (X_\ell^c)^\top \bar{\mathcal{H}}_i X_\ell^c - \mathbb{E}\left[ (Y_i^c)^\top \bar{\mathcal{H}}_i Y_i^c \mid Z^{n,i,\bar{X}^n}, X^n \right] \right\|_{L_1}$$

$$\leq \left\| \sup_{k_1,k_2 \leq d_n} \left| \frac{1}{n} \sum_{\ell=1}^n X_{\ell,k_1}^c X_{\ell,k_2}^c - \mathbb{E}\left[ Y_{i,k_1}^c Y_{i,k_2}^c \mid X^n \right] \right| \right\|_{L_2} \left\| \sum_{k_1,k_2 \leq d_n} |\bar{\mathcal{H}}_{i,k_1,k_2}| \right\|_{L_2}$$

$$\leq \left\| \sup_{k_1,k_2 \leq d_n} \left| \frac{1}{n} \sum_{\ell=1}^n X_{\ell,k_1}^n X_{\ell,k_2}^n - \mathbb{E}\left[ Y_{i,k_1}^n Y_{i,k_2}^n \mid X^n \right] \right| \right\|_{L_2} \left\| \sum_{k_1,k_2 \leq d_n} |\bar{\mathcal{H}}_{i,k_1,k_2}| \right\|_{L_2}$$

$$+ 2 \left\| \sup_{k_1,k_2 \leq d_n} |\bar{X}_{k_1}^n| \left| \frac{1}{n} \sum_{\ell=1}^n X_{\ell,k_2}^n - \mathbb{E}\left[ Y_{i,k_2}^n \mid X^n \right] \right| \right\|_{L_2} \left\| \sum_{k_1,k_2 \leq d_n} |\bar{\mathcal{H}}_{i,k_1,k_2}| \right\|_{L_2} \quad (9)$$

Using theorem 3 we know that there is a constant $C \in \mathbb{R}$ that does not depend on $n$ such that

$$\left\| \sup_{k_1,k_2 \leq d_n} \left| \frac{1}{n} \sum_{\ell=1}^n X_{\ell,k_1}^n X_{\ell,k_2}^n - \mathbb{E}\left[ Y_{i,k_1}^n Y_{i,k_2}^n \mid X^n \right] \right| \right\|_{L_2} \leq \frac{C \log(d_n)}{\sqrt{n}} \|\sup_{k_1} |X_{l,k_1}^n|\|_{L_4}^2;$$

and such that

$$\left\| \sup_{k_1,k_2 \leq d_n} \left| \frac{1}{n} \sum_{\ell=1}^n X_{\ell,k_2}^n - \mathbb{E}\left[ Y_{i,k_2}^n \mid X^n \right] \right| \right\|_{L_4} \leq \frac{C \log(d_n)}{\sqrt{n}} \|\sup_{k_1} |X_{l,k_1}^n|\|_{L_4}.$$

Therefore we can upper-bound (9) as:

$$\left\| \frac{1}{n} \sum_{\ell=1}^n (X_\ell^c)^\top \bar{\mathcal{H}}_i X_\ell^c - \mathbb{E}\left[ (Y_i^c)^\top \bar{\mathcal{H}}_i Y_i^c \mid Z^{n,i,\bar{X}^n}, X^n \right] \right\|_{L_1} \leq \frac{3 \max\left[1, C \log(d_n)\right] R_{2,n}^*}{n} \quad (10)$$

Moreover, since for any two vectors $a, b$ and symmetric matrix $M$, we have that: $a^\top M a - b^\top M b = (a-b)^\top M a + b^\top M(b-a)$ and since $Y_i^c - \tilde{Y}_i^c = \mathbb{E}[X_1^n] - \bar{X}^n$, we have:

$$\mathbb{Q}_{1i}^{(b)} := \left\| \mathbb{E}\left[ (Y_i^c)^\top \bar{\mathcal{H}}_i Y_i^c - \left(\tilde{Y}_i^c\right)^\top \bar{\mathcal{H}}_i \tilde{Y}_i^c \mid Z^{n,i,\bar{X}^n}, X^n \right] \right\|_{L_1}$$

$$= \left\| \mathbb{E}\left[ \left(\bar{X}^n - \mathbb{E}[X_1^n]\right)^\top \bar{\mathcal{H}}_i\, Y_i^c + \left(\tilde{Y}_i^c\right)^\top \bar{\mathcal{H}}_i \left(\bar{X}^n - \mathbb{E}[X_1^n]\right) \mid Z^{n,i,\bar{X}^n}, X^n \right] \right\|_{L_1}$$

As we established that $\left\| \sup_{k \leq d_n} |\bar{X}_k^n - \mathbb{E}\left[X_{1,k}^n\right]| \right\|_{L_4} \leq C \log(d_n) \left\| \sup_{k \leq d_n} |X_{i,k}| \right\|_{L_4}$ we have:

$$\mathbb{Q}_{1i}^{(b)} \leq \left\| \mathbb{E}\left[ \left(\bar{X}^n - \mathbb{E}[X_1^n]\right)^\top \bar{\mathcal{H}}_i\, Y_i^c \right] \right\|_{L_1} + \left\| \mathbb{E}\left[ \left(\tilde{Y}_i^c\right)^\top \bar{\mathcal{H}}_i \left(\bar{X}^n - \mathbb{E}[X_1^n]\right) \mid Z^{n,i,\bar{X}^n}, X^n \right] \right\|_{L_1}$$

$$\leq \left\| \sup_{k \leq d_n} |\bar{X}_k^n - \mathbb{E}[X_{1,k}^n]| \right\|_{L_3} \left\| \sup_{k \leq d_n} |Y_{i,k}^c| + |\tilde{Y}_{i,k}^c| \right\|_{L_3} \left\| \sum_{k_1, k_2 \leq d_n} |\bar{\mathcal{H}}_{i,k_1,k_2}| \right\|_{L_3}$$

$$\leq \frac{4C \log(d_n)}{n} \left\| \sup_{k \leq d_n} |X_{i,k}| \right\|_{L_4}^2 \left\| \sum_{k_1, k_2 \leq d_n} |\bar{\mathcal{H}}_{i,k_1,k_2}| \right\|_{L_3}$$

$$\leq \frac{2C \log(d_n) R_{2,n}^*}{n} \tag{11}$$

Combining (8), (10) and (11) we obtain the result. $\qquad\square$

**Lemma 25 (Bounding $\mathbb{Q}_{2i}$ under $(H_1^*)$)** *For any statistic sequence $(f_n)$ that satisfies $(H_1^*)$, we have that there is a constant $C$ that does not depend on $n$ such that for all $i \in [n]$:*

$$\mathbb{Q}_{2i} \leq \frac{R_{n,3}^*}{3n^{4/3}} \left( 4R_{n,1}^* + \frac{2}{n^{1/6}} R_{n,2}^* + \frac{1}{n^{2/3}} R_{n,3}^* \right) + \frac{C \max(\log(d_n), 1)}{n^{7/6}} \left( \left(R_{n,1}^*\right)^2 + \frac{1}{n^{1/3}} \left(R_{n,2}^*\right)^2 \right)$$

**Proof:** For shorthand notation, let:

$$U(x) = \Delta_i(f_n, x) \qquad\qquad V(x) = \bar{\mathcal{J}}_i^\top x + \frac{1}{2} x^\top \bar{\mathcal{H}}_i^\top x$$

We will use the fact that for any two random variables $U, V$:

$$\|U^2 - V^2\|_{L_1} = \|(U - V)(U + V)\|_{L_1} \leq \|U - V\|_{L_2} \left( \|U\|_{L_2} + \|V\|_{L_2} \right).$$

We instantiate the latter with $U = U(X_\ell^n)$ and $V = V(X_\ell^c)$. Then by Theorem 19, we then have that: $\|U - V\|_{L_2} \leq \frac{R_{n,3}^*}{6n}$. By Theorem 23, we have that $\|U\|_{L_2} \leq \frac{3}{n^{1/3}} \left( R_{n,1}^* + \frac{1}{2n^{1/6}} R_{n,2}^* + \frac{1}{2n^{2/3}} R_{n,3}^* \right)$. Moreover, by a sequence of triangle and Cauchy-Schwarz inequalities, we also have that: $\|V\|_{L_2} \leq \frac{R_{n,1}^*}{n^{1/3}} + \frac{R_{n,2}^*}{\sqrt{n}}$. We can thus measure the approximation error of a second degree Taylor approximation:

$$\|U(X_\ell^n)^2 - V(X_\ell^c)^2\|_{L_1} \leq \frac{R_{n,3}^*}{6n} \left( \frac{3}{n^{1/3}} \left( R_{n,1}^* + \frac{1}{2n^{1/6}} R_{n,2}^* + \frac{1}{2n^{2/3}} R_{n,3}^* \right) + \frac{R_{n,1}^*}{n^{1/3}} + \frac{R_{n,2}^*}{\sqrt{n}} \right)$$

$$\leq \frac{R_{n,3}^*}{6n^{4/3}} \left( 4R_{n,1}^* + \frac{2}{n^{1/6}} R_{n,2}^* + \frac{1}{n^{2/3}} R_{n,3}^* \right) =: \epsilon_n$$

With identical steps the same bound holds for the analogous quantities $U(\tilde{Y}_i^n), V(\tilde{Y}_i^c)$. Therefore we have

$$\mathbb{Q}_{2i} \leq 2\epsilon_n + \frac{1}{2} \left\| \frac{1}{n} \sum_{\ell \leq n} V(X_\ell^c)^2 - \mathbb{E}\left[ V(\tilde{Y}_i^c)^2 \mid Z^{n,i,\bar{X}^n}, X^n \right] \right\|_{L_1}$$

Moreover, if we denote $\tilde{X}_\ell^n := X_\ell^n - \mathbb{E}[X_1^n]$ and $\zeta := \bar{X}^n - \mathbb{E}[X_1^n]$, then we have:

$$\left\| V(\tilde{X}_\ell^n)^2 - V(X_\ell^c)^2 \right\|_{L_1}$$

$$\leq \left\| \left[ \bar{\mathcal{J}}_i^T \zeta + \frac{1}{2} (X_\ell^c)^T \bar{\mathcal{H}}_i\, \zeta + \frac{1}{2} \zeta^T \bar{\mathcal{H}}_i \tilde{X}_\ell^n \right] \right\|_{L_2} \left( \|V(X_\ell^c)\|_{L_2} + [\|V(\tilde{X}_\ell^n)\|_{L_2} \right) \tag{12}$$

Observe that by Theorem 3 we know that there is a constant $C$ that does not depend on $n$ such that: $\| \sup_k \zeta_k \|_{L_6} \leq C \log(d_n) \frac{\| \sup_k |X_{1,k}^n| \|_{L_6}}{\sqrt{n}}$. Therefore, by applying a series of Cauchy–Schwarz and triangle inequalities, the first term in the product on the right-hand side of (12) is at most $\frac{C \log(d_n)}{\sqrt{n}} \left( \frac{R_{1,n}^*}{n^{1/3}} + \frac{R_{2,n}^*}{\sqrt{n}} \right)$. Thus we get:

$$\left\| V(\tilde{X}_\ell^n)^2 - V(X_\ell^c)^2 \right\|_{L_1} \leq \frac{2C \log(d_n)}{n^{7/6}} \left( R_{1,n}^* + \frac{1}{n^{1/6}} R_{2,n}^* \right)^2$$

Thus it suffices to upper bound the term:

$$\left\| \frac{1}{n} \sum_{\ell \leq n} V(\tilde{X}_\ell^n)^2 - \mathbb{E}\left[ V(\tilde{Y}_i^c)^2 \mid Z^{n,i,\bar{X}^n}, X^n \right] \right\|_{L_1}$$

Moreover, note that $\tilde{X}_\ell^n \stackrel{d}{=} \tilde{Y}_\ell^c$. Noting that by the form of $V$, we can expand the latter as:

$$\left\| \sum_{k_{1:2} \leq d_n} \left| \bar{\mathcal{J}}_{i,k_1} \bar{\mathcal{J}}_{i,k_2} \left( \frac{1}{n} \sum_{\ell \leq n} \tilde{X}_{\ell,k_1}^n \tilde{X}_{\ell,k_2}^n - \mathbb{E}\left[ \tilde{X}_{\ell,k_1}^n \tilde{X}_{\ell,k_2}^n \right] \right) \right| \right\|_{L_1}$$

$$+ 2 \left\| \sum_{k_{1:3} \leq d_n} \left| \bar{\mathcal{J}}_{i,k_1} \bar{\mathcal{H}}_{i,k_2,k_3} \left( \frac{1}{n} \sum_{\ell \leq n} \tilde{X}_{\ell,k_1}^n \tilde{X}_{\ell,k_2}^n \tilde{X}_{\ell,k_3}^n - \mathbb{E}\left[ \tilde{X}_{\ell,k_1}^n \tilde{X}_{\ell,k_2}^n \tilde{X}_{\ell,k_3}^n \right] \right) \right| \right\|_{L_1}$$

$$+ \left\| \sum_{k_{1:4} \leq d_n} \left| \bar{\mathcal{H}}_{i,k_1,k_2} \bar{\mathcal{H}}_{i,k_2,k_3} \left( \frac{1}{n} \sum_{\ell \leq n} \tilde{X}_{\ell,k_1}^n \tilde{X}_{\ell,k_2}^n \tilde{X}_{\ell,k_3}^n \tilde{X}_{\ell,k_4}^n - \mathbb{E}\left[ \tilde{X}_{\ell,k_1}^n \tilde{X}_{\ell,k_2}^n \tilde{X}_{\ell,k_3}^n \tilde{X}_{\ell,k_4}^n \right] \right) \right| \right\|_{L_1}$$

By invoking theorem 3 and Cauchy–Schwarz we can find $C'$ such that:

$$\left\| \sup_{k_{1:2} \leq d_n} \left| \frac{1}{n} \sum_{\ell \leq n} \tilde{X}_{\ell,k_1}^n \tilde{X}_{\ell,k_2}^n - \mathbb{E}\left[ \tilde{X}_{\ell,k_1}^n \tilde{X}_{\ell,k_2}^n \right] \right| \right\|_{L_2} \leq \frac{C' \log(d_n) \left\| \sup_{k \leq d_n} \left| X_{l,k}^n \right| \right\|_{L_4}^2}{\sqrt{n}}$$

$$\left\| \sup_{k_{1:3} \leq d_n} \left| \frac{1}{n} \sum_{\ell \leq n} \tilde{X}_{\ell,k_1}^n \tilde{X}_{\ell,k_2}^n \tilde{X}_{\ell,k_3}^n - \mathbb{E}\left[ \tilde{X}_{\ell,k_1}^n \tilde{X}_{\ell,k_2}^n \tilde{X}_{\ell,k_3}^n \right] \right| \right\|_{L_2} \leq \frac{C' \log(d_n) \left\| \sup_{k \leq d_n} \left| X_{l,k}^n \right| \right\|_{L_6}^3}{\sqrt{n}}$$

$$+ \left\| \sup_{k_{1:4} \leq d_n} \left| \frac{1}{n} \sum_{\ell \leq n} \tilde{X}_{\ell,k_1}^n \tilde{X}_{\ell,k_2}^n \tilde{X}_{\ell,k_3}^n \tilde{X}_{\ell,k_4}^n - \mathbb{E}\left[ \tilde{X}_{\ell,k_1}^n \tilde{X}_{\ell,k_2}^n \tilde{X}_{\ell,k_3}^n \tilde{X}_{\ell,k_4}^n \right] \right| \right\|_{L_2} \leq \frac{C' \log(d_n) \left\| \sup_{k \leq d_n} \left| X_{l,k}^n \right| \right\|_{L_8}^4}{\sqrt{n}}.$$

Therefore by Cauchy–Schwarz inequality we have that

$$\left\| \frac{1}{n} \sum_{\ell \leq n} V(\tilde{X}_\ell^n)^2 - \mathbb{E}\left[ V(\tilde{Y}_i^n)^2 \mid Z^{n,i,\bar{X}^n}, X^n \right] \right\|_{L_1} \leq \frac{\max(C' \log(d_n), 1)}{\sqrt{n}} \left( \frac{R_{n,1}^*}{n^{1/3}} + \frac{R_{n,2}^*}{\sqrt{n}} \right)^2.$$

We can then conclude that:

$$\mathbb{Q}_{2i} \leq 2\epsilon_n + \frac{2 \max((C' + C) \log(d_n), 1)}{n^{7/6}} \left( \left( R_{n,1}^* \right)^2 + \frac{1}{n^{1/3}} \left( R_{n,2}^* \right)^2 \right)$$

$\square$

Therefore by combining Theorem 16, Theorem 17, Theorem 18, with the three lemmas in this section we obtain that for some sufficiently large universal constant $K$:

$$\left\| d_{\mathcal{F}}\left( g_n(Z^n), g_n(\tilde{Y}^n) \mid X^n \right) \right\|_{L_1} \leq \left\| g_n(\tilde{Y}^n) - f_n(\tilde{Y}^n) \right\|_{L_1} + \left\| g_n(Z^n) - f_n(Z^n) \right\|_{L_1}$$

$$+ K \left( \log(d_n) \left( (R_{n,1}^*)^2 \max\left\{ \frac{1}{n^{1/6}}, R_{n,1}^* \right\} + R_{n,2}^* \right) + R_{n,3}^* \right).$$

## L.4 Proof of corollary 1

**Proof:** Firstly, we remark that by using Theorem 5 we have:

$$\left\| d_\mathcal{F}\big(g_n(Z^n) - \mathbb{E}(g_n(Z^n) \mid X^n),\ g_n(\tilde{Y}^n) - \mathbb{E}(g_n(\tilde{Y}^n) \mid X^n) \mid X^n\big) \right\|_{L_1}$$

$$\leq \left\| d_\mathcal{F}\big(g_n(Z^n) - \mathbb{E}(g_n(Z^n) \mid X^n),\ g_n(Z^n) - \mathbb{E}(g_n(\tilde{Y}^n) \mid X^n) \mid X^n\big) \right\|_{L_1}$$

$$+ \left\| d_\mathcal{F}\big(g_n(Z^n) - \mathbb{E}(g_n(\tilde{Y}^n) \mid X^n),\ g_n(\tilde{Y}^n) - \mathbb{E}(g_n(\tilde{Y}^n) \mid X^n) \mid X^n\big) \right\|_{L_1}$$

$$\leq (A) + (B)$$

In the goal of bounding $(A)$, denote $h : x \to x$ the identity function. We easily note that $h$ belongs to the function class $\mathcal{F}$. Indeed it is three times differentiable with all its derivatives bounded by $1$. Therefore by theorem 1 we have

$$(A) \leq \mathbb{E}\left(\left| \mathbb{E}\left( g_n(Z^n) - g_n(\tilde{Y}^n) \mid X^n\right) \right|\right) \leq \left\| d_\mathcal{F}\big(g_n(Z^n), g_n(\tilde{Y}^n) \mid X^n\big) \right\|_{L_1}.$$

To upper-bound $(B)$ we note that Theorem 6 guarantees that:

$$\left\| d_\mathcal{F}\big(g_n(Z^n) - \mathbb{E}(g_n(Z^n) \mid X^n),\ g_n(\tilde{Y}^n) - \mathbb{E}(g_n(Z^n) \mid X^n) \mid X^n\big) \right\|_{L_1}$$

$$= \left\| d_\mathcal{F}\big(g_n(Z^n),\ g_n(\tilde{Y}^n) \mid X^n\big) \right\|_{L_1}$$

This implies that $(A) + (B) \leq 2 \left\| d_\mathcal{F}\big(g_n(Z^n),\ g_n(\tilde{Y}^n) \mid X^n\big) \right\|_{L_1}$ which proves the desired result.

$\square$

# M  Proofs from Section 2

## M.1  Proof of Proposition 1

**Proof:** We choose a measurable subset $A \subset \mathbb{R}$ and define the characteristic function

$$f : x \to \mathbb{I}(x \in A_{3\epsilon}).$$

Choose $\epsilon > 0$ and set $h_\epsilon : \mathbb{R} \to \mathbb{R}$ to be the following three-times differentiable function:

$$h_\epsilon(x) := \frac{1}{\epsilon^3} \int_{x-\epsilon}^{x} \int_{t-\epsilon}^{t} \int_{y-\epsilon}^{y} f(z) dz\, dy\, dt.$$

By simple observation we obtain that $\sup_{x \in \mathbb{R}} \max_{i \leq 3} \left| h_\epsilon^{(i)}(x) \right| \leq \frac{1}{\epsilon^3}$. Therefore we have for any two random variables $U, V$ and any event $\mathcal{E}$:

$$\mathbb{E}\left[ h_\epsilon(V) \mid \mathcal{E}\right] - \mathbb{E}\left[ h_\epsilon(U) \mid \mathcal{E}\right] \leq \frac{d_\mathcal{F}(U, V \mid \mathcal{E})}{\epsilon^3}.$$

Moreover, we remark that $h_\epsilon(x) \neq 0$ only if $x \in A_{6\epsilon}$ and that $h_\epsilon(x) = 1$ if $x \in A$. Thus for any random variable $Z$:

$$\mathbb{E}[h_\epsilon(Z) \mid \mathcal{E}] \in [\Pr(Z \in A \mid \mathcal{E}), \Pr(Z \in A_{6\epsilon} \mid \mathcal{E})]$$

which then implies that:

$$\mathbb{E}\left[ h_\epsilon(V) \mid \mathcal{E}\right] - \mathbb{E}\left[ h_\epsilon(U) \mid \mathcal{E}\right] \geq P(V \in A \mid \mathcal{E}) - P(U \in A_{6\epsilon} \mid \mathcal{E}).$$

Thus we have that for any two random variables $U, V$:

$$P(U \in A_{6\epsilon} \mid \mathcal{E}) \geq P(V \in A \mid \mathcal{E}) - \frac{d_\mathcal{F}(U, V \mid \mathcal{E})}{\epsilon^3} \tag{13}$$

Finally we observe that since for any $h \in \mathcal{F}$, we have $|h'(u)| \leq 1$ for all $u$, we have that $h(X - \mathbb{E}[X \mid \mathcal{E}]) \leq h(X) + \mathbb{E}[X \mid \mathcal{E}]$ and $h(Y - \mathbb{E}[Y]) \geq h(Y) - \mathbb{E}[Y \mid \mathcal{E}]$. Thus:

$$
\begin{aligned}
d_{\mathcal{F}}(X - \mathbb{E}[X \mid \mathcal{E}], Y - \mathbb{E}[Y \mid \mathcal{E}] \mid \mathcal{E}) = & \sup_{h \in \mathcal{F}} \mathbb{E}[h(X - \mathbb{E}[X \mid \mathcal{E}]) \mid \mathcal{E}] - \mathbb{E}[h(Y - \mathbb{E}[Y \mid \mathcal{E}]) \mid \mathcal{E}] \\
\leq & \sup_{h \in \mathcal{F}} \mathbb{E}[h(X) \mid \mathcal{E}] - \mathbb{E}[h(Y) \mid \mathcal{E}] + \mathbb{E}[X \mid \mathcal{E}] - \mathbb{E}[Y \mid \mathcal{E}] \Big| \\
\leq & \sup_{h \in \mathcal{F}} \mathbb{E}[h(X) \mid \mathcal{E}] - \mathbb{E}[h(Y) \mid \mathcal{E}] + \sup_{h \in \mathcal{F}} \mathbb{E}[h(X) \mid \mathcal{E}] - \mathbb{E}[h(Y) \mid \mathcal{E}] \\
\leq & \, 2 \, d_{\mathcal{F}}(X, Y \mid \mathcal{E})
\end{aligned}
$$

Thus applying Equation (13) to the centered random variables and invoking the fact that $\Pr(Y - \mathbb{E}[Y] \in A) \geq 1 - \alpha$, we get that:

$$
P(X - \mathbb{E}[X \mid \mathcal{E}] \in A_{6\epsilon} \mid \mathcal{E}) \geq P(Y - \mathbb{E}[Y \mid \mathcal{E}] \in A \mid \mathcal{E}) - \frac{2 d_{\mathcal{F}}(X, Y \mid \mathcal{E})}{\epsilon^3} \geq 1 - \alpha - \frac{2 d_{\mathcal{F}}(X, Y \mid \mathcal{E})}{\epsilon^3}
$$

which concludes the proof of the proposition.

$\square$

# N  Further Proofs from Section 3

## N.1  Proof of Example 2

**Proof:**  For simplicity, we write:

$$
J_i := \mathbb{I}(\min_{j \neq i} |X_j - X_i| > 1/n) \qquad g_n(X_{1:n}) := \frac{1}{\sqrt{n}} \sum_{i \leq n} J_i - \mathbb{E}(J_i)
$$

Moreover we note that for all $i \neq j$ we have $\tilde{Z}_i - \tilde{Z}_j = Z_i - Z_j$ this implies that

$$
g_n(\tilde{Z}_{1:n}) = g_n(Z_{1:n}).
$$

It is therefore enough to study $g_n(Z_{1:n})$. We remark that $g_n$ is stable in the perturbation of one of the observations, since changing the value of $X_1$ can change at most 3 of the random variables $(J_i)$. To make this rigorous, we denote the distance to closest neighbour of $x \in [0, 1]$, larger than $x$, as $d^+(x) = \min_{j \geq 2, X_j \geq x} |X_j - x|$ and to the closest neighbour, smaller than $x$, as $d^-(x) = \min_{j \geq 2, X_j \leq x} |X_j - x|$. By convention, if there is no $j \geq 2$ such that $X_j \geq x$ (respectively $X_j \leq x$) then we take $d^+(x)$ to be 0 (respectively $d^-(x) = 0$). We then have

$$
\begin{aligned}
\left\| g_n(X_{1:n}) - g_n(0 X_{2:n}) \right\|_{L_3} \leq & \frac{1}{\sqrt{n}} \left( \left\| J_1 - \mathbb{I}(\min_{j \geq 2} |X_j| > 1/n) \right\|_{L_3} + \left\| d^+(X_1) - d^+(0) \right\|_{L_3} + \left\| d^-(X_1) \right\|_{L_3} \right) \\
\overset{(a)}{\leq} & \frac{3}{\sqrt{n}}
\end{aligned}
$$

We show that $d_{\mathcal{F}}\Big(g_n(Y^n), g_n(Z^n) | X^n\Big)$ does not go to 0. We prove it by contradiction. Suppose that $\|d_{\mathcal{F}}\Big(g_n(Y^n), g_n(Z^n) | X^n\Big)\|_{L_1} \to 0$;, as the random variables $(g_n(Y^n))$ and $(g_n(Z^n))$ are uniformly integrable we have that $\mathbb{E}[g_n(Y^n) \mid X^n]$ converges to $\mathbb{E}[g_n(Z^n) \mid X^n]$. Moreover, we note that by the definition of $g_n$: $\mathbb{E}[g_n(Y^n) \mid X^n] = 0$. We remark that

$$
P\left( \min_{i \neq 1} |X_i - X_1| \geq 1/n \right) = \left( 1 - \frac{1}{n} \right)^{n-1} + O\left( \frac{1}{n} \right) = e^{-1} + o(1/\sqrt{n}).
$$

Moreover if we denote $C_{X_i}^n := \text{card}\left( j \neq i \text{ s.t } |X_i - X_j| \leq \frac{1}{n} \right)$ we have

$$
P\left( \min_{i \neq 1} |Z_i^n - Z_1^n| \geq 1/n \mid X^n \right) = \frac{1}{n} \sum_{i \leq n} e^{-1 - C_{X_i}^n} + o(1/\sqrt{n}).
$$

This notably implies that:

$$\mathbb{E}\left[g_n(Z^n) \mid X^n\right] = \sqrt{n}\left(\frac{1}{n}\sum_{i\leq n} e^{-1-C^n_{X_i}} - e^{-1}\right) + o(1)$$

$$= \sqrt{n}e^{-1}\left(\frac{1}{n}\sum_{i\leq n} e^{-C^n_{X_i}} - 1\right) + o(1)$$

We show that $\mathbb{E}\left[g_n(Z^n) \mid X^n\right]$ is asymptotically non-positive and takes asymptotically, strictly negative values with non-zero probability. This would then imply that $\mathbb{E}\left[g_n(Z^n) \mid X^n\right]$ is not asymptotically converging to $\mathbb{E}\left[g_n(Y^n)\right] = 0$, which contradicts the fact that $\|d_{\mathcal{F}}\left(g_n(Y^n), g_n(Z^n) \mid X^n\right)\|_{L_1} \to 0$. The first part follows since, $e^{-C^n_{X_i}} \leq 1$, and we therefore have that $\limsup_{n\to\infty} \mathbb{E}\left[g_n(Z^n) \mid X^n\right] \leq 0$. For the second part, we note that it is enough to lower bound the probability that $\frac{1}{n}\sum_{i\leq n} e^{-C^n_{X_i}} - 1$ is strictly negative. We observe that $\frac{1}{n}\sum_{i\leq n} e^{-C^n_{X_i}} - 1$ is bounded by 1 and that $\mathbb{E}\left[e^{-C^n_{X_1}}\right]$ is the moment generating function of a binomial distribution with $n-1$ trials and success probability at most $2/n$. This is at most $\left(1 - 2/n + 2\,e^{-1}/n\right)^{n-1} \leq e^{-(2-2/e)\,(n-1)/n} \leq e^{-1.2} + o(1)$. Thus for sufficiently large $n$, we have that with probability bounded away from zero: $\frac{1}{n}\sum_{i\leq n} e^{-C^n_{X_i}} - 1 < 0$, implying:

$$\liminf_{n\to\infty} P\left(\mathbb{E}\left[g_n(Z^n) \mid X^n\right] < 0\right) > 0;$$

Thus $\|d_{\mathcal{F}}\left(g_n(Y^n), g_n(Z^n) \mid X^n\right)\|_{L_1} \not\to 0$ and the bootstrap method is not consistent. $\qquad\square$

### N.2 Proof of Theorem 12

**Proof:** Let $(Y_i^n)$ be an independent copy of $(X_i^n)$. Since, by Theorem 5, $d_{\mathcal{F}}$ satisfies the triangular inequality:

$$\|d_{\mathcal{F}}\left(g_n(Z^n) - \mathbb{E}[g_n(Z^n)|X^n],\ g_n(Y^n) - \mathbb{E}[g_n(Y^n)] \mid X^n\right)\|_{L_1}$$

$$\leq \left\|d_{\mathcal{F}}\left(g_n(Z^n) - \mathbb{E}[g_n(Z^n)|X^n],\ g_n(\tilde{Y}^n) - \mathbb{E}[g_n(\tilde{Y}^n)|X^n],\ \mid X^n\right)\right\|_{L_1}$$

$$+ \left\|d_{\mathcal{F}}\left(g_n(\tilde{Y}^n) - \mathbb{E}[g_n(\tilde{Y}^n)|X^n],\ g_n(Y^n) - \mathbb{E}[g_n(Y^n)] \mid X^n\right)\right\|_{L_1} =: \mathbb{I}_1 + \mathbb{I}_2 \quad (14)$$

The first term $\mathbb{I}_1$ can be upper-bounded using Corollary 1. We therefore focus on bounding the second term $\mathbb{I}_2$ of Equation (14)

Let $(B_n)$ be a an increasing sequence such that (i) $B_n \to \infty$ and (ii) $r^{n,B_n} \to 0$. We remark that under Assumption 3 such a sequence always exists. For example set $B_1 = 1$ and $L_1 = 1$; then for all $n$ if $r^{n,2\,B_n} \leq 2^{-L_n}$ then set $L_{n+1} = L_n + 1$ and $B_{n+1} = 2\,B_n$ (by Assumption 3 the latter will occur at some finite $n$); otherwise keep $B_{n+1} = B_n$ and $L_{n+1} = L_n$.

We note that:

$$\mathbb{I}_2 \leq\ \left\|g_n(Y^n) - \mathbb{E}[g_n(Y^n)] - \left(g_n(\tilde{Y}^n) - \mathbb{E}[g_n(\tilde{Y}^n) \mid X^n]\right)\right\|_{L_1}$$

$$\overset{(a)}{\leq}\ \left\|\mathbb{I}(\sqrt{n}\|\bar{X}^n - \mathbb{E}\left[X^n_1\right]\|_2 \leq B_n)\right.$$

$$\times \sup_{x \in B_{d_n}(0, B_n)} \left|g_n\left(Y^n + \frac{x}{\sqrt{n}}\right) - \mathbb{E}\left[g_n\left(Y^n + \frac{x}{\sqrt{n}}\right) \mid X^n\right] - g_n(Y^n) + \mathbb{E}\left[g_n\left(Y^n\right)\right]\right|\right\|_{L_1}$$

$$+ \left\|\mathbb{I}(\sqrt{n}\|\bar{X}^n - \mathbb{E}\left[X^n_1\right]\|_2 \geq B_n) \times \left[g_n(Y^n) - \mathbb{E}[g_n(Y^n)] - \left(g_n(\tilde{Y}^n) - \mathbb{E}[g_n(\tilde{Y}^n) \mid X^n]\right)\right]\right\|_{L_1}$$

where (a) is a consequence of the triangle inequality. The first term is bounded by

$$\left\|\sup_{x \in B_{d_n}(0, B_n)} \left|g_n\left(Y^n + \frac{x}{\sqrt{n}}\right) - \mathbb{E}\left[g_n\left(Y^n + \frac{x}{\sqrt{n}}\right) \mid X^n\right] - g_n(Y^n) + \mathbb{E}\left[g_n\left(Y^n\right)\right]\right|\right\|_{L_1} \leq r^{n,B_n}.$$

The second term can be bounded by the use of the Cauchy-Swartz inequality:

$$\left\| \mathbb{I}(\sqrt{n} \| \bar{X}^n - \mathbb{E}\left[X_1^n\right] \|_2 \geq B_n) \times \left[ g_n(Y^n) - \mathbb{E}[g_n(Y^n)] - \left(g_n(\tilde{Y}^n) - \mathbb{E}[g_n(\tilde{Y}^n) \mid X^n]\right) \right] \right\|_{L_1}$$

$$\leq \left\| \mathbb{I}\left\{ \sqrt{n} \| \bar{X}^n - \mathbb{E}\left[X_1^n\right] \|_2 \geq B_n \right\} \right\|_{L_2} \left( \| g_n(Y^n) \|_{L_2} + \left\| g_n(\tilde{Y}^n) \right\|_{L_2} \right)$$

Observe that:

$$\mathbb{E}\left[ \left\| \bar{X}^n - \mathbb{E}\left[X_1^n\right] \right\|_2^2 \right] = \mathbb{E}\left[ \sum_{k \leq d_n} \left( \bar{X}_k^n - \mathbb{E}\left[X_{1,k}^n\right] \right)^2 \right]$$

$$= \sum_{k \leq d_n} \mathrm{Var}\left( \bar{X}_k^n \right) \leq \frac{4 \sum_{k \leq d_n} \|X_{1,k}^n\|_{L_2}^2}{n}$$

Hence, by Chebyshev's inequality:

$$P\left( \left\| \bar{X}^n - \mathbb{E}\left[X_1^n\right] \right\|_2 \geq B_n / \sqrt{n} \right) \leq \frac{n \mathrm{Var}\left( \| \bar{X}^n - \mathbb{E}\left[X_1^n\right] \|_2 \right)}{B_n^2} \leq \frac{4 \sum_{k \leq d_n} \|X_{1,k}^n\|_{L_2}^2}{B_n^2}$$

Thus we have:

$$\left\| \mathbb{I}\left\{ \sqrt{n} \left\| \bar{X}^n - \mathbb{E}\left[X_1^n\right] \right\|_2 \geq B_n \right\} \right\|_{L_2} \leq \sqrt{P\left( \left\| \bar{X}^n - \mathbb{E}\left[X_1^n\right] \right\|_2 \geq B_n / \sqrt{n} \right)} \leq \frac{2 \sqrt{\sum_{k \leq k_n} \|X_{1,k}^n\|_{L_2}^2}}{B_n}$$

Thus we conclude that:

$$\left\| d_{\mathcal{F}}\left( g_n(Y^n) - \mathbb{E}[g_n(Y^n)], g_n(\tilde{Y}^n) - \mathbb{E}[g_n(\tilde{Y}^n) \mid X^n] \mid X^n \right) \right\|_{L_1}$$

$$\leq \frac{2 \sqrt{\sum_{k \leq k_n} \|X_{1,k}^n\|_{L_2}^2}}{B_n} \left( \| g_n(Y^n) \|_{L_2} + \left\| g_n(\tilde{Y}^n) \right\|_{L_2} \right) + r^{n, B_n}.$$

$\square$

## N.3 Proof of example 8

**Proof:** We remark that both $g_{n,1}$ and $g_{n,2}$ are invariant under uniform perturbations. Indeed for all $x \in \mathbb{R}$ we have:

$$g_{n,1}(X_{1:n} + x/\sqrt{n}) = \left( \frac{1}{\sqrt{n}} \sum_{i \leq \lfloor n/2 \rfloor} X_i - \frac{x}{\sqrt{n}} - \left( x_{i + \lfloor n/2 \rfloor} - \frac{x}{\sqrt{n}} \right) \right)^2 = g_{n,1}(X_{1:n});$$

and

$$g_{n,2}(X_{1:n} + x/\sqrt{n}) = \sqrt{n} \left( \prod_{i=1}^n \left( 1 + \frac{x_i - x/\sqrt{n} - (\bar{x}^n - x/\sqrt{n})}{n} \right) - 1 \right) = g_{n,2}(X_{1:n}).$$

Therefore both functions $(g_{n,1})$ and $(g_{n,2})$ satisfy $(H_2)$. Moreover by a direct application of the chain rule we can verify that both $(g_{n,1})$ and $(g_{n,2})$ verify conditions $(H_0)$ and $(H_1)$. Hence Theorem 12 implies that:

$$d_{\mathcal{F}}\left( g_{n,1}(Z^n) - \mathbb{E}(g_{n,1}(Z^n)|X), g_{n,1}(Y_{1:n}) - \mathbb{E}(g_{n,1}(Y_{1:n})) \right) \to 0;$$

$$d_{\mathcal{F}}\left( g_{n,2}(Z^n) - \mathbb{E}(g_{n,2}(Z^n)|X), g_{n,2}(Y_{1:n}) - \mathbb{E}(g_{n,2}(Y_{1:n})) \right) \to 0.$$

$\square$

### N.4 Proof of Theorem 13

**Proof:** The proof works by contradiction. Suppose that there is a measurable function $\mathcal{Q}_n : x_1, \ldots, x_n \to \mathcal{P}_M(\mathbb{R})$ such that for all sequence of measures $(\nu_n) \in \prod_{n=1}^{\infty} \mathcal{P}_n$ the following holds:

$$\left\| d_{\mathcal{F}}\left( \mathcal{Q}_n(X^n), g_n(Y^n) - \mathbb{E}(g_n(Y^n)) \mid X^n \right) \right\|_{L_1} \xrightarrow{n \to \infty} 0.$$

By hypothesis we know that there is a sequence of measures $(p_{\theta_n}^n) \in \times_{l=1}^{\infty} \mathcal{P}_n$, an $\epsilon > 0$ and and a sequence of vectors $(z_n)$ such that

(i.) $\limsup \sup_{\tilde{\theta}_n \in [\theta_n, \theta_n + \frac{z_n}{\sqrt{n}}]} \| \mathcal{I}_n(\tilde{\theta}_n)^{1/2} \frac{z_n}{\sqrt{n}} \|_{v, d_n} < \infty;$

(ii.) The following holds for $(X^n) \overset{i.i.d}{\sim} p_{\theta_n}^n$

$$d_{\mathcal{F}}\left( g_n\left( X^n + \frac{z_n}{\sqrt{n}} \right) - \mathbb{E}\left[ g_n\left( X^n + \frac{z_n}{\sqrt{n}} \right) \right], g_n(X^n) - \mathbb{E}\left[ g_n(X^n) \right] \right) > \epsilon. \qquad (15)$$

(iii.) $\theta_n + \frac{z_n}{\sqrt{n}} \in \Omega_n$

By abuse of notations we denote $p_{\theta_n}^n + \frac{z_n}{\sqrt{n}}$ the distribution of $X + \frac{z_n}{\sqrt{n}}$ for $X \sim p_{\theta_n}^n$. We define the following subset of distributions $\mathcal{P}_n^* := \{ p_{\theta_n}^n, p_{\theta_n}^n + \frac{z_n}{\sqrt{n}} \}$ and let $X^n$ be i.i.d random variables distributed according to $\mu_n \in \mathcal{P}_n^*$. We want to test if $H_0 : \mathbb{E}(X_1^n) = \theta_n$ against the alternative hypothesis $H_1 : \mathbb{E}(X_1^n) = \theta_n + \frac{z_n}{\sqrt{n}}$. Using (15) we know that it is possible to find a rejection region $R_n$ such that:

$$P(X^n \in R_n | H_0) + P(X^n \notin R_n | H_1) \to 0.$$

However by hypothesis the Kullback-Leibler divergence is smooth and according to the Taylor expansion we know that

$$KL\left( p_{\theta_n}^n + \frac{z_n}{\sqrt{n}}, p_{\theta_n}^n \right) = \frac{1}{2n} z_n^\top \mathcal{I}_n(\tilde{\theta}_n) z_n,$$

where $\tilde{\theta}_n \in [\theta_n, \theta_n + \frac{z_n}{\sqrt{n}}]$. Moreover by definition of the total variation distance and the inequality: $\| p(\cdot) - q(\cdot) \|_{TV} \geq 1 - \frac{1}{2} e^{-KL(p,q)}$ (see Theorem 7) we know that for all rejection region $R_n$ we have

$$P(X^n \in R_n | H_0) + P(X^n \notin R_n | H_1)$$
$$\geq \frac{1}{2} e^{-nKL(p_{\theta_n}^n + \frac{z_n}{\sqrt{n}}, p_{\theta_n}^n)}$$
$$\geq \frac{1}{2} e^{-\frac{1}{2}\sqrt{z_n^\top \mathcal{I}_n(\tilde{\theta}_n) z_n}}.$$

Therefore using (i) we obtain that $\liminf_n P(X^n \in R_n | H_0) + P(X^n \notin R_n | H_1) > 0$ and note there is a contradiction. Hence we have successfully showed that the desired result holds. $\square$

### N.5 Proof of Theorem 11

**Proof:** This is a direct consequence of using theorem 10 for the random functions

$$\tilde{g}_n : x_{1:n} \to g_n\left( x_1 + \mathbb{E}(X_1^n) - \bar{X}^n, \ldots, x_n + \mathbb{E}(X_1^n) - \bar{X}^n \right);$$

which can be approximated by the following smooth functions:

$$\tilde{f}_n : x_{1:n} \to f_n\left( x_1 + \mathbb{E}(X_1^n) - \bar{X}^n, \ldots, x_n + \mathbb{E}(X_1^n) - \bar{X}^n \right).$$

$\square$

### N.6 Proof of Theorem 2

**Proof:** By the triangle inequality we know that:

$$|g_n(Z^n) - \mathbb{E}\left[g_n(Y^n) \mid X^n\right]| \leq \left|\mathbb{E}\left[g_n(Y^n) - g_n(\tilde{Y}^n) \mid X^n\right]\right| + \left|g_n(\tilde{Y}^n) - \mathbb{E}\left[g_n(\tilde{Y}^n) \mid X^n\right]\right|.$$

We bound successively each terms of the right-hand side. Firstly using theorem 1 we remark that for all $\delta > 0$ we have

$$P\left[\left|\mathbb{E}\left[g_n(\tilde{Y}^n) \mid X^n\right] - \mathbb{E}\left[g_n(Z^n) \mid X^n\right]\right| \geq \delta\right] \to 0.$$

Hence by combining this with the definition of $t_{b,n}^{\beta/2}$ we know that

$$\limsup_{\delta \downarrow 0} P\left(\left|g_n(Z^n) - \mathbb{E}\left[g_n(\tilde{Y}^n) \mid X^n\right]\right| \geq t_{b,n}^{\beta/2}(X^n) + \delta\right) \leq \beta/2.$$

Let $N^n \sim N(0, \Sigma_n)$ be a Gaussian vector with variance-covariance $\Sigma_n$. Using [19] we know that there is a constant $C$ that does not depend on $n$ such that the following holds:

$$\sup_t \left|P\left[\sqrt{n}\max_k|\bar{X}_k^n - \mathbb{E}(X_{1,k}^n)| \geq t\right] - P(\max_k|N^{n,k}| \geq t)\right| \leq \frac{C\log(d_n)^{7/6}\max(\|\max_j|X_{1,j}^n|\|_{L_4}^4, 1)}{n^{1/6}}$$

Combining this with hypothesis $(H_3)$ we obtain that:

$$P\left(\left|\mathbb{E}\left[g_n(Y^n + \bar{X}^n - \mathbb{E}(X_1^n)) - g_n(Y^n) \mid X^n\right]\right| \geq t_{g,n}^{\beta/2}\right)$$
$$\leq P\left(\|\sqrt{n}[\bar{X}^n - \mathbb{E}(X_1^n)]\|_\infty^\alpha \geq \frac{t_{g,n}^{\beta/2}}{C_n}\right)$$
$$\leq \beta/2 + o(1)$$

Therefore for all $\delta > 0$ we have

$$P\left(\left|g_n(Z^n) - \mathbb{E}(g_n(Y^n))\right| \leq t_{b,n}^{\beta/2}(X^n) + t_{g,n}^{\beta/2} + \delta\right)$$
$$\leq P\left(\left|g_n(Z^n) - \mathbb{E}(g_n(\tilde{Y}^n))\right| \leq t_{b,n}^{\beta/2}(X^n) + \delta\right)$$
$$+ P\left(\left|\mathbb{E}(g_n(\tilde{Y}^n) - g_n(Y^n) \mid X^n)\right| \leq t_{g,n}^{\beta/2}\right)$$
$$\leq \beta + o(1);$$

$\square$

### N.7 Proof of Theorem 9

**Proof:** By the triangle inequality we know that:

$$|g_n(Z^n) - \mathbb{E}\left[g_n(Y^n) \mid X^n\right]| \leq \left|\mathbb{E}\left[g_n(Y^n) - g_n(\tilde{Y}^n) \mid X^n\right]\right| + \left|g_n(\tilde{Y}^n) - \mathbb{E}\left[g_n(\tilde{Y}^n) \mid X^n\right]\right|.$$

We bound successively each terms of the right-hand side. Firstly using theorem 8 we remark that for all $\delta > 0$ we have

$$P\left[\left|\mathbb{E}\left[g_n(\tilde{Y}^n) \mid X^n\right] - \mathbb{E}\left[g_n(Z^n) \mid X^n\right]\right| \geq \delta\right] \to 0.$$

Hence by combining this with the definition of $t_{b,n}^{\beta/2}$ we know that

$$\limsup_{\delta \downarrow 0} P\left(\left|g_n(Z^n) - \mathbb{E}\left[g_n(\tilde{Y}^n) \mid X^n\right]\right| \geq t_{b,n}^{\beta/2}(X^n) + \delta\right) \leq \beta/2.$$

Let $N^n \sim N(0, \mathrm{Id})$ be a Gaussian vector of dimension $d_n$. Using [19] we know that there is a constant $C$ that does not depend on $n$ such that the following holds:

$$\sup_t \left| P\left[ \sqrt{n} \max_k \left| \bar{X}_k^n - \mathbb{E}(X_{1,k}^n) \right| \geq t \right] - P(\max_k |(\Sigma_n N^n)_k| \geq t) \right| \leq \frac{C \log(d_n)^{7/6} \max(\| \max_j |X_{1,j}^n| \|_{L_4}^4, 1)}{n^{1/6}}$$

Moreover by hypothesis $\hat{\Sigma}_n$ is an estimator of $\Sigma_n$ verifying

$$\sup_t \left| P(\max_k |(\hat{\Sigma}_n N^n)_k| \geq t) - P(\max_k |(\Sigma_n N^n)_k| \geq t) \right| \leq \to 0.$$

Combining this with hypothesis $(H_3)$ we obtain that:

$$P\left( \left| \mathbb{E}\left[ g_n(Y^n + \bar{X}^n - \mathbb{E}(X_1^n)) - g_n(Y^n) \mid X^n \right] \right| \geq \hat{t}_{g,n}^{\beta/2} \right)$$

$$\leq P\left( \| \sqrt{n}[\bar{X}^n - \mathbb{E}(X_1^n)] \|_\infty^\alpha \geq \frac{\hat{t}_{g,n}^{\beta/2}}{C_n} \right)$$

$$\leq \beta/2 + o(1)$$

Therefore for all $\delta > 0$ we have

$$P\left( \left| g_n(Z^n) - \mathbb{E}(g_n(Y^n)) \right| \leq t_{\mathrm{b},n}^{\beta/2}(X^n) + \hat{t}_{g,n}^{\beta/2} + \delta \right)$$

$$\leq P\left( \left| g_n(Z^n) - \mathbb{E}(g_n(\tilde{Y}^n)) \right| \leq t_{\mathrm{b},n}^{\beta/2}(X^n) + \delta \right)$$

$$+ P\left( \left| \mathbb{E}(g_n(\tilde{Y}^n) - g_n(Y^n) \mid X^n) \right| \leq \hat{t}_{g,n}^{\beta/2} \right)$$

$$\leq \beta + o(1);$$

$\square$

## N.8  Proof of Example 10

**Proof:** Define the function: $g_n : x_{1:n} \to \left( \frac{1}{\sqrt{n}} \sum_{i \leq n} x_i \right)^2$. It is straightforward to check that $(g_n)$ and $(X^n)$ satisfy conditions $(H_0)$ and $(H_1)$. Moreover for all $x$ we note that:

$$\left| \mathbb{E}\left( g_n(Y^n + \frac{x}{\sqrt{n}}) - g_n(Y^n) \right) \right| = x^2.$$

Therefore $(g_n)$ also satisfy conditions $(H_3)$ with $C_n = 1$ and $\alpha = 2$. The result is a direct consequence of Theorem 2. $\square$

## N.9  Proof of Theorem 14

**Proof:** Firstly we prove that with high-probability $\bar{X}^n - \mathbb{E}(X_1^n)$ is in $B_{d_n}(\gamma_n)$. Indeed, using Theorem 3 and Chebystchev inequality we know that there is a constant $C$ such that

$$P\left[ \max_k \left| \bar{X}_k^n - \mathbb{E}(X_{1,k}^n) \right| \geq \gamma_n \right] \leq \frac{\mathbb{E}\left( \max_k \left| \bar{X}_k^n - \mathbb{E}(X_{1,k}^n) \right|^2 \right)}{\gamma_n^2}$$

$$\leq \frac{C \log(d_n)^2 \left\| \max_{k \leq k_n} \left| X_{1,k}^n \right| \right\|_{L_2}^2}{\gamma_n^2 n} \to 0.$$

Therefore with high-probability $\tilde{\mu} := \bar{X}^n - \mathbb{E}(X_1^n)$ is in $B_{d_n}(\gamma)$. Moreover, we note that:

$$P\left( |g_n(Z^n - \tilde{\mu}) - \mathbb{E}(g_n(Z^n - \tilde{\mu}) \mid X^n)| \geq t_*^\alpha(X^n) \mid X^n \right) \leq \alpha.$$

We remark that $Z^n - \tilde{\mu} = \tilde{Z}^n$ therefore according to Theorem 11 for all $\delta > 0$ we have:

$$P\left( |g_n(Y^n) - \mathbb{E}(g_n(Y^n)| \geq t_*^\alpha(X^n) + \delta) \mid X^n \leq \alpha + o_n(1). \right.$$

Which implies that asymptotically the confidence intervals $\left[ t_*^\beta(X^n) - \delta, \ t_*^\alpha(X^n) + \delta \right]$ has an asymptotic coverage of at least $1 - \beta$. $\square$

## N.10 Proof of Theorem 15

**Proof:** Denote
$$\epsilon_n := \sup_{\nu \in \mathcal{P}_n} \mathbb{E}_{X_1^n \in \mathcal{P}_n} \left( \left| f_n(\tilde{Z}^n) - g_n(\tilde{Z}^n) \right| \right) + \mathbb{E} \left( |f_n(Y^n) - g_n(Y^n)| \right) + \sup_{\nu \in \mathcal{P}_n} \max \left( R_{n,1}^{c,\nu}, R_{n,2}^{c,\nu}, R_{n,3}^{c,\nu} \right).$$
Choose $\nu \in \mathcal{Q}_n$, and let $X^n \sim \nu$ if $t_n^\alpha(X^n)$ is chosen such that:
$$P \left( \left| g_n(Z^n + \mathbb{E}(X_1^n) - \bar{X}^n) - \mathbb{E} \left[ g_n(Z^n - \mathbb{E}(X_1^n) + \bar{X}^n) \mid X^n \right] \right| \geq t_n^\alpha(X^n) \mid X^n \right) \leq \alpha.$$
then according to Theorem 11 and Proposition 1 for all $\delta > 0$ we have
$$P \left( |g_n(X^n) - \mathbb{E}[g_n(X^n)]| \geq t_n^\alpha + \delta \right) \leq \alpha + O(\epsilon_n).$$
But as $\mathbb{E}(X_1^n) \in A_n$ by definition of $t_n^\alpha(X^n)$ we know that: $t_n^\alpha(X^n) \leq t_\alpha^{*,n}$. This directly implies that
$$P_{X^n, Y^n \overset{i.i.d}{\sim} \nu} \left( |g_n(Y^n) - \mathbb{E}[g_n(Y^n)]| \geq t_\alpha^{*,n}(X^n) + \delta \right) \leq \alpha + O(\epsilon_n).$$
As this holds for all $\nu \in A_n$ we get the desired result. $\square$

# O  Proofs from Appendix D

## O.1  Proof of example 3

**Proof:** We will use Theorem 1. We note that the condition $(H_0)$ holds automatically as the functions $(g_n)$ are three-times differentiable. Therefore to we only need to verify that $(H_1)$ holds. By the chain rule we have:
$$\partial_i g_n(Z^{n,i,\bar{X}^n}) = \frac{p}{\sqrt{n}} \left( \frac{1}{\sqrt{n}} \sum_{j \neq i}^n X_j + \frac{1}{\sqrt{n}} \bar{X}^n \right)^{p-1};$$
and
$$\partial_i^2 g_n(Z^{n,i,\bar{X}^n}) = \frac{p(p-1)\mathbb{I}(p \geq 2)}{n} \left( \frac{1}{\sqrt{n}} \sum_{i=1}^n X_i + \frac{1}{\sqrt{n}} \bar{X}^n \right)^{p-2};$$
$$\partial_i^3 g_n(Z^{n,i,x}) = \frac{p(p-1)(p-2)\mathbb{I}(p \geq 3)}{\sqrt{n}^3} \left( \frac{1}{\sqrt{n}} \sum_{i=1}^n X_i + \frac{1}{\sqrt{n}} x \right)^{p-3}.$$
Moreover, according to the Rosenthal inequality for martingales [35], there is a constant $C$ such that
$$\left\| \frac{1}{\sqrt{n}} \sum_{i=1}^n X_i \right\|_{12p} \leq C \|X_1\|_{12p}$$
This implies that: $R_{n,1} = O(n^{-1/2})$, $R_{n,2} = O(n)$, & $R_{n,3} = O(n^{-3/2})$. Theorem 1 concludes the proof. $\square$

## O.2  Proof of Example 4

**Proof:** We define $g_n(X) := \sqrt{n} \prod_{l=1}^n (1 + \frac{X_l}{n})$. Condition $(H_0)$ holds automatically as $g_n$ is smooth. To obtain the desired result we only need to verify that $(H_1)$ holds. By the chain rule we have:
$$\partial_i g_n(Z^{n,i,\bar{X}^n}) = \frac{1}{\sqrt{n}} \prod_{l \neq i} (1 + \frac{Z_l^n}{n}) \times (1 + \frac{\bar{X}^n}{n}); \quad \text{and} \quad \partial_i^2 g_n(Z^{n,i}) = 0.$$
Therefore we note that for condition $(H_1)$ to hold we only need to upper bound $\partial_i g_n(Z^{n,i,\bar{X}^n})$. Using the fact that the observations $(X_i)$ are assumed to be bounded we know that there is a constant $C$ such that
$$\| \prod_{l \neq i} \left( 1 + \frac{Z_l^n}{n} \right) \|_{L_{12}}^{12}$$
$$\overset{(a)}{\leq} \mathbb{E} \left[ \prod_{l \neq i} \mathbb{E} \left( (1 + \frac{Z_l^n}{n})^{12} \Big| X \right) \right]$$
$$\overset{(b)}{\leq} \prod_{l \neq i} (1 + \frac{\max(C,1)}{n}) \sim e^{\max(C,1)}$$

where to get (a) we exploited the conditional independence of the observations $(Z_i^n)$ and to obtain (b) we used the fact that the observations $(X_i)$ are assumed to be bounded. This implies that $(H_1)$ holds.
□

## O.3 Proof of example 8

**Proof:** We want to used Theorem 1 and Corollary 1. Let $(\beta_n) \in \mathbb{R}^{\mathbb{N}}$ be a sequence satisfying: (i) $\beta_n \to \infty$ and (ii) $\beta_n = o(n^{1/4})$

We define $(f_n : \mathbb{R}^n \to \mathbb{R})$ as the following sequence of functions:

$$f_n(x_{1:n}) := \frac{1}{\beta_n} \log \left( 1 + e^{\beta_n \frac{1}{\sqrt{n}} \sum_{i \le n} x_i} \right).$$

We note that the functions $(f_n)$ are three times differentiable functions and that the following holds:

$$\|f_n(Z^n) - g_n(Z^n)\| \le \frac{\log(2)}{\beta_n} \quad \text{and} \quad \|f_n(Z^n) - g_n(Z^n)\| \le \frac{\log(2)}{\beta_n}.$$

Therefore the functions $(f_n)$ and $(g_n)$ satisfy conditions $(H_0)$. Moreover by the chain rule we have:

$$\partial_i f_n(x_{1:n}) = \frac{1}{\sqrt{n}}, \quad \partial_i^2 f_n(x_{1:n}) = \frac{\beta_n}{n}, \quad \partial_i^3 f_n(x_{1:n}) = \frac{\beta_n^2}{n^{3/2}}.$$

This implies that condition $(H_1)$ also holds as $R_{n,1} = O(n^{-1/2})$, $R_{n,2} = o(n^{-1/2})$, & $R_{n,3} = 0(n^{-1})$. Corollary 1 and Theorem 1 conclude the proof.

□

## O.4 Proof of Example 6

**Proof:** For ease of notations for all element $\mathbf{z} \in \mathbb{Z}^2$ we denote by $(\mathbf{z}_1, \mathbf{z}_2)$ its coordinates. Moreover we remark that the i.i.d random variables $(X_{i,j}^n)$ are indexed by $[\![n]\!]^2$ rather than $[\![n^2]\!]$. As there is a one-to-one mapping between those two sets, we note that the random variables $(X_{i,j}^n)$ could be indexed by $[\![n^2]\!]$. Therefore Theorem 1 applies. However for ease of notations we keep the original indexing.

We note that $(g_n)$ is three times differentiable so $(H_0)$ holds. Moreover by the chain rule we have:

$$\partial_{\mathbf{z}} g_n(ZZ^{n,\mathbf{z},\overline{X^n}}) := \frac{1}{n\sqrt{n}} \frac{\sum_{m \in \{-1,1\}^n} m_{\mathbf{z}_1} m_{\mathbf{z}_2} e^{\frac{1}{\sqrt{n}} m^\top ZZ^{n,\mathbf{z},\overline{X^n}} m}}{\sum_{m \in \{-1,1\}^n} e^{\frac{1}{\sqrt{n}} m^\top ZZ^{n,\mathbf{z},\overline{X^n}} m}}.$$

This implies that

$$\left\| \partial_{\mathbf{z}} g_n(ZZ^{n,\mathbf{z},\overline{X^n}}) \right\|_{L_{12}} \le \frac{1}{n\sqrt{n}}.$$

Using the chain rule we have:

$$\partial_{\mathbf{z}}^2 g_n(ZZ^{n,\mathbf{z},\overline{X^n}}) := \frac{1}{n^2} \frac{\sum_{m \in \{-1,1\}^n} (m_{\mathbf{z}_1} m_{\mathbf{z}_2})^2 e^{\frac{1}{\sqrt{n}} m^\top ZZ^{n,\mathbf{z},\overline{X^n}} m}}{\sum_{m \in \{-1,1\}^n} e^{\frac{1}{\sqrt{n}} m^\top ZZ^{n,\mathbf{z},\overline{X^n}} m}} - \frac{1}{n^2} \left[ \frac{\sum_{m \in \{-1,1\}^n} m_{\mathbf{z}_1} m_{\mathbf{z}_2} e^{\frac{1}{\sqrt{n}} m^\top ZZ^{n,\mathbf{z},\overline{X^n}} m}}{\sum_{m \in \{-1,1\}^n} e^{\frac{1}{\sqrt{n}} m^\top ZZ^{n,\mathbf{z},\overline{X^n}} m}} \right]^2.$$

This implies that

$$\left\| \partial_{\mathbf{z}}^2 g_n(ZZ^{n,\mathbf{z},\overline{X^n}}) \right\|_{L_{12}} \le \frac{1}{n^2}.$$

Finally by a last use of the chain rule we have

$$\partial_{\mathbf{z}}^3 g_n(ZZ^{n,\mathbf{z},x}) := \frac{1}{n^{5/2}} \frac{\sum_{m \in \{-1,1\}^n} m_{\mathbf{z}_1} m_{\mathbf{z}_2} e^{\frac{1}{\sqrt{n}} m^\top ZZ^{n,\mathbf{z},\overline{X^n}} m}}{\sum_{m \in \{-1,1\}^n} e^{\frac{1}{\sqrt{n}} m^\top ZZ^{n,\mathbf{z},\overline{X^n}} m}} \left[ 1 - \left[ \frac{\sum_{m \in \{-1,1\}^n} m_{\mathbf{z}_1} m_{\mathbf{z}_2} e^{\frac{1}{\sqrt{n}} m^\top ZZ^{n,\mathbf{z},\overline{X^n}} m}}{\sum_{m \in \{-1,1\}^n} e^{\frac{1}{\sqrt{n}} m^\top ZZ^{n,\mathbf{z},\overline{X^n}} m}} \right]^2 \right]$$

This implies that

$$\left\| \sup_{x \in [Z_{\mathbf{z}}^n, \tilde{X}^n] \cup [\tilde{Y}_{\mathbf{z}}^n, \tilde{X}^n]} |\partial_{\mathbf{z}}^3 g_n(ZZ^{n,\mathbf{z},x})| \right\|_{L_{12}} \le \frac{1}{n^{5/2}}.$$

This implies that the desired result holds. The same can be proved for the centered bootstrap $(\tilde{Z}_{\mathbf{z}}^n)$ and $(Y_{\mathbf{z}}^n)$.

□

# P  Proofs from Appendix J

## P.1  Proof of Proposition 5

**Proof:**  We note that we can suppose without loss of generality that $\mathbb{E}(X_1^n) = 0$. Let $(\beta_n)$ be a sequence of reals satisfying: $(i)$ $\beta_n \to \infty$ and $(ii)$ $\beta_n = o\big(\frac{n^{1/8}}{\sqrt{\log(p_n)}}\big)$.

To get the desired result we use Theorem 8. In this goal we define $(f_n)$ to be the following sequence of function:

$$f_n(x_{1:n}) := \frac{\log\Big(\sum_{l \leq p_n} e^{\beta_n \log(p_n)\frac{1}{\sqrt{n}}\sum_{i \leq n} x'_{i,l}}\Big)}{\beta_n \log(p_n)}.$$

We remark that the functions $(f_n)$ are three times differentiable, and that they satisfy:

$$\begin{aligned}
\Big\|f_n(\tilde{Z}^n) - g_n(\tilde{Z}^n)\Big\|_{L_1} &\leq \frac{1}{\beta_n} \\
\Big\|f_n(Y^n) - g_n(Y^n)\Big\|_{L_1} &\leq \frac{1}{\beta_n}.
\end{aligned} \tag{16}$$

We prove that the conditions of Theorem 8 hold. Using (16) we know that hypothesis $(H_0)$ is holding; and we only need to prove that $(H_1^*)$ also holds. For ease of notations we write:

$$\omega_k(x_{1:n}) := \frac{e^{\beta \log(p_n)\frac{1}{\sqrt{n}}\sum_{i \leq n} x_{i,k}}}{\sum_{l \leq p_n} e^{\beta \log(p_n)\frac{1}{\sqrt{n}}\sum_{i \leq n} x_{i,l}}}.$$

By the chain rule we remark that for all $k_1 \leq p_n$ and all $x_{1:n} \in \mathbb{R}^{p_n}$ we have

$$\Big|\partial_{i,k_1} f_n(x_{1:n})\Big| \leq \frac{1}{\sqrt{n}}\omega_{k_1}(x'_{1:n})$$

As $\sum_{k \leq p_n} \omega_k(x_{1:n}) = 1$ we obtain for all $i \leq n$ that

$$\Big\|\big\|\partial_i f_n(\tilde{Z}^{n,i,\bar{X}^n})\big\|_{v,p_n}\Big\|_{L_{12}} \leq \frac{1}{\sqrt{n}}\Big\|\sum_{j \leq p_n} \omega_j(\tilde{Z}_{1:n}^{n,i,\bar{X}^n})\Big\|_{L_{12}} \leq \frac{1}{\sqrt{n}}.$$

This directly implies that $R_{n,1}^* = O(\frac{1}{n^{1/6}})$. Moreover by using the chain rule we obtain that:

$$\Big|\partial_{i,k_1,k_2}^2 f_n(x_{1:n})\Big| \leq \frac{\beta_n \log(p_n)}{n}\omega_{k_1}(x_{1:n})\omega_{k_2}(x_{1:n}) + \mathbb{I}(k_1 = k_2)\frac{\beta_n \log(p_n)}{n}\omega_{k_1}(x_{1:n})$$

Therefore as $\sum_{k \leq p_n} \omega_k(x_{1:n}) = 1$ we have for all $i \leq n$

$$\Big\|\big\|\partial_i^2 f_n(\tilde{Z}^{n,i,\bar{X}^n})\big\|_{m,p_n}\Big\|_{L_{12}} \leq \frac{2\beta_n \log(p_n)}{n}.$$

this implies that $R_{n,2}^* = O(\beta_n \log(p_n)/\sqrt{n})$. Finally, by another application of the chain rule we have:

$$\Big|\partial_{i,k_{1:3}}^3 f_n(x_{1:n})\Big|$$
$$\leq 2\frac{\beta_n^2 \log(p_n)^2}{n^{\frac{3}{2}}}\omega_{k_1}(x_{1:n})\omega_{k_2}(x_{1:n})\omega_{k_3}(x_{1:n}) + [\mathbb{I}(k_2 = k_3) + \mathbb{I}(k_1 = k_3)$$
$$+ \mathbb{I}(k_1 = k_2)]\frac{\beta_n^2 \log(p_n)^2}{n^{\frac{3}{2}}}\omega_{k_1}(x_{1:n})\omega_{k_2}(x_{1:n}) + \mathbb{I}(k_1 = k_2 = k_3)\frac{\beta_n^2 \log(p_n)^2}{n^{\frac{3}{2}}}\omega_{k_1}(x_{1:n}).$$

Therefore we obtain for all $i \leq n$ that:

$$\Big\|\max_{x \in [\bar{X}^n, Z_i^n]\bigcup[0,\tilde{Y}_i^n]}\big\|\partial_i^3 f_n(Z_{1:n}^{n,i,x})\big\|_{t,p_n}\Big\|_{L_{12}} \leq \frac{6\beta_n^2 \log(p_n)^2}{n^{\frac{3}{2}}}$$

This implies that $R_{3,n}^* = O(\frac{\beta_n^3 \log(p_n)^2}{n^{1/2}})$ Finally by assumption we know that $\|\sup_{l \leq p_n} X_{1,l}^n\|_{L_{12}} < \infty$, hence the assumption $(H_1^*)$ holds as well as the desired result.

$$\square$$

# Q Proofs from Section 4

## Q.1 Proof of Proposition 2

**Proof:** For simplicity we write $\Theta_n := \{\theta_k, \ k \leq p_n\}$ and for all $\theta \in \Theta_n$ we denote

$$g_\theta(x_{1:2}, y_{1:2}) := K_\theta(x_1, x_2) + K_\theta(y_1, y_2) - K_\theta(x_1, y_2) - K_\theta(y_1, x_2);$$

$$D_n := 4 \max \left( \left\| \sup_{k \leq p_n} K_{\theta_k}(X_{1,1}^M, X_{1,1}^M) \right\|_{L_{120}}, \sup_{k \leq p_n} \sum_{l \leq l_n} \lambda_{l,k}, \ 1 \right).$$

Using Mercer's theorem we note that there are orthonormal eigenfunctions $(\psi_{l,k})$ and positive eigenvalues $(\lambda_{l,k})$ such that:

$$K_{\theta_k}(\cdot, \cdot) := \sum_l \lambda_{l,k} \psi_{l,k}(\cdot) \psi_{l,k}(\cdot).$$

As we assumed that $\max_k \sum_l \lambda_{l,k} < \infty$ then there is a sequence $(l_n)$ such that:

$$\max_k \sum_{l \geq l_n} \lambda_{l,k} = o\left( \min(\lambda_n, 1)^2 (\beta_n n p_n D_n^3)^{-1} \right).$$

We show that we can suppose without loss of generality that the kernels $(K_{\theta_k})$ are of rank $l_n$ or less. In this goal, we denote $K_{\theta_k}^*(\cdot, \cdot)$ the following kernels $K_{\theta_k}^*(\cdot, \cdot) := \sum_{l \leq l_n} \lambda_{l,k} \psi_{l,k}(\cdot) \psi_{l,k}(\cdot)$ and shorthand

$$H_{i,j}^{*\theta_k}(X^M) := K_{\theta_k}^*(X_{j,1}^M, X_{i,1}^M) + K_{\theta_k}^*(X_{j,2}^M, X_{i,2}^M) - K_{\theta_k}^*(X_{j,1}^M, X_{i,2}^M) - K_{\theta_k}^*(X_{j,2}^M, X_{i,1}^M);$$

$$p_\theta^*(X^M) := \frac{\frac{1}{n^2} \sum_{i,j \leq n} H_{i,j}^{*\theta}(X^M)}{\frac{4}{|n|^3} \sum_{i \leq n} \left[ \sum_{j \leq n} H_{i,j}^{*,\theta}(X^M) \right]^2 - \frac{4}{n^4} \left[ \sum_{i,j \leq n} H_{i,j}^{*,\theta}(X^M) \right]^2 + \lambda_n}.$$

$$\omega_k^*(x_{1:n}) := \frac{e^{\beta_n p_{\theta_k}^*(X^M)}}{\sum_{k' \leq p_n} e^{\beta_n p_{\theta'_k}^*(X^M)}}$$

We prove that:

$$\left\| n T_n(X^M) - \sum_{k \leq p_n} \frac{1}{n} \sum_{i,j \leq n} H_{i,j}^{*\theta_k}(X^M) \omega_k^*(X^M) \right\|_{L_1} \to 0$$

and that

$$\left\| n T_n(Z^M) - \sum_{k \leq p_n} \frac{1}{n} \sum_{i,j \leq n} H_{i,j}^{*\theta_k}(Z^M) \omega_k^*(Z^M) \right\|_{L_1} \to 0.$$

In this goal, we remark that:

$$\sup_{i,j \leq n} \left\| \sup_{k \leq p_n} \left| \left( H_{i,j}^{*\theta_k}(X^M) - H_{i,j}^{\theta_k} \right) \right| \right\|_{L_4} \leq \sup_{i,j \leq n} \sum_{k \leq p_n} \left\| \left( H_{i,j}^{*\theta_k}(X^M) - H_{i,j}^{\theta_k} \right) \right\|_{L_4}$$

$$\overset{(a)}{\leq} 4 \sup_{i,j \leq n} \left\| \sup_{k \leq p_n} \left| K_{\theta_k}^*(X_{i,1}^M, X_{j,2}^M) - K_{\theta_k}(X_{i,1}^M, X_{j,2}^M) \right| \right\|_{L_4}$$

$$\leq 4 \sup_{i,j \leq n} \left\| \sup_{k \leq p_n} \sum_{l \geq l_n} \lambda_{l,k} \left| \psi_{l,k}(X_{i,1}^M) \psi(X_{j,2}^M) \right| \right\|_{L_4}$$

$$\leq 4 p_n \sup_{k \leq p_n} \sum_{l \geq l_n} \lambda_{l,k} \sup_{i \leq n} \sup_{k \leq p_n, l \in \mathbb{N}} \left\| \psi_{l,k}(X_{i,1}^M)^2 \right\|_{L_4}$$

$$\leq 4 p_n D_n \sup_{k \leq p_n} \sum_{l \geq l_n} \lambda_{l,k}$$

where to get (a) we used the fact that $X_{i,1}^M$ has the same distribution than $X_{i,2}^M$.

We remark that the function $\sigma_n : (x_1, \ldots, x_{p_n}) \to \sum_{k \le p_n} H_{i,j}^{*\theta_k}(X^M) \frac{e^{\beta_n x_k}}{\sum_{k' \le p_n} e^{\beta_n x_{k'}}}$ is Lipchitz in the max norm: $|\sigma_n(x) - \sigma_n(y)| \le \beta_n \sup_{k \le p_n} \left| H_{i,j}^{*\theta_k}(X^M) \right| \max_{k \le p_n} |x_k - y_k|$. Therefore coupling this with the triangle inequality we get that

$$\left\| n T_n(X^M) - \sum_{k \le p_n} \frac{1}{n} \sum_{i,j \le n} H_{i,j}^{*\theta_k}(X^M) \omega_k^*(X^M) \right\|_{L_1}$$

$$\le \left\| \sum_{\substack{k \le p_n \\ i,j \le n}} \frac{1}{n} \left( H_{i,j}^{*\theta_k}(X^M) - H_{i,j}^{\theta_k} \right) \omega_k(X^M) \right\|_{L_1} + \left\| \frac{1}{n} \sum_{\substack{k \le p_n \\ i,j \le n}} H_{i,j}^{*\theta_k}(X^M) \left( \omega_k(X^M) - \omega_k^*(X^M) \right) \right\|_{L_1} \quad (17)$$

$$\overset{(a)}{\le} n \left\| \sup_{k \le p_n} \left| \frac{1}{n^2} \sum_{i,j \le n} \left( H_{i,j}^{*\theta_k}(X^M) - H_{i,j}^{\theta_k} \right) \right| \right\|_{L_1} + 2n\beta_n \sup_{i,j \le n} \left\| \sup_{k \le p_n} \left| H_{i,j}^{*\theta_k}(X^M) \right| \right\|_{L_2} \left\| \sup_{k \le p_n} \left| p_{\theta_k}^*(X^n) - p_{\theta_k}(X^n) \right| \right\|_{L_2}$$

The first term of (17) is bounded by:

$$n \left\| \sup_{k \le p_n} \left| \frac{1}{n^2} \sum_{i,j \le n} \left( H_{i,j}^{*\theta_k}(X^M) - H_{i,j}^{\theta_k} \right) \right| \right\|_{L_1} \le n p_n \sup_{k \le p_n} \sum_{l \ge l_n} \lambda_{l,k} \to 0.$$

Moreover we can bound the second term of (17) using the triangular inequality, (18) and the fact that

$$\left( \frac{4}{|n|^3} \sum_{i \le n} \left[ \sum_{j \le n} H_{i,j}^{*,\theta} \right]^2 - \frac{4}{n^4} \left[ \sum_{i,j \le n} H_{i,j}^{*,\theta} \right]^2 + \lambda_n \right)^{-1} \ge \lambda_n, \qquad \left\| \sup_{k \le p_n} \left| H_{i,j}^{*\theta_k}(X^M) \right| \right\|_{L_4} \le D_n.$$

Indeed we have,

$$2n\beta_n \sup_{i,j \le n} \left\| \sup_{k \le p_n} \left| H_{i,j}^{*\theta_k}(X^M) \right| \right\|_{L_2} \left\| \sup_{k \le p_n} \left| p_{\theta_k}^*(X^n) - p_{\theta_k}(X^n) \right| \right\|_{L_2}$$

$$\le \frac{2n}{\lambda_n} \beta_n D_n \left\| \sup_{k \le p_n} \left| \frac{1}{n^2} \sum_{i,j \le n} \left( H_{i,j}^{*\theta_k}(X^M) - H_{i,j}^{\theta_k} \right) \right| \right\|_{L_2} + \frac{2n\beta_n}{\lambda_n^2} D_n^2 \sup_{k \le p_n} \left\| \frac{4}{|n|^3} \sum_{i \le n} \left( [\sum_{j \le n} H_{i,j}^{*\theta_k}(X^M)]^2 \right. \right.$$

$$\left. \left. - [\sum_{j \le n} H_{i,j}^{\theta_k}]^2 \right) - \frac{4}{n^4} \left( \left[ \sum_{i,j \le n} H_{i,j}^{*\theta_k}(X^M) \right]^2 - \left[ \sum_{i,j \le n} H_{i,j}^{\theta_k} \right]^2 \right) \right\|_{L_2} \quad (18)$$

$$\le 4n p_n \sup_{k \le p_n} \sum_{l \ge l_n} \lambda_{l,k} \left[ 1 + \frac{2D_n \beta_n}{\lambda_n} + \frac{16 D_n^3}{\lambda_n^2} \right] \to 0.$$

This implies directly that: $\left\| n T_n(X^M) - \sum_{k \le p_n} \frac{1}{n} \sum_{i,j \le n} H_{i,j}^{*\theta_k}(X^M) \omega_k^*(X^M) \right\|_{L_1} \to 0.$ Following the exact same road map we can show that: $\left\| n T_n(Z^M) - \sum_{k \le p_n} \frac{1}{n} \sum_{i,j \le n} H_{i,j}^{*\theta_k}(Z^M) \omega_k^*(Z_{1:n}^M) \right\|_{L_1} \to 0.$

Therefore we have

$$\left\| d_{\mathcal{F}} \left( n T_n(Z^M), \sum_{k \le p_n} \frac{1}{n} \sum_{i,j \le n} H_{i,j}^{*\theta_k}(Z^M) \omega_k^*(Z^M) \mid X^n \right) \right\|_{L_1} \to 0;$$

$$\left\| d_{\mathcal{F}} \left( n T_n(Y^M), \sum_{k \le p_n} \frac{1}{n} \sum_{i,j \le n} H_{i,j}^{*\theta_k}(Y^M) \omega_k^*(Y^M) \mid X^n \right) \right\|_{L_1} \to 0.$$

Hence using theorem 5 we know that to establish the desired result it is sufficient to study the asymptotics of $\frac{1}{n}\sum_{i,j\leq n} H_{i,j}^{*\theta_k}(Z^M)\omega_k^*(Z^M)$. Hence we can suppose without loss of generality that all the Kernels $(K_{\theta_i})$ have ranks of $l_n$ or less, and do so

As the distribution of $X_{i,1}^M$ is the same than the one of $X_{i,2}^M$ they also have the same mean embedding. Moreover we note that for all $c\in\mathbb{R}$ we have :

$$H_{i,j}^{\theta_k} = \sum_{l\leq l_n}\lambda_{l,k}\big[\psi_{l,k}(X_{i,1}^M)-\psi_{l,k}(X_{i,2}^M)\big]\big[\psi_{l,k}(X_{j,1}^n)-\psi_{l,k}(X_{j,2}^n)\big]$$

$$= \sum_{l\leq l_n}\lambda_{l,k}\big[\psi_{l,k}(X_{i,1}^M)+c-\big(\psi_{l,k}(X_{i,2}^M)+c\big)\big]\big[\big(\psi_{l,k}(X_{j,1}^n)+c\big)-\big(\psi_{l,k}(X_{j,2}^n)+c\big)\big].$$

Therefore as the test statistics $\hat{T}_n$ depends only on $(H_{i,j}^{\theta_k})$ we can suppose without loss of generality that

$$\mathbb{E}(\psi_{l,k}(X_{1,1}^M))=\mathbb{E}(\psi_{l,k}(X_{1,2}^M))=0,\quad \forall\leq l_n,\ k\leq p_n.$$

We define $X_i^{*,n}:=(X_{i,l,k}^*)_{l,k}$ to be random variables, defined as

$$X_{i,l,k}^*:=\psi_{l,k}(X_{i,1}^n)-\psi_{l,k}(X_{i,2}^n).$$

We define the process $X^{*,n}:=\big(X_i^{*,n}\big)$ and note that the observations $(X_i^{*,n})$ take value in $M_{p_n\times l_n}(\mathbb{R})$. We remark that we could have taken the observations $(X_i^{*,n})$ to take value in $\mathbb{R}^{p_nl_n}$ as there is a one to one mapping from $M_{p_n\times l_n}(\mathbb{R})$ to $\mathbb{R}^{p_nl_n}$. However for ease of notations we keep them as defined. Let $(Z_i^*)$ and $(Y_i^*)$ be defined in the following way:

$$Z_i^*:=\big(\psi_{l,k}(Z_{i,1}^M)-\psi_{l,k}(Z_{i,2}^M)\big)\quad\text{and}\quad Y_i^*:=\big(\psi_{l,k}(Y_{i,1}^M)-\psi_{l,k}(Y_{i,2}^M)\big).$$

We note that they form respectively a bootstrap sample and an independent copy of $(X_i^*)$. Moreover we remark that $Z_{i,1}^M|X^n\stackrel{d}{=}Z_{i,2}^M|X^n$ therefore we can also assume with out loss of generality that $\mathbb{E}(Z_{i,l,k}^*|X^n)=0$. We choose $(g_n)$ to be the following sequence of functions:

$$g_n(x_{1:n}):=\sum_k\omega_k(x_{1:n})\sum_l\lambda_{l,k}\Big(\frac{1}{\sqrt{n}}\sum_{j\leq n}x_{j,l,k}\Big)^2$$

where we have set $\omega_k(x_{1:n})=\frac{e^{\beta_n h_{n,k}(x_{1:n})}}{\sum_k e^{\beta_n h_{n,k}(x_{1:n})}}$ and

$$h_{n,k}(x_{1:n}):=\frac{\sum_l\lambda_{l,k}\Big(\frac{1}{n}\sum_{j\leq n}x_{j,l,k}\Big)^2}{\frac{4}{n^3}\sum_l\lambda_{l,k}(\sum_i x_{i,l,k})^3-\frac{4}{n^4}(\sum_l\lambda_{l,k}(\sum_i x_{i,l,k})^2)^2+\lambda_n}.$$

We remark that we have

$$g_n(Z_{1:n}^*)=nT_n(Z^M),\quad g_n(\tilde{Y}_{1:n}^*)=nT_n(Y^M),\quad g_n(X_{1:n}^*)=nT_n(X^M).$$

It is therefore enough to study $g_n$ and $(X_{1:n}^*)$. For ease of notations we write:

$$\widehat{\sigma_{n,k}}(x_{1:n}):=\frac{4}{n^3}\sum_l\lambda_{l,k}(\sum_i x_{i,l,k})^3-\frac{4}{n^4}(\sum_l\lambda_{l,k}(\sum_i x_{i,l,j})^2)^2+\lambda_n;$$

and

$$\overline{x_{l,k_1}}:=\frac{1}{n}\sum_{j\leq n}x_{j,l,k_1},\quad \hat{H}_{n,k}(x_{1:n}):=\sum_l\lambda_{l,k}\Big(\frac{1}{\sqrt{n}}\sum_{j\leq n}x_{j,l,k}\Big)^2.$$

Moreover we note that we have

$$\big\|\sup_{k\leq p_n}\sum_{l\leq l_n}\lambda_{l,k}[X_{1,l,k}^*]^2\big\|_{L_{120}}\stackrel{(a)}{\leq}2\big\|\sup_{k\leq p_n}\sum_{l\leq l_n}\lambda_{l,k}\big([\psi_{l,k}(X_{i,1}^M)]^2+[\psi_{l,k}(X_{i,2}^M)]^2\big)\big\|_{L_{120}}$$

$$\stackrel{(b)}{\leq}4\big\|\sup_{k\leq p_n}K_{\theta_k}(X_{1,1}^M,X_{1,1}^M)\big\|_{L_{120}}$$

(19)

where (a) and (b) come from the Cauchy-Schwarz inequality. Similarly we have:

$$
\left\| \sup_{k \leq p_n} \sum_{l \leq l_n} \lambda_{l,k} \left| X_{1,l,k}^* \right| \right\|_{L_{120}} \overset{(a)}{\leq} \sqrt{\sup_{k \leq p_n} \sum_{l \leq l_n} \lambda_{l,k}} \sqrt{\left\| \sup_{k \leq p_n} \sum_{l \leq l_n} \lambda_{l,k} \left| X_{1,l,k}^* \right|^2 \right\|_{L_{120}}}
$$

$$
\leq 2 \sqrt{\sup_{k \leq p_n} \sum_{l \leq l_n} \lambda_{l,k}} \sqrt{\left\| \sup_{k \leq p_n} K_{\theta_k}(X_{1,1}^M, X_{1,1}^M) \right\|_{L_{120}}} \tag{20}
$$

where (a) comes from the Cauchy-Schwarz inequality. Therefore using Theorem 3 we note that there is a constant $C$ such that:

$$
\left\| \sup_{k \leq p_n} \sup_{z \in [\bar{X}^n, Z_1^*] \cup [\bar{X}^n, \tilde{Y}_1^*]} \frac{1}{\sqrt{n}} \hat{H}_{n,k}(Z^{*,i,z}) \right\|_{L_{120}}
$$

$$
\overset{(a)}{\leq} \frac{2}{n^{3/2}} \left\| \sup_{k \leq p_n} \sum_{l \leq l_n} \lambda_{l,k} [X_{1,l,k}^*]^2 \right\|_{L_{120}} + 2 \left\| \sup_{k \leq p_n} \frac{1}{\sqrt{n}} \sum_{l \leq l_n} \lambda_{l,k} \left( \frac{1}{\sqrt{n}} \sum_{l \leq n} Z_l^{*,n,i} \right)^2 \right\|_{L_{120}}
$$

$$
\overset{(b)}{\leq} C \log(p_n) \left\| \sup_{k \leq p_n} K_{\theta_k}(X_{1,1}^M, X_{1,1}^M) \right\|_{L_{120}}.
$$

where (a) is a consequence of the triangle inequality and (b) of Theorem 3 and (19). Similarly using Theorem 3 and (20) we can establish that there is a constant $C'$ such that:

$$
\left\| \sup_{k \leq p_n} \sup_{z \in [\bar{X}^n, Z_1^*] \cup [\bar{X}^n, \tilde{Y}_1^*]} \sum_{l \leq l_n} \lambda_{l,k} \frac{1}{\sqrt{n}} \left| z_{l,k} + \sum_{i \geq 2} Z_{i,l,k}^* \right| \right\|_{L_{120}}
$$

$$
\leq C' \log(p_n) \sqrt{\sup_{k \leq p_n} \sum_{l \leq l_n} \lambda_{l,k}} \sqrt{\left\| \sup_{k \leq p_n} K_{\theta_k}(X_{1,1}^M, X_{1,1}^M) \right\|_{L_{120}}}
$$

To prove the desired result we use Theorem 8. We remark that the functions $(h_{n,k})$ and $(g_n)$ are three times differentiable. This implies that $(H_0)$ hold. We check that $(H_1^*)$ also holds. In this goal we first check that the partial derivatives of $(h_{n,k})$ are bounded. For ease of notations for a function $f_n$ we shorthand:

$$
\partial_{i,k,l} f_n(x_{1:n}) := \partial_{x_{i,k,l}} f_n(x_{1:n}), \quad \partial_{i,k_{1:2},l_{1:2}}^2 f_n(x_{1:n}) := \partial_{x_{i,k_1,l_1}, x_{i,k_2,l_2}}^2 f_n(x_{1:n})
$$

$$
\partial_{i,k_{1:3},l_{1:3}}^2 f_n(x_{1:n}) := \partial_{x_{i,k_1,l_1}, x_{i,k_2,l_2} x_{i,k_3,l_3}}^3 f_n(x_{1:n}).
$$

In this goal, using the chain rule we note that for all $k \leq p_n$ and all $l \leq l_n$ we have:

$$
\partial_{i,k,l} \hat{\sigma}_{n,k}(x_{1:n}) := \frac{\lambda_{l,k}}{n} \left( 12 \overline{x_{l,k}}^2 - 16 \overline{x_{l,k}} \frac{\hat{H}_{n,k}(x_{1:n})}{n} \right)
$$

and:

$$
\partial_{i,k,l} h_{n,k}(x_{1:n}) = \frac{2}{n} \frac{\lambda_{l,k} \overline{x_{l,k}}}{\widehat{\sigma_{n,k}}(x_{1:n})} - \frac{\hat{H}_{n,k}(x_{1:n}) \partial_{i,k,l} \hat{\sigma}_{n,k}(x_{1:n})}{\widehat{\sigma_{n,k}}(x_{1:n})^2}
$$

Therefore we obtain that there is a constant $K$ such that:

$$
n \left\| \sup_{k \leq p_n} \sum_{l \leq l_n} \left| \partial_{i,k,l} h_{n,k}(Z^{n,i,\bar{X}^n}) \right| \right\|_{L_{12}} \leq K \left[ \frac{D_n}{\lambda_n} + \frac{D_n^2 + D_n^3}{\lambda_n^2} \right]
$$

Using once again the chain rule we have that:

$$
\partial_{i,k,l} \omega_{k'}(x_{1:n}) = \beta_n \partial_{i,k,l} h_{n,k}(x_{1:n}) \omega_{k'}(x_{1:n}) \left( \mathbb{I}(k' = k) - \omega_k(x_{1:n}) \right)
$$

as well as:

$$\partial_{i,k,l}g_n(x_{1:n}) = \frac{2}{\sqrt{n}}\lambda_{l,k}\left(\frac{1}{\sqrt{n}}\sum_{j\leq n}x_{j,l,k}\right)\omega_k(x_{1:n}) + \sum_{k'\leq p_n}\hat{H}_{n,k'}(x_{1:n})\partial_{i,k,l}\omega_{k'}(x_{1:n})$$

Therefore, using (19) and (20) and as $\sum_{k\leq p_n}\omega_k(x_{1:n}) = 1$ we obtain that there is a constant $K_2$ that do not depend on $n$ such that

$$\sup_{i\leq n}\left\|\left\|\partial_{i,k,l}g_n(Z_{1:n}^{*,i,\bar{X}^n})\right\|_{v,p_n\times l_n}\right\|_{L_{12}}$$

$$\leq \beta_n \sup_{i\leq n}\left\|\sup_{k\leq p_n}\left|\frac{1}{\sqrt{n}}\hat{H}_{n,k}(Z^{*,i,\bar{X}^n})\right|\right\|_{L_{24}}\left\|\sup_{k\leq p_n}\sum_{l\leq l_n}\sqrt{n}|\partial_{i,k,l}h_{n,k}(Z^{*,i,\bar{X}^n})|\right\|_{L_{24}}$$

$$+ \sup_{i\leq n}\left\|\sup_{k\leq p_n}\sum_{l\leq l_n}|\lambda_{l,k}\left(\frac{1}{\sqrt{n}}\sum_{j\leq n}Z_{j,l,k}^{*,i,\bar{X}^n}\right)|\right\|_{L_{12}}$$

$$\leq \frac{K_2\beta_n\log(p_n)D_n^4}{\sqrt{n}\min(\lambda_n^2,1)}.$$

This implies that $R_{n,1}^* := O\left(\frac{\beta_n\log(p_n)D_n^4}{n^{1/6}\min(\lambda_n^2,1)}\right)$. Moreover all $k_1, l_1, k_2, l_2$ using the chain rule we have: $\partial_{i,k_{1:2},l_{1:2}}^2 h_{n,k_1}(x_{1:n}) = 0$ if $k_1$ is distinct from $k_2$. Moreover if $k_1 = k_2$ we have:

$$\partial_{i,k_{1:2},l_{1:2}}\hat{\sigma}_{n,k}(x_{1:n}) := \frac{\lambda_{k_1,l_1}}{n^2}\left(24\overline{x_{l_1,k_1}}\mathbb{I}(l_1=l_2) - 16\frac{\hat{H}_{n,k_1}(x_{1:n})}{n}\mathbb{I}(l_1=l_2) - 16\lambda_{k_2,l_2}\overline{x_{l_1,k_1}x_{l_2,k_1}}\right)$$

and by the Chain rule we have:

$$\partial_{i,k_{1:2},l_{1:2}}^2 h_{n,k_1}(x_{1:n}) = \frac{2\lambda_{k_1,l_1}}{n\widehat{\sigma_{n,k_1}}(x_{1:n})}\left[\frac{\mathbb{I}(l_1=l_2)}{n} + \frac{\overline{x_{l_2,k_1}}\partial_{i,k_2,l_2}\widehat{\sigma_{n,k_1}}(x_{1:n})}{\widehat{\sigma_{n,k_1}}(x_{1:n})}\right]$$

$$- \frac{\partial_{i,k_2,l_2}h_{n,k_2}(x_{1:n})}{n}\frac{\partial_{i,k_1,l_1}\widehat{\sigma_{n,k_1}}(x_{1:n})}{\widehat{\sigma_{n,k_1}}(x_{1:n})} - \frac{\hat{H}_{n,k_1}(x_{1:n})}{n}\frac{\partial_{i,k_{1:2},l_{1:2}}^2\widehat{\sigma_{n,k_1}}(x_{1:n})}{\widehat{\sigma_{n,k_1}}(x_{1:n})}$$

$$+ 2\frac{\hat{H}_{n,k_1}(x_{1:n})}{n}\frac{\partial_{i,k_1,l_1}\widehat{\sigma_{n,k_1}}(x_{1:n})\partial_{i,k_2,l_2}\widehat{\sigma_{n,k_1}}(x_{1:n})}{\widehat{\sigma_{n,k_1}}(x_{1:n})^3}$$

Therefore there is a constant $K_3$ such that:

$$\left\|\sup_{k_{1:2}\leq p_n}\sum_{l_{1:2}\leq l_n}\left|\partial_{i,k_{1:2},l_{1:2}}^2 h_{n,k_1}(Z_{1:n}^{*,i,\bar{X}^n})\right|\right\|_{L_{12}} \leq \frac{K_3D_n^5}{\max(\lambda_n^3,1)n^2}.$$

By another application of the chain rule we have:

$$\partial_{i,k_{1:2},l_{1:2}}\omega_{k'}(x_{1:n}) = \beta_n\left(\mathbb{I}(k'=k_1) - \omega_{k_1}(x_{1:n})\right)$$
$$\times \left(\partial_{i,k_{1:2},l_{1:2}}^2 h_{n,k_1}(x_{1:n})\omega_{k'}(x_{1:n}) + \partial_{i,k_1,l_1}h_{n,k_1}(x_{1:n})\partial_{i,k_2,l_2}\omega_{k'}(x_{1:n})\right)$$
$$- \beta_n\partial_{i,k_1,l_1}h_{n,k_1}(x_{1:n})\omega_{k'}(x_{1:n})\partial_{i,k_2,l_2}\omega_{k_1}(x_{1:n})$$

by the chain rule we also have that:

$$\partial_{i,k_{1:2},l_{1:2}}g_n(x_{1:n}) = \frac{2}{n}\lambda_{k_1,l_1}\mathbb{I}(k_1=k_2)\omega_{k_1}(x_{1:n}) + \frac{2}{\sqrt{n}}\lambda_{k_1,l_1}\left(\frac{1}{\sqrt{n}}\sum_{j\leq n}x_{j,l,k_1}\right)\partial_{i,k_2,l_2}\omega_{k_1}(x_{1:n})$$

$$+ \frac{2}{\sqrt{n}}\lambda_{k_2,l_2}\left(\frac{1}{\sqrt{n}}\sum_{j\leq n}x_{j,k_2,l_2}\right)\partial_{i,k_1,l_1}\omega_{k_2}(x_{1:n})$$

$$+ \sum_{k'\leq p_n}\hat{H}_{n,k'}(x_{1:n})\partial_{i,k_{1:2},l_{1:2}}^2\omega_{k'}(x_{1:n})$$

Therefore we know that there are constants $K_3$ such that:

$$\left\|\left\|\partial_{x_1}^2 g_n(Z_{1:n}^{*,i,\bar{X}^n})\right\|_{m,p_n\times l_n}\right\|_{L_{12}} \leq \frac{K_3D_n^7}{\min(\lambda_n^4,1)n}\beta_n^2$$

This implies that $R_{n,2}^* = O(\frac{\beta_n^2 D_n^7}{\min(\lambda_n^4,1)\sqrt{n}})$. Finally using a similar line of reasoning we note that Moreover all $k_1, l_1, k_2, l_2, l_3, k_3$ using the chain rule we have: $\partial_{i,k_{1:3},l_{1:3}}^3 h_{n,k_1}(x_{1:n}) = 0$ if $k_1$ is distinct from $k_2$ or from $k_3$. Moreover if $k_1 = k_2 = k_3$ we have:

$$\partial_{i,l_{1:3},k_{1:3}}^3 \hat{\sigma}_{n,k}(x_{1:n}) = \frac{\lambda_{k_1,l_1}}{n^3}\Big(24\mathbb{I}(l_1 = l_2 = l_3) - 32\lambda_{k_1,l_3}\overline{x_{l_3,k_1}}\mathbb{I}(l_1 = l_2)$$
$$- 16\lambda_{k_2,l_2}(\overline{x_{l_1,k_1}}\mathbb{I}(l_3 = l_2) + \overline{x_{l_2,k_1}}\mathbb{I}(l_1 = l_3)))\Big)$$

and by the Chain rule we have:

$$\partial_{i,k_{1:3},l_{1:3}}^3 h_{n,k_1}(x_{1:n})$$
$$= -\frac{\partial_{l_3,k_3}\widehat{\sigma_{n,k_1}}(x_{1:n})}{n\widehat{\sigma_{n,k_1}}(x_{1:n})^2}\Big(\frac{\overline{x_{l_1,k_1}}\partial_{i,k_2,l_2}\widehat{\sigma_{n,k_1}}(x_{1:n})}{\widehat{\sigma_{n,k_1}}(x_{1:n})}\Big[2\lambda_{k_1,l_1} + 3\frac{\hat{H}_{n,k_1}(x_{1:n})\partial_{i,k_1,l_1}\widehat{\sigma_{n,k_1}}(x_{1:n})}{\widehat{\sigma_{n,k_1}}(x_{1:n})}\Big]$$
$$- \partial_{i,k_2,l_2}\hat{H}_{n,k_2}(x_{1:n})\partial_{i,k_1,l_1}\widehat{\sigma_{n,k_1}}(x_{1:n}) - h_{n,k_1}(x_{1:n})\partial_{i,k_{1:2},l_{1:2}}^2\widehat{\sigma_{n,k_1}}(x_{1:n}) + 2\lambda_{k_1,l_1}\frac{\mathbb{I}(l_1 = l_2)}{n}\Big)$$
$$+\frac{2\lambda_{k_1,l_1}}{n\widehat{\sigma_{n,k_1}}(x_{1:n})^2}\Big(\frac{\partial_{i,k_2,l_2}\widehat{\sigma_{n,k_1}}(x_{1:n})\mathbb{I}(l_2 = l_3)}{n} + \overline{x_{l_3,k_1}}\partial_{i,k_{2:3},l_{2:3}}^2\widehat{\sigma_{n,k_1}}(x_{1:n})\Big)$$
$$-\frac{\partial_{i,k_{2:3},l_{2:3}}h_{n,k_2}(x_{1:n})}{n}\frac{\partial_{i,k_1,l_1}\widehat{\sigma_{n,k_1}}(x_{1:n})}{\widehat{\sigma_{n,k_1}}(x_{1:n})} - \frac{\partial_{i,k_1,l_1,k_3,l_3}^2\widehat{\sigma_{n,k_1}}(x_{1:n})}{n\widehat{\sigma_{n,k_1}}(x_{1:n})}\Big(\partial_{i,k_2,l_2}h_{n,k_2}(x_{1:n})$$
$$- \hat{H}_{n,k_1}(x_{1:n})\frac{\partial_{i,k_2,l_2}\widehat{\sigma_{n,k_1}}(x_{1:n})}{\widehat{\sigma_{n,k_1}}(x_{1:n})}\Big) - \frac{2\lambda_{l_3,k_1}\bar{X}_{l_3,k_1}}{n\widehat{\sigma_{n,k_1}}(x_{1:n})}\Big(\partial_{i,k_{1:2},l_{1:2}}^2\widehat{\sigma_{n,k_1}}(x_{1:n}) - \frac{\widehat{\sigma_{n,k_1}}(x_{1:n})\partial_{i,k_2,l_2}\widehat{\sigma_{n,k_1}}(x_{1:n})}{\widehat{\sigma_{n,k_1}}(x_{1:n})}\Big)$$
$$- \frac{\hat{H}_{n,k_1}(x_{1:n})}{n\widehat{\sigma_{n,k_1}}(x_{1:n})}\Big(\partial_{i,k_{1:3},l_{1:3}}^3\widehat{\sigma_{n,k_1}}(x_{1:n}) - \frac{\partial_{i,k_1,l_1}\widehat{\sigma_{n,k_1}}(x_{1:n})\partial_{i,k_{2:3},l_{2:3}}^2\widehat{\sigma_{n,k_1}}(x_{1:n})}{\widehat{\sigma_{n,k_1}}(x_{1:n})}\Big)$$

Therefore there is a constant $K_3$ such that:

$$\Big\|\sup_{k_{1:3}\leq p_n}\sup_{z\in[\bar{X}^n,Z_i^*]\cup[\bar{X}^n,\tilde{Y}_i^*]}\sum_{l_{1:3}\leq l_n}\Big|\partial_{i,k_{1:3},l_{1:3}}^3 h_{n,k_1}(Z_{1:n}^{*,i,z})\Big|\Big\|_{L_{12}} \leq \frac{K_3 D_n^7}{\max(\lambda_n^4,1)n^3}.$$

By another application of the chain rule we have:

$$\partial_{i,k_{1:3},l_{1:3}}\omega_{k'}(x_{1:n}) = \beta_n\left(\mathbb{I}(k' = k_1) - \omega_{k_1}(x_{1:n})\right)$$
$$\times\Big(\partial_{i,k_{1:3},l_{1:3}}^3 h_{n,k_1}(x_{1:n})\omega_{k'}(x_{1:n}) + \partial_{i,k_{1:2},l_{1:2}}^2 h_{n,k_1}(x_{1:n})\partial_{i,k_3,l_3}\omega_{k'}(x_{1:n})$$
$$+ \partial_{i,k_1,l_1}h_{n,k_1}(x_{1:n})\partial_{i,k_{2:3},l_{2:3}}^2\omega_{k'}(x_{1:n}) + \partial_{i,k_1,l_1,i,k_3,l_3}^2 h_{n,k_1}(x_{1:n})\partial_{i,k_2,l_2}\omega_{k'}(x_{1:n})\Big)$$
$$-\beta_n\partial_{i,k_3,l_3}\omega_{k_1}(x_{1:n})$$
$$\times\left(\partial_{i,k_{1:2},l_{1:2}}^2 h_{n,k_1}(x_{1:n})\omega_{k'}(x_{1:n}) - \partial_{i,k_1,l_1}h_{n,k_1}(x_{1:n})\partial_{i,k_2,l_2}\omega_{k'}(x_{1:n})\right)$$
$$-\beta_n\omega_{k'}(x_{1:n})$$
$$\times\Big(\partial_{i,k_1,l_1,i,k_3,l_3}^2 h_{n,k_1}(x_{1:n})\partial_{i,k_2,l_2}\omega_{k_1}(x_{1:n}) + \partial_{i,k_1,l_1}h_{n,k_1}(x_{1:n})\partial_{i,k_{2:3},l_{2:3}}^2\omega_{k_1}(x_{1:n})\Big)$$

as well as:

$$\partial_{i,k_{1:3},l_{1:3}}^3 g_n(x_{1:n})$$
$$=\frac{2}{n}\lambda_{k_1,l_1}\mathbb{I}(k_1 = k_2 = k_3)\partial_{i,k_3,l_3}\omega_{k_1}(x_{1:n}) + \frac{2}{\sqrt{n}}\lambda_{k_1,l_1}(\frac{1}{\sqrt{n}}\sum_{j\leq n}x_{j,l,k_1})\partial_{i,k_{2:3},l_{2:3}}\omega_{k_1}(x_{1:n})$$
$$+ \frac{2\mathbb{I}(k_1 = k_3)}{n}\lambda_{k_1,l_1}\partial_{i,k_2,l_2}\omega_{k_1}(x_{1:n}) + \frac{2\mathbb{I}(k_2 = k_3)}{n}\lambda_{k_2,l_2}\partial_{i,k_1,l_1}\omega_{k_2}(x_{1:n})$$
$$+ \frac{2}{\sqrt{n}}\lambda_{k_2,l_2}(\frac{1}{\sqrt{n}}\sum_{j\leq n}x_{j,k_2,l_2})\partial_{i,k_1,l_1,i,k_3,l_3}^2\omega_{k_2}(x_{1:n})$$
$$+ \frac{2}{\sqrt{n}}\lambda_{k_3,l_3}(\frac{1}{\sqrt{n}}\sum_{j\leq n}x_{j,k_3,l})\partial_{i,k_{1:2},l_{1:2}}^2\omega_{k_3}(x_{1:n}) + \sum_{k'\leq p_n}\hat{H}_{n,k'}(x_{1:n})\partial_{i,k_{1:3},l_{1:3}}^3\omega_{k'}(x_{1:n})$$

Therefore we know that there are constants $K_4$ such that:

$$\max_i \left\| \max_{z \in [\bar{X}^n, Z_i^*] \cup [\bar{X}^n, \tilde{Y}_i^*]} \left\| \partial_{x_1}^2 g_n(Z_{1:n}^{*,i,z}) \right\|_{t, p_n \times l_n} \right\|_{L_{12}} \leq \frac{K_4 D_n^{10}}{\min(\lambda_n^6, 1) n^2} \beta_n^3$$

Therefore we have $R_{n,3}^* = O(\frac{D_n^{10} \beta_n^3}{\min(\lambda_n^6, 1) n})$.

This implies that both $(H_0)$ and $(H_1^*)$ hold and Theorem 8 guarantees that

$$\left\| d_{\mathcal{F}}\left( n\hat{T}_n(Z^M), g_n(Y^M) \mid X^n \right) \right\|_{L_1} \to 0.$$

$\square$

# R    Proof of Proposition 3

**Proof:**   Let $(Y_i^{n,2})$ be an independent copy of $(X_i^n)$ and of $(Y_i^n)$. We shorthand

$$\omega_n^p(x_{1:n-m_n}) := \frac{e^{-\beta_n \mathcal{R}_n^k(x_{1:n})}}{\sum_{k' \leq p_n} e^{-\beta_n \mathcal{R}_n^{k'}(x_{1:n})}}, \qquad \hat{\theta}_n^p := \hat{\theta}_n^p(X_{1:m_n}^n);$$

$$\hat{\Theta}_n := \hat{\Theta}_n(X_{1:n}^n), \qquad \hat{\Theta}_n^Y := \hat{\Theta}_n(X_{1:m_n}^n Y_{m_n+1:n}^{n,2}).$$

The proof works in two stage. In a first time we show that: $\frac{1}{\sqrt{n-m_n}} \sum_{i \leq n-m_n} \mathcal{L}_n(Y_{i+m_n}^n, \hat{\Theta}_n(Y_{i+m_n}^n))$ has approximately the same distribution than $\frac{1}{\sqrt{n-m_n}} \sum_{i \leq n-m_n} \mathcal{L}_n(Y_{i+m_n}^n, \hat{\Theta}_n^Y(Y_{i+m_n}^n))$; and we show that $\frac{1}{\sqrt{n-m_n}} \sum_{i \leq n-m_n} \mathcal{L}_n(Z_{i+m_n}^n, \hat{\Theta}_n(Z_{i+m_n}^n))$ as approximately the same distribution than $\frac{1}{\sqrt{n-m_n}} \sum_{i \leq n-m_n} \mathcal{L}_n(Z_{i+m_n}^n, \hat{\Theta}_n^Y(Z_{i+m_n}^n))$. Then we notice that this implies that we only need to show that the bootstrap method is consistent for $\frac{1}{\sqrt{n-m_n}} \sum_{i \leq n-m_n} \mathcal{L}_n(Y_{i+m_n}^n, \hat{\Theta}_n^Y(Y_{i+m_n}^n))$.

Firstly, we realize that as conditionally on $\hat{\Theta}_n$ and $\hat{\Theta}_n^Y$ the observations $\left( \mathcal{L}_n\left( Y_{i+m_n}^n, \hat{\Theta}_n^Y(Y_{i+m_n}^n) \right) - \mathcal{L}_n\left( Y_{i+m_n}^n, \hat{\Theta}_n(Y_{i+m_n}^n) \right) \right)_{i \geq 0}$ are independent and identically distributed we have

$$\left\| \sqrt{\mathrm{var}\left[ \frac{1}{\sqrt{n-m_n}} \sum_{i \leq n-m_n} \mathcal{L}_n\left( Y_{i+m_n}^n, \hat{\Theta}_n^Y(Y_{i+m_n}^n) \right) - \mathcal{L}_n\left( Y_{i+m_n}^n, \hat{\Theta}_n(Y_{i+m_n}^n) \right) \Big| \hat{\Theta}_n^Y, \hat{\Theta}_n \right]} \right\|$$

$$= \left\| \sqrt{\mathrm{var}\left[ \mathcal{L}_n\left( X_{m_n+1}^n, \hat{\Theta}_n^Y(Y_{m_n+1}^n) \right) - \mathcal{L}_n\left( Y_{1+m_n}^n, \hat{\Theta}_n(Y_{m_n+1}^n) \right) \Big| \hat{\Theta}_n, \hat{\Theta}_n^Y \right]} \right\|_{L_1}$$
(21)

Moreover by exploiting Taylor expansions we know that

$$\left\| \sqrt{\mathrm{var}\left[ \mathcal{L}_n\left( Y_{m_n+1}^n, \hat{\Theta}_n^Y(Y_{m_n+1}^n) \right) - \mathcal{L}_n\left( Y_{m_n+1}^n, \hat{\Theta}_n(Y_{m_n+1}^n) \right) \Big| \hat{\Theta}_n, \hat{\Theta}_n^Y \right]} \right\|_{L_1}$$

$$\leq \sum_{l \leq d_n'} \left\| \sup_{\theta \in \Omega(\{\hat{\theta}_n^p, \, p \leq p_n\})} |\partial_{2,l} \mathcal{L}_n(X_{m_n+1}^n, \theta(X_{m_n+1}^n))| \right\|_{L_\infty} \times \left\| \hat{\Theta}_{n,l}(X_{m_n+1}^n) - \hat{\Theta}_{n,l}^Y(X_{m_n+1}^n) \right\|_{L_2}$$

$$\leq L_n d_n' \sup_{l \leq d_n'} \left\| \hat{\Theta}_{n,l}(X_{m_n+1}^n) - \hat{\Theta}_{n,l}^Y(X_{m_n+1}^n) \right\|_{L_2}$$

We define $(X_i^{n,j})$ as the following interpolating process: $X_i^{n,j} := \begin{cases} X_i^n & \text{if } i \leq j \\ Y_j^n & \text{otherwise} \end{cases}$ and shorthand $\hat{\Theta}_{n,l}^j := \hat{\Theta}_{n,l}(X_{1:n}^{n,j})$. By the Effron-Stein inequality we have:

$$\left\| \hat{\Theta}_{n,l}(X_{m_n+1}^n) - \hat{\Theta}_{n,l}^Y(X_{m_n+1}^n) \right\|_{L_2}^2 \leq \sum_{j \geq m_n+1} \left\| \hat{\Theta}_{n,l}^j(X_{m_n+1}^n) - \hat{\Theta}_{n,l}^{j-1}(X_{m_n+1}^n) \right\|_{L_2}^2$$

$$\leq (n-m_n) \left\| \hat{\Theta}_{n,l}^n(X_{m_n+1}^n) - \hat{\Theta}_{n,l}^{n-1}(X_{m_n+1}^n) \right\|_{L_2}^2$$
(22)

Moreover using the triangle inequality and the inequality $(a + b)^2 \leq 2(a^2 + b^2)$ we can upper-bound the right-hand side as

$$\left\| \hat{\Theta}_{n,l}^n(X_{m_n+1}^n) - \hat{\Theta}_{n,l}^{n-1}(X_{m_n+1}^n) \right\|_{L_2}^2$$

$$= \left\| \sum_{p \leq p_n} \hat{\theta}_n^p(X_{m_n+1}^n)_l \left( \omega_n^p(X_{m_n+1:n}^n) - \omega_n^p(X_{m_n+1:n-1}^n Y_n^n) \right) \right\|_{L_2}^2$$

$$\leq 2 \left\| \sum_{p \leq p_n} \hat{\theta}_n^p(X_{m_n+1}^n)_l \omega_n^p(X_{m_n+1:n}^n) \left[ e^{\frac{\beta_n}{n-m_n}[\mathcal{L}_n(X_n^n, \hat{\theta}_n^p) - \mathcal{L}_n(Y_n^n, \hat{\theta}_n^p)]} - 1 \right] \right\|_{L_2}^2$$

$$+ 2 \left\| \sum_{p \leq p_n} \hat{\theta}_n^p(X_{m_n+1}^n)_l \omega_n^p(X_{m_n+1:n}^n) \sum_{p' \leq p_n} \omega_n^{p'}(X_{m_n+1:n}^n) \left[ e^{\frac{\beta_n}{n-m_n}[\mathcal{L}_n(X_n^n, \hat{\theta}_n^{p'}) - \mathcal{L}_n(Y_n^n, \hat{\theta}_n^{p'})]} - 1 \right] \right\|_{L_2}^2$$

$$\leq 8 \frac{\beta_n^2}{(n-m_n)^2} \left\| \sup_{p \leq p_n} |\hat{\theta}_n^p(X_{m_n+1}^n)_l| \right\|_{L_\infty}^2 \left\| \sup_{p \leq p_n} \left| \mathcal{L}_n(X_1^n, \hat{\theta}_n^p) \right| \right\|_{L_\infty}^2 e^{\frac{2\beta_n}{n-m_n} \| \sup_{p \leq p_n} \mathcal{L}_n(X_1^n, \hat{\theta}_n^p) \|_{L_\infty}}$$

$$\leq 8 \frac{\beta_n^2}{(n-m_n)^2} T_n^2 L_n^2 e^{\frac{2\beta_n}{n-m_n} L_n}$$

By combining (21) and (22) we therefore obtain that:

$$\left\| \sqrt{\mathrm{var} \left[ \frac{1}{\sqrt{n-m_n}} \sum_{i \leq n-m_n} \mathcal{L}_n\left( Y_{i+m_n}^n, \hat{\Theta}_n^Y(Y_{i+m_n}^n) \right) - \mathcal{L}_n\left( Y_{i+m_n}^n, \hat{\Theta}_n(Y_{i+m_n}^n) \right) \Big| \hat{\Theta}_n^Y, \hat{\Theta}_n \right]} \right\|_{L_1}$$

$$\longrightarrow 0.$$

This implies that

$$\left\| d_{\mathcal{F}} \left( \frac{1}{\sqrt{n-m_n}} \sum_{i \leq n-m_n} \mathcal{L}_n\left( Y_{i+m_n}^n, \hat{\Theta}_n^Y(Y_{i+m_n}^n) \right), \frac{1}{\sqrt{n-m_n}} \sum_{i \leq n-m_n} \mathcal{L}_n\left( Y_{i+m_n}^n, \hat{\Theta}_n(Y_{i+m_n}^n) \right) \right) \right\|_{L_1}$$

$$\longrightarrow 0.$$

Using the same line of reasoning we also prove that

$$\left\| d_{\mathcal{F}} \left( \frac{1}{\sqrt{n-m_n}} \sum_{i \leq n-m_n} \mathcal{L}_n\left( Z_{i+m_n}^n, \hat{\Theta}_n^Y(Z_{i+m_n}^n) \right), \frac{1}{\sqrt{n-m_n}} \sum_{i \leq n-m_n} \mathcal{L}_n\left( Z_{i+m_n}^n, \hat{\Theta}_n(Z_{i+m_n}^n) \right) \right) \right\|_{L_1}$$

$$\longrightarrow 0.$$

Hence by using the fact that $d_{\mathcal{F}}$ satisfies the triangle inequality (see Theorem 5) we observe that it is enough to prove that the distribution of $\frac{1}{\sqrt{n}} \sum_{i \leq n} \mathcal{L}_n\left( X_{i+m_n}^n, \hat{\Theta}_n^Y(X_{i+m_n}^n) \right)$ can be correctly approximated by the bootstrap method. We note that the process $\left( \mathcal{L}_n\left( X_{i+m_n}^n, \hat{\Theta}_n^Y(X_{i+m_n}^n) \right) \right)$ is exchangeable and that if we define the function: $g_n(x_1, \ldots, x_n) = \frac{1}{\sqrt{n}} \sum_{i \leq n} x_i$ then:

$$g_n\left( \left( \mathcal{L}_n\left( X_{i+m_n}^n, \hat{\Theta}_n^Y(X_{i+m_n}^n) \right) \right)_{1:n} \right) := \frac{1}{\sqrt{n}} \sum_{i \leq n} \mathcal{L}_n\left( X_{i+m_n}^n, \hat{\Theta}_n^Y(X_{i+m_n}^n) \right).$$

Therefore we only need to check that $(g_n)$ satisfy all the conditions of Theorem 10. Firstly we note that:

$$\partial_1 g_n(x_{1:n}) := \frac{1}{\sqrt{n}}, \quad \partial_2 g_n(x_{1:n}) = 0 \quad \text{and} \quad \partial_3 g_n(x_{1:n}) = 0.$$

Therefore we have

$$R_{n,1} = n^{-1/6}, \quad R_{n,2} = 0, \quad \& \quad R_{n,3} = 0.$$

Therefore the conditions of Theorem 10 hold and

$$\left\| d_{\mathcal{F}}\left( \frac{1}{\sqrt{n-m_n}} \sum_{i \leq n-m_n} \mathcal{L}_n\left(Z^n_{i+m_n}, \hat{\Theta}^Y_n(Z^n_{i+m_n})\right), \frac{1}{\sqrt{n-m_n}} \sum_{i \leq n-m_n} \mathcal{L}_n\left(Y^n_{i+m_n}, \hat{\Theta}_n(Y^n_{i+m_n})\right) \right) \Big| X^n \right\|_{L_1}$$
$$\longrightarrow 0.$$

We obtain the desired result by noting that $d_{\mathcal{F}}$ satisfies the triangle inequality.

$\square$

# S    Proof of Proposition 6

For simplicity we write

$$T_{l,n} := \left\| \sup_{p \leq p_n} \left| \hat{\theta}^p_n(X^n_{1:m_n})(Y^{n,1}_1)_l \right| \right\|_{L_\infty} \vee 1,$$
$$L^*_{l_{1:i},n} := \left\| \max_{p \leq p_n} \left| \partial^i_{2,l_{1:i}} \mathcal{L}_n(X^n_n, \hat{\theta}^p_n(X^n_{1:m_n})(X^n_n)) \right| \right\|_{L_\infty} \vee 1.$$

**Proof:**    For ease of notations, we shorthand:

$$\hat{\Theta}^{Z^{n,2}}_n := \hat{\Theta}_n(X^n_{1:m_n} Z^{n,2}_{1:n-m_n}), \qquad \hat{\theta}^p_n := \hat{\theta}^p_n(X^n_{1:m_n}).$$

We first notice using theorem 5 and theorem 6 that:

$$\left\| d_{\mathcal{F}}\left( \mathcal{R}^{\mathrm{s}}_{\hat{\Theta}^{Z^{n,2}}_n}(Z^{n,1}_{1:n-m_n}) - \mathbb{E}\left( \mathcal{R}^{\mathrm{s}}_{\hat{\Theta}^{Z^{n,2}}_n}(Z^{n,1}_{1:n-m_n}) \mid \hat{\Theta}^{Z^{n,2}}_n, X^n \right), \right.\right.$$
$$\left.\left. \mathcal{R}^{\mathrm{s}}_{\hat{\Theta}'_n}(Y^{n,1}_{1:n-m_n}) - \mathbb{E}\left( \mathcal{R}^{\mathrm{s}}_{\hat{\Theta}'_n}(Y^{n,1}_{1:n-m_n}) \mid \hat{\Theta}'_n \right) \mid X^n \right) \right\|_{L_1}$$
$$\leq \left\| d_{\mathcal{F}}\left( \mathcal{R}^{\mathrm{s}}_{\hat{\Theta}^{Z^{n,2}}_n}(Z^{n,1}_{1:n-m_n}) - \mathbb{E}\left( \mathcal{R}^{\mathrm{s}}_{\hat{\Theta}^{Z^{n,2}}_n}(Z^{n,1}_{1:n-m_n}) \mid \hat{\Theta}^{Z^{n,2}}_n, X^n \right) \right.\right.$$
$$\left.\left. , \mathcal{R}^{\mathrm{s}}_{\hat{\Theta}^{Z^{n,2}}_n}(Y^{n,1}_{1:n-m_n}) - \mathbb{E}\left( \mathcal{R}^{\mathrm{s}}_{\hat{\Theta}^{Z^{n,2}}_n}(Y^{n,1}_{1:n-m_n}) \mid \hat{\Theta}^{Z^{n,2}}_n \right) \Big| X^n, Z^{n,2} \right) \right\|_{L_1} \qquad (23)$$
$$+ \left\| d_{\mathcal{F}}\left( \mathcal{R}^{\mathrm{s}}_{\hat{\Theta}^{Z^{n,2}}_n}(Y^{n,1}_{1:n-m_n}) - \mathbb{E}\left( \mathcal{R}^{\mathrm{s}}_{\hat{\Theta}^{Z^{n,2}}_n}(Y^{n,1}_{1:n-m_n}) \mid \hat{\Theta}^{Z^{n,2}}_n \right), \right.\right.$$
$$\left.\left. \mathcal{R}^{\mathrm{s}}_{\hat{\Theta}'_n}(Y^{n,1}_{1:n-m_n}) - \mathbb{E}\left( \mathcal{R}^{\mathrm{s}}_{\hat{\Theta}'_n}(Y^{n,1}_{1:n-m_n}) \mid \hat{\Theta}'_n \right) \mid X^n, Y^{n,1} \right) \right\|_{L_1}$$

We upper-bound each term separately. In this goal, we define as $(X'_i)$ the process defined as $X'_i = \mathcal{L}_n(X^n_{i+m_n}, \hat{\Theta}^{Z^{n,2}}_n(X^n_{i+m_n}))$. We remark that the sequence $(X'_i)$ is an exchangeable sequence and we define:

$$Z'_i := \mathcal{L}_n(Z^{n,1}_i, \hat{\Theta}^{Z^{n,2}}_n(Z^{n,1}_i)), \quad Y'_i := \mathcal{L}_n(Y^{n,1}_i, \hat{\Theta}^{Z^{n,2}}_n(Y^{n,1}_i)).$$

We note that $(Z'_i)$ and $(Y'_i)$ respectively form a bootstrap sample of $(X'_i)_{i \leq n-m_n}$ and a copy of $(X'_i)$ that is conditionally on $\hat{\Theta}^{Z^{n,2}}_n$ independent. We define $F_{1,n}$ to be the following functions:

$$F_{1,n}(x_{1:n-m_n}) := \frac{1}{\sqrt{n-m_n}} \sum_{i \leq n-m_n} x_i.$$

Moreover we denote $(X^*_i)$ the following random vectors $X^*_i = \left( \mathcal{L}_n(X^n_{i+m_n}, \hat{\theta}^p_n(X^n_{i+m_n})) \right)_{p \leq p_n}$.
We also define:

$$Z^*_i = \left( \mathcal{L}_n(Z^{n,2}_{i+m_n}, \hat{\theta}^p_n(Z^{n,2}_{i+m_n})) \right)_{p \leq p_n}, \quad Y^*_i = \left( \mathcal{L}_n(Y^{n,2}_{i+m_n}, \hat{\theta}^p_n(Y^{n,2}_{i+m_n})) \right)_{p \leq p_n}.$$

We remark that $(Z^*_i)$ and $(Y^*_i)$ are respectively bootstrap samples and (conditionally) independent copy of $(X^*_i)$.

We define the following weight functions: $\omega_n^p : \times_{i=1}^{n-m_n} \mathbb{R} \to \mathbb{R}$ as:

$$\omega_n^p(x_{1:n-m_n}) := \frac{e^{\frac{-\beta_n}{n-m_n} \sum_{i \leq n-m_n} x_{i,p}}}{\sum_{p' \leq p_n} e^{\frac{-\beta_n}{n-m_n} \sum_{i \leq n-m_n} x_{i,p'}}}.$$

We define $F_{2,n} : \times_{l=1}^{n} \mathbb{R}^{p_n} \to \mathbb{R}$ as the following random function:

$$F_{2,n}(x_{1:n-m_n}) := \frac{1}{\sqrt{n-m_n}} \sum_{i \leq n-m_n} \mathcal{L}_n \left( Y_i^{n,1}, \sum_{p \leq p_n} \hat{\theta}_n^p(Y_i^{n,1}) \omega_n^p(x_{1:n-m_n}) \right)$$

$$- \mathbb{E} \left( \mathcal{L}_n \left( Y_i^{n,1}, \sum_{p \leq p_n} \hat{\theta}_n^p(Y_i^{n,1}) \omega_n^p(x_{1:n-m_n}) \right) \mid X^n \right).$$

We remark that, conditionally on $X^n$ and $Y^{n,1}$, the functions $(F_{2,n})$ are independent from $(Z_i^*)$ and $(Y_i^*)$. Using (23) we note that if the following hold then the desired result also holds:

$$\left\| d_{\mathcal{F}} \left( F_{2,n}(Z_{1:n-m_n}^*), F_{2,n}(\tilde{Y}_{1:n-m_n}^*) | X^n \right) \right\|_{L_1} \to 0;$$

$$\left\| d_{\mathcal{F}} \left( F_{1,n}(Z_{1:n-m_n}'), F_{1,n}(\tilde{Y}_{1:n-m_n}') \mid X^n \right) \right\|_{L_1} \to 0.$$

To prove that those hold we use Theorem 8 extended to exchangeable sequences and random functions. We notice that the functions $(F_{2,n})$ and $(F_{1,n})$ are three times differentiable (random) functions. Therefore for both functions hypothesis $(H_0)$ holds and to get the desired results we only need to check that hypothesis $(H_1^*)$ also holds. For simplicity for all $i \leq 3$ we write

$$\hat{R}_j^{n,l_{1:i}}(x_{1:n}) := \frac{1}{\sqrt{n-m_n}} \left( \partial_{2,l_{1:i}}^i \mathcal{L}_n \left( Y_j^{n,1}, \sum_{p \leq p_n} \hat{\theta}_n^p(Y_j^{n,1}) \omega_n^p(x_{1:n-m_n}) \right) \right.$$

$$\left. - \mathbb{E} \left( \partial_{2,l_{1:i}}^i \mathcal{L}_n \left( Y_j^{n,1}, \sum_{p \leq p_n} \hat{\theta}_n^p(Y_j^{n,1}) \omega_n^p(x_{1:n-m_n}) \right) \mid X^n \right) \right).$$

We denote $\Omega_n^{\mathcal{P}} := \{\omega_i^*, \ i \leq p_n\}$ a maximal $d_n' n^{-3/4}$-packing of $B_{d_n'}^{d_n' T_n}$, it is well known that $p_n \leq \left( n^{3/4} 2 T_n + 1 \right)^{d_n'}$. We remark that for all $\theta \in B_{d_n'}^{d_n' T_n}$ there is an $\omega^* \in \Omega_n^{\mathcal{P}}$ such that $\|\omega^* - \omega\| \leq d_n' T_n$. Indeed if this would not be the case then $\{\omega\} \bigcup \Omega_n^{\mathcal{P}}$ would also be a packing set of $B_{d_n'}^{d_n' T_n}$ contradicting the fact that we assumed it to be maximal. We define the following random functions: $f_{l_{1:3},\theta}(z) := \partial_{2,l_{1:3}}^3 \mathcal{L}_n(z,\theta)$. We note that for every $\theta, \theta' \in \Omega_{\{\hat{\theta}_p, \ p \leq p_n\}}$ we have

$$\left\| \sup_{\theta, \theta' \in \Omega_{\{\hat{\theta}_p, \ p \leq p_n\}}} \left| f_{l_{1:3},\theta}(Y_1^{n,1}) - f_{l_{1:3},\theta}(Y_1^{n,2}) \right| \right\|_{L_\infty} \leq \frac{L_n^* \beta_n T_n \|\theta' - \theta\|_{d_n'}}{(n-m_n)}.$$

Using theorem 3 we observe that this implies that there are constant $K, K'$ such that:

$$\max_{i \leq n} \left\| \sup_{x \in [\bar{X}^{n,*}, Z_1^*] \bigcup [\bar{X}^{n,*}, \tilde{Y}_1^*]} \left| \sum_{j \leq n-m_n} \hat{R}_j^{n,l_{1:3}}(Z_{1:n}^{*,i,x}) \right| \right\|_{L_{72}}$$

$$\leq d_n' L_n^* \left( \frac{\beta_n T_n}{n^{3/4}} + K \log(2 T_n n^{3/4} + 1) \right) \tag{24}$$

$$\leq K' d_n' L_n^* \log(n).$$

Moreover we note that for all $k \leq 2$ there is a constant $C$ such that

$$\max_{i \leq n} \left\| \sum_{j \leq n-m_n} \hat{R}_j^{n,l_{1:k}}(Z_{1:n}^{*,i,\overline{Z^*}}) \right\|_{L_{72}} \leq C L_{n,l_{1:k}}^*. \tag{25}$$

For ease of notations we write: $\overline{\theta_{n,l}^p}(x_{1:n-m_n},y) := \hat{\theta}_n^p(y)_l - \sum_{p' \leq p_n} \hat{\theta}_n^{p'}(y)_l \omega_n^{p'}(x_{1:n-m_n})$. We remark that for all $p \leq p_n$ we have

$$\partial_{i,p} F_{2,n}(x_{1:n-m_n})$$

$$= -\frac{\beta_n}{n-m_n} \omega_n^p(x_{1:n-m_n}) \sum_{l \leq d_n'} \sum_{j \leq m_n} \hat{R}_l^{n,j}(x_{1:n}) \Big[ \hat{\theta}_n^p(Y_j^{n,1})_l - \sum_{p' \leq p_n} \sum_{j \leq m_n} \hat{\theta}_n^{p'}(Y_j^{n,1})_l \omega_n^{p'}(x_{1:n-m_n}) \Big]$$

$$= -\frac{\beta_n}{n-m_n} \omega_n^p(x_{1:n-m_n}) \sum_{l \leq d_n'} \sum_{j \leq m_n} \hat{R}_l^{n,j}(x_{1:n}) \overline{\theta_{n,l}^p}(x_{1:n-m_n}, Y_j^{n,1})$$

Therefore as $\sum_{p \leq p_n} \omega_n^p(x_{1:n}) = 1$, using (25) we obtain that there is a constant $C$ that does not depend on $n$ such that

$$\max_{i \leq n} \Big\| \big\| \partial_i F_{2,n}(Z_{1:n-m_n}^{*,i,\overline{X^*}}) \big\|_{v,p_n} \Big\|_{L_{12}} \leq \frac{C\beta_n}{n-m_n} \sum_{l \leq d_n'} L_{l,n}^* \Big\| \max_{p \leq p_n} |\hat{\theta}_n^p(Y_1^{n,1})_l| \Big\|_{L_{24}}.$$

Moreover for all $p \leq p_n$ we have $\partial_i F_{1,n}(x_{1:n-m_n}) = \frac{1}{\sqrt{n-m_n}}$. Therefore we obtain that

$$\max_i \Big\| \partial_i F_{1,n}(Z_{1:n-m_n}^{',i,\overline{X'}}) \Big\|_{L_{12}} \leq \frac{1}{\sqrt{n-m_n}}.$$

In addition, we remark that we have $\partial_i^2 F_{1,n}(x_{1:n-m_n}) = 0$. Therefore the condition $(H_1^*)$ holds for $F_{1,n}$. For ease of notations we write $\omega_n^{p_1,p_2}(x_{1:n-m_n}) := \mathbb{I}(p_1 = p_2) - \omega_n^{p_2}(x_{1:n-m_n})$. We have

$$\partial_{i,p_1,p_2}^2 F_{2,n}(x_{1:n-m_n})$$

$$= \frac{\beta_n^2 \omega_n^{p_1}(x_{1:n-m_n})}{(n-m_n)^2} \omega_n^{p_1,p_2}(x_{1:n-m_n}) \sum_{l \leq d_n'} \sum_{j \leq m_n} \hat{R}_l^{n,j}(x_{1:n}) \overline{\theta_{n,l}^{p_1}}(x_{1:n}, Y_j^{n,1})$$

$$+ \frac{\beta_n^2 \omega_n^{p_1}(x_{1:n-m_n}) \omega_n^{p_2}(x_{1:n-m_n})}{(n-m_n)^2} \sum_{l_1,l_2 \leq d_n'} \sum_{j \leq m_n} \hat{R}_{l_1,l_2}^{n,j}(x_{1:n}) \overline{\theta_{n,l_1}^{p_1}}(x_{1:n}, Y_j^{n,1}) \overline{\theta_{n,l_2}^{p_2}}(x_{1:n}, Y_j^{n,1})$$

$$- \frac{\beta_n^2}{(n-m_n)^2} \omega_n^{p_1}(x_{1:n-m_n}) \omega_n^{p_2}(x_{1:n-m_n}) \sum_{l \leq d_n'} \sum_{j \leq m_n} \hat{R}_l^{n,j}(x_{1:n}) \overline{\theta_{n,l}^{p_2}}(x_{1:n}, Y_j^{n,1}).$$

This combined with (25) implies that there is a constant $C_2$ such that:

$$\max_i \Big\| \big\| \partial_i^2 F_{2,n}(Z_{1:n-m_n}^{*,i,\overline{X^*}}) \big\|_{m,p_n} \Big\|_{L_{12}} \leq \frac{C_2 \beta_n^2}{(n-m_n)^2} \Big[ \sum_{l \leq d_n'} L_{l,n}^* T_{l,n} + \sum_{l \leq d_n'} L_{l_1,l_2,n}^* T_{l_1,n} T_{l_2,n} \Big]$$

Finally using the chain rule for all $p_1, p_2, p_3 \leq p_n$ we have

$$\partial_{i,p_{1:3}}^3 F_{2,n}(x_{1:n-m_n})$$

$$= \frac{\beta_n^3 \sum_{l \leq d_n'} \sum_{j \leq m_n} \hat{R}_l^{n,j}(x_{1:n}) \overline{\theta_{n,l_1}^{p_1}}(x_{1:n}, Y_j^{n,1}) \omega_n^{p_1}(x_{1:n-m_n})}{(n-m_n)^3} \Big[ \omega_n^{p_2}(x_{1:n-m_n}) \omega_n^{p_2,p_3}(x_{1:n-m_n})$$

$$- \omega_n^{p'}(x_{1:n-m_n}) \omega_n^{p_1,p_3}(x_{1:n-m_n}) \Big] - \frac{\beta_n^3 \omega_n^{p_1}(x_{1:n-m_n})}{(n-m_n)^3} \sum_{l_1,l_2 \leq d_n'} \sum_{j \leq m_n} \hat{R}_{l_1,l_2}^{n,j}(x_{1:n}) \overline{\theta_{n,l_1}^{p_1}}(x_{1:n}, Y_j^{n,1})$$

$$\times \overline{\theta_{n,l_3}^{p_3}}(x_{1:n}, Y_j^{n,1}) \Big[ \omega_n^{p_3}(x_{1:n-m_n}) \omega_n^{p_1,p_2}(x_{1:n-m_n}) + \omega_n^{p_3}(x_{1:n-m_n}) \omega_n^{p_1,p_2}(x_{1:n-m_n}) \Big]$$

$$- \frac{\beta_n^3 \prod_{i=1}^3 \omega_n^{p_i}(x_{1:n-m_n})}{(n-m_n)^3} \Big[ \sum_{l_{1:3} \leq d_n'} \sum_{j \leq p_n} \hat{R}_{l_1,l_2,l_3}^{n,j}(x_{1:n}) \prod_{k=1}^3 \overline{\theta_{n,l_k}^{p_k}}(x_{1:n}, Y_j^{n,1}) - \sum_{l_{1:2} \leq d_n'} \hat{R}_{l_1,l_2}^{n,j}(x_{1:n}, Y_j^{n,1})$$

$$\times \overline{\theta_{n,l_1}^{p_2}}(x_{1:n}, Y_j^{n,1}) \overline{\theta_{n,l_2}^{p_3}}(x_{1:n}, Y_j^{n,1}) \Big] + \frac{\beta_n^3 \omega_n^{p_1}(x_{1:n}) \omega_n^{p_2}(x_{1:n}) \big[ \omega_n^{p_1,p_3}(x_{1:n-m_n}) + \omega_n^{p_2,p_3}(x_{1:n-m_n}) \big]}{(n-m_n)^3}$$

$$\times \sum_{l \leq d_n'} \sum_{j \leq m_n} \hat{R}_l^{n,j}(x_{1:n}) \overline{\theta_{n,l_1}^{p_2}}(x_{1:n}, Y_j^{n,1})$$

Therefore using (24) we establish that that there is a constant $C_3 < \infty$ such that

$$\max_{i \leq n} \left\| \max_{x \in [\bar{X}^n, Z_1^*] \cup [\bar{X}^n, \tilde{Y}_1^*]} \left\| \partial_i^3 F_{2,n}(Z_{1:n-m_n}^{*,i,x}) \right\|_{t,p_n} \right\|_{L_{12}}$$

$$\leq \frac{C_3 \beta_n^3 d_n' L_n^* \log(n)}{(n-m_n)^3} \sum_{l_1 \leq d_n'} T_{l_1,n} \Big[ L_{l_1,n}^* + \sum_{l_2 \leq d_n'} L_{l_{1:2},n}^* T_{l_2,n} + \sum_{l_{2:3} \leq d_n'} L_{l_{1:3},n}^* T_{l_2,n} T_{l_3,n} \Big]$$

This implies that hypothesis $(H_1^*)$ holds for $F_{2,n}$; which means that

$$\left\| d_{\mathcal{F}} \left( F_{2,n}(Z_{1:n-m_n}^*), F_{2,n}(\tilde{Y}_{1:n-m_n}^*) | X^n \right) \right\|_{L_1} \to 0; \tag{26}$$

We note that $F_{2,n}(\tilde{Y}_{1:n-m_n}^*) := \mathcal{R}_{\hat{\Theta}_n'}^s(Y_{1:n-m_n}^{n,1})$.

Moreover, we remark that we have proved that the condition $(H_1^*)$ also holds for $F_{1,n}$. This implies that we have

$$\left\| d_{\mathcal{F}} \left( F_{1,n}(Z_{1:n-m_n}'), F_{1,n}(\tilde{Y}_{1:n-m_n}') \mid X^n \right) \right\|_{L_1} \to 0.$$

As $F_{1,n}$ is linear this gives us the desired result.

We now prove Proposition 7. Firstly using the same line of reasoning we can prove that

$$\left\| d_{\mathcal{F}} \Big( \mathbb{E} \big( \mathcal{R}_{\hat{\Theta}_n^{Z^{n,2}}}^s (Z_{1:n-m_n}^{n,1}) \mid Z^{n,1}, X^n \big), \mathbb{E} \big( \mathcal{R}_{\hat{\Theta}_n^{Z^{n,2}}}^s (Y_{1:n-m_n}^{n,1}) \mid Y^{n,1}, X^n \big) \Big| X^n \Big) \right\|_{L_1}$$
$$\to 0.$$

Secondly using (26) we note that we have:

$$\left\| d_{\mathcal{F}} \Big( \mathbb{E} \big( F_{2,n}(Z_{1:n-m_n}^*) \big| Y^{n,1}, X^n \big), \mathbb{E} \big( F_{2,n}(\tilde{Y}_{1:n-m_n}^*) \big| Y^{n,1}, X^n \big) | X^n \Big) \right\|_{L_1} \to 0.$$

Therefore for all $\delta > 0$ we have:

$$\liminf_{n \to \infty} P\left( \big| \mathbb{E}\big( \mathcal{R}_{\hat{\Theta}_n^{Z^{n,2}}}^s (Y_{1:n-m_n}^{n,1}) \mid Y^{n,1}, X^n \big) - \mathbb{E}\big( \mathcal{R}_{\hat{\Theta}_n'}^s (Y_{1:n-m_n}^{n,1}) \mid X^n \big) \big| > t_{b,n}^{\alpha/2}(X^n) + \delta \right) \geq 1 - \alpha/2.$$

Moreover let $N^n \sim N(0, \mathrm{Id})$ be a Gaussian vector of dimension $d_n$. Using [19] we know that there is a constant $C$ that does not depend on $n$ such that the following holds:

$$\sup_t \left| P\left[ \sqrt{n} \max_k \big| \bar{Y}_k^* - \mathbb{E}(Y_k^*) \big| \geq t \right] - P(\max_k |(\Sigma_n N^n)_k| \geq t) \right|$$
$$\leq \frac{C \log(d_n)^{7/6} \max(\| \max_j |X_{1,j}^n| \|_{L_4}^4, 1)}{n^{1/6}}$$

For ease of notation we write $S_{i,p} := \mathcal{L}(X_i^n, \hat{\theta}_p^{\setminus i})$ and set $\epsilon > 0$. We note that conditionally on $X^n$ the random variable $|\hat{\Sigma} - \Sigma|Z$ is a Gaussian vector. Therefore using Appendix A of [12] we obtain that:

$$P\left( \max_{p \leq p_n} \left| (\hat{\Sigma} - \Sigma)Z \right|_p \geq \epsilon \right)$$
$$\leq \frac{1}{\epsilon} \mathbb{E}\left( \max_{p \leq p_n} \left| (\hat{\Sigma} - \Sigma)Z \right|_p \right)$$
$$\leq \frac{1}{\epsilon} \sqrt{2 \log(2p_n)} \mathbb{E}\left( \max_{p \leq p_n} \sqrt{((\hat{\Sigma} - \Sigma)^2)_{p,p}} \right)$$
$$\overset{(a)}{\leq} \frac{1}{\epsilon} \sqrt{2 \log(2p_n)} \mathbb{E}\left( \max_{p \leq p_n} \sqrt{(\hat{\Sigma}^2 - \Sigma^2)_{p,p}} \right)$$
$$\leq \frac{1}{\epsilon} \sqrt{2 \log(2p_n)} \sqrt{\mathbb{E}\left( \max_{p \leq p_n} |\hat{\Sigma}^2 - \Sigma^2|_{p,p} \right)}.$$

where to get (a) we exploited the fact that $\hat{\Sigma}$ and $\Sigma$ are positive definite symmetric matrices. We denote the variance of $S_{1,p}$ as $(\Sigma_p^{*,n})^2 := \mathrm{Var}\Big(\mathcal{L}(X_1^n, \hat{\theta}_p^{\backslash 1})\Big)$. By the triangle inequality have:

$$\mathbb{E}\big(\max_{p \leq p_n} |\hat{\Sigma}_{p,p}^2 - \Sigma_{p,p}^2|\big)$$

$$\leq \mathbb{E}\left(\max_{p \leq p_n} \left| \frac{1}{m_n - 1} \sum_{i \leq m_n} \big(S_{i,p} - \bar{S}_p\big)^2 - (\Sigma^{*,n})_{p,p}^2 \right|\right) + \max_{p \leq p_n} |(\Sigma^{*,n})_{p,p}^2 - \Sigma_{p,p}^2|$$

By definition we note that:

$$\left|(\Sigma^{*,n})_{p,p}^2 - \Sigma_{p,p}^2\right| \leq 4\delta_{m_n} L_n'.$$

Moreover using Theorem 3 we know that there is $C > 0$ such that:

$$\mathbb{E}\left(\max_{p \leq p_n} \left| \frac{1}{m_n - 1} \sum_{i \leq m_n} \big(S_{i,p} - \bar{S}_p\big)^2 - (\Sigma^{*,n})_{p,p}^2 \right|\right)$$

$$\leq \mathbb{E}\left(\max_{p \leq p_n} \left| \frac{1}{m_n} \sum_{i \leq m_n} \left( S_{i,p}^2 - \frac{1}{m_n - 1} \sum_{j \neq i} S_{i,p} S_{j,p} \right) - (\Sigma^{*,n})_{p,p}^2 \right|\right)$$

$$\overset{(a)}{\leq} \frac{C \log(p_n)}{\sqrt{m_n}} \big[3(L_n')^2 + 2L_n' \epsilon_{m_n}\big] + 4C \log(p_n) \sqrt{m_n} L_n' \epsilon_{m_n}$$

where (a) uses Theorem 3. Therefore this implies that for all $\epsilon > 0$

$$P\left(\max_{p \leq p_n} \left|(\hat{\Sigma} - \Sigma)Z\right|_p \geq \epsilon\right) \to 0.$$

This directly implies that

$$\sup_t \left| P\left(\max_{p \leq p_n} |(\Sigma Z)_p| \geq t\right) - P\left(\max_{p \leq p_n} \left|(\hat{\Sigma}Z)_p\right| \geq t\right) \right| \to 0.$$

Therefore we have

$$P\left[\sqrt{n} \max_k \left|\bar{Y}_k^* - \mathbb{E}(Y_k^*)\right| \geq \hat{t}_{g,n}^{\alpha/2}\right] \leq \alpha/2 + o_n(1).$$

Finally by exploiting the smoothness properties of $\mathcal{L}_n$, for every $\mu \in \mathbb{R}$ we have

$$\mathbb{E}\Big(\Big|\mathbb{E}\big(F_{2,n}(Y^*)\big|Y^{n,1}, X^n\big) - \mathbb{E}\big(F_{2,n}(Y^* + \frac{\mu}{\sqrt{n - m_n}})|Y^{n,1}, X^n\big)\Big|\Big)$$

$$\leq \sum_{j \leq d_n'} \mathbb{E}\Big(\Big| \sum_{p \leq p_n} \mu_p \omega_p \sup_{\tilde{\mu} \in [0,\mu]} \mathbb{E}\Big( \sum_{i \leq n - m_n} \hat{R}_i^{n,j}(Y^{n,2} + \frac{\tilde{\mu}}{\sqrt{n}})\theta_{n,j}^{\bar{p}}(Y^{n,2} + \frac{\tilde{\mu}}{\sqrt{n}}, Y_i^{1,n})\Big|X^n, Y^{n,1}\Big)\Big|\Big)$$

$$\leq \sum_{j \leq d_n'} \sup_{\tilde{\mu} \in [0,\mu]} \|\sum_{i \leq n - m_n} \hat{R}_i^{n,j}(Y^{n,2} + \frac{\tilde{\mu}}{\sqrt{n}})\|_{L_1} \sum_{p \leq p_n} |\mu_p| \omega_p \sup_{\tilde{\mu} \in [0,\mu]} \|\theta_{n,j}^{\bar{p}}(Y^{n,2} + \frac{\tilde{\mu}}{\sqrt{n}}, Y_i^{1,n})\|_{L_\infty}$$

$$\leq 2T_n L_n^* d_n' \sup_p |\mu_p|.$$

Therefore this implies that

$$P\left(\left|\mathbb{E}\big(\mathcal{R}_{\hat{\Theta}_n'}^{\mathrm{s}}(Y_{1:n-m_n}^{n,1}) \mid X^n\big) - \mathbb{E}\big(\mathcal{R}_{\hat{\Theta}_n^{Y^{n,2}}}^{\mathrm{s}}(Y_{1:n-m_n}^{n,1}) \mid X^n\big)\right| \geq 2T_n L_n^* d_n' \hat{t}_{g,n}^{\beta/2}\right)$$
$$\leq \alpha/2 + o_n(1)$$

We obtain in conclusion that for all $\delta > 0$ we have:

$$P\left(\mathbb{E}\big(\mathcal{R}_{\hat{\Theta}_n^{Y^{n,2}}}^{\mathrm{s}}(Y_{1:n-m_n}^{n,1}) \mid X^n\big) \in \mathbb{E}\big(\mathcal{R}_{\hat{\Theta}_n^{Z^{n,2}}}^{\mathrm{s}}(Z_{1:n-m_n}^{n,1}) \mid Z^{n,1}, X^n\big) \pm (t_{b,n}^{\alpha/2}(X^n) + 2T_n L_n^* d_n' \hat{t}_{g,n}^{\alpha/2} + \delta)\right)$$
$$\geq 1 - \alpha + o_n(1).$$

$\square$

# T    Proofs from Appendix A

## T.1    Proof of Theorem 3

**Proof:**    The first step is to bound $\mathbb{E}\Big[\max_{k \leq p_n} \frac{1}{\sqrt{n}} \sum_{i \leq n} \tilde{X}^n_{i,k}\Big]$. Define $(f_n : \mathbb{R}^d \to \mathbb{R})$ to be the following sequence of functions:

$$f_n(x_{1:n}) := \frac{\log\left(\sum_{l \leq p_n} e^{\log(p_n)\frac{1}{\sqrt{n}}\sum_{i \leq n} x_{i,l}}\right)}{\log(p_n)}.$$

We observe that $\|f_n(\tilde{X}^n) - \max_{k \leq p_n} \frac{1}{\sqrt{n}} \sum_{i \leq n} \tilde{X}^n_{i,k}\|_{L_1} \leq 1$; and that the functions $(f_n)$ are infinitely differentiable. We denote, for all $z \in \mathbb{R}$, $(\tilde{X}^{i,n}_l)$ and $(\tilde{X}^{i,z}_l)$ the processes respecting

$$\tilde{X}^{i,n}_l := \begin{cases} \tilde{X}^n_l \text{ if } i \leq l \\ 0 \text{ otherwise.} \end{cases} \qquad \tilde{X}^{i,z}_l := \begin{cases} \tilde{X}^{i,n}_l \text{ if } i \neq l \\ z \text{ otherwise} \end{cases}.$$

We first suppose that $\|\max_{k \leq p_n} \tilde{X}^n_{1,k}\|_{L_p} \leq 1$. Using the Taylor expansion we have:

$$
\begin{aligned}
\left|\mathbb{E}\Big[f_n(\tilde{X}^n)\Big]\right| &\leq \sum_{i \leq n} \left|\mathbb{E}\Big[f_n(\tilde{X}^{i,n}_{1:n}) - f_n(\tilde{X}^{i-1,n}_{1:n})\Big]\right| \\
&\leq \sum_{i \leq n} \left|\mathbb{E}\Big[(\tilde{X}^n_i)^\top \partial_i f_n(\tilde{X}^{i,0}_{1:n})\Big]\right| + \sum_{i \leq n} \left|\mathbb{E}\Big[(\tilde{X}^n_i)^\top \partial^2_i f_n(\tilde{X}^{i,0}_{1:n}) \tilde{X}^n_i\Big]\right| \\
&\quad + \sum_{i \leq n} \mathbb{E}\Big[\sup_{z \in [0,\tilde{X}^n_i]} \Big\langle \big|\partial^3_i f_n(\tilde{X}^{i,z}_{1:n})\big|, \big|\tilde{X}^{\otimes 3}_i\big|\Big\rangle\Big] \\
&\overset{(a)}{\leq} n\Big| \sum_{d_1,d_2 \leq p_n} \mathbb{E}\Big[\partial^2_{i,d_1,d_2} F^n_\beta(\tilde{X}^{i,0}_{1:n})\Big] \mathbb{E}(\tilde{X}^n_{1,d_1} \tilde{X}^n_{1,d_2})\Big| \\
&\quad + n \sum_{d_1,d_2,d_2 \leq n} \mathbb{E}\Big[\sup_{z \in [0,\tilde{X}^n_i]} \big|\partial^3_{i,d_{1:3}} F^n_\beta(\tilde{X}^{i,z}_{1:n})\big| \big|\tilde{X}^n_{1,d_1} \tilde{X}^n_{1,d_2} \tilde{X}^n_{1,d_3}\big|\Big]
\end{aligned}
$$

(27)

where to get (a) we used the fact that as $\sigma(\tilde{X}^{i,0}_{1:n}) = \sigma(\tilde{X}^n_0, \ldots, \tilde{X}^n_{i-1})$ we have $\mathbb{E}(\tilde{X}^n_i | \tilde{X}^{i,0}_{1:n}) = 0$. The next step is to upper-hand the right-hand side of (27). For ease of notations we write:

$$\omega_k(x_{1:n}) := \frac{e^{\log(p_n)\frac{1}{\sqrt{n}}\sum_{i \leq n} x_{i,k}}}{\sum_{l \leq p_n} e^{\log(p_n)\frac{1}{\sqrt{n}}\sum_{i \leq n} x_{i,l}}}.$$

For all $k_1, k_2 \leq p_n$ we obtain by the chain rule that

$$\left|\partial^2_{i,k_1,k_2} f_n(x_{1:n})\right| \leq \begin{cases} \frac{\log(p_n)}{n} \omega_{k_1}(x_{1:n})\omega_{k_2}(x_{1:n}) & \text{if } k_1 \neq k_2 \\ \frac{\log(p_n)}{n}\Big[\omega^2_{k_1}(x_{1:n}) + \omega_{k_1}(x_{1:n})\Big] & \text{if } k_1 = k_2. \end{cases}$$

As $\sum_{k \leq p_n} \omega_k(x_{1:n}) = 1$, this implies that

$$n\Big| \sum_{d_1,d_2 \leq p_n} \mathbb{E}\Big[\partial^2_{i,d_1,d_2} F_\beta(\tilde{X}^{i,0})\Big] \mathbb{E}(\tilde{X}^n_{1,d_1} \tilde{X}^n_{1,d_2})\Big| \leq 2\log(p_n).$$

We now bound the second term of the right hand side of (4). For all integers $k_1, k_2, k_3 \leq p_n$, by the chain rule, we obtain that:

$$
\begin{aligned}
&\left|\partial^3_{i,k_{1:3}} F_\beta(x'_{1:n})\right| \\
&\leq \frac{\log(p_n)^{3/2}}{n^{3/2}}\omega_{k_1}(x_{1:n})\omega_{k_2}(x_{1:n})\omega_{k_3}(x_{1:n})\left(1 + \mathbb{I}(k_1 = k_3) + \mathbb{I}(k_2 = k_3)\right) \\
&\quad + \frac{\log(p_n)^{3/2}}{n^{3/2}}\omega_{k_1}(x_{1:n})\omega_{k_2}(x_{1:n})\left(\mathbb{I}(k_1 = k_3) + \mathbb{I}(k_2 = k_3)\right) \\
&\quad + \frac{\log(p_n)^{3/2}}{n^{3/2}}\omega_{k_1}(x_{1:n})\left(\omega_{k_3}(x_{1:n})\mathbb{I}(k_1 = k_2) + \mathbb{I}(k_1 = k_2 = k_3)\right)
\end{aligned}
$$

Therefore we obtain that

$$n \sum_{d_1,d_2,d_3 \leq n} \mathbb{E} \left( \sup_{z \in [0, \tilde{X}_i^n]} \left| \partial_{i,d_{1:3}}^3 F^{\beta,n}(\tilde{X}^{i,z}) \right| \left| \tilde{X}_{1,d_1}^n \tilde{X}_{1,d_2}^n \tilde{X}_{1,d_3}^n \right| \right) \leq \frac{7\beta^2 \log(p_n)^2}{\sqrt{n}}.$$

Hence using (27) we establish that:

$$\left| \mathbb{E} \left( \max_{k \leq p_n} \frac{1}{\sqrt{n}} \sum_{i \leq n} \tilde{X}_{i,k}^n \right) \right| \leq 1 + \left[ \log(p_n) + \frac{7 \log(p_n)^2}{6\sqrt{n}} \right].$$

By potentially renormalizing $\| \max_{k \leq p_n} \left| \tilde{X}_{i,k}^n \right| \|_{L_p}$ we obtain therefore than in general we have:

$$\left| \mathbb{E} \left( \max_{k \leq p_n} \frac{1}{\sqrt{n}} \sum_{i \leq n} \tilde{X}_{i,k}^n \right) \right|$$

$$\leq \left\| \max_{k \leq p_n} |X_{1,k}| \right\|_{L_p} \left( 1 + \log(p_n) + \frac{6 \log(p_n)^2}{\sqrt{n}} \right).$$

Lastly, according to the Rosenthal inequality for martingales [35], there are constants $(C_p)$ that do not depend on $d$ or $(\tilde{X}_i^n)$ such that

$$\left\| \max_{k \leq p_n} \frac{1}{\sqrt{n}} \sum_{i \leq n} \tilde{X}_{i,k}^n - \mathbb{E} \left( \max_{k \leq p_n} \frac{1}{\sqrt{n}} \sum_{i \leq n} \tilde{X}_{i,k}^n \right) \right\|_{L_p} \leq 2C_p \left\| \sup_{d_1 \leq p_n} \left| \tilde{X}_{i,d_1}^n \right| \right\|_{L_p}.$$

Therefore we get the desired results using the triangular inequality.

Finally, let $(\mathbb{F}_i)$ designates the filtration $\mathbb{F}_i := \sigma \left( X_1^n, \ldots, X_i^n \right)$ then the following is an array of martingale differences $(\mathbb{E}(g_{k,n}(X^n)|\mathbb{F}_i) - \mathbb{E}(g_{k,n}(X^n)|\mathbb{F}_{i-1}))$. The last point of Theorem 3 follows directly from this observation. $\qquad \square$