# OpenReview forum: "Asymptotics of the Bootstrap via Stability with Applications to Inference with Model Selection"
_NeurIPS.cc/2021/Conference — NeurIPS 2021 Poster_

### Official Review · Reviewer_iK1w · 2021-07-10

**Rating:** 7
**Confidence:** 3

**Summary:**

---------------------
---------------------
**Update after discussion with authors and other reviewers**

Thank you to the authors and other reviewers for engaging so much in the discussion period. I think the discussion has made clear that the  authors could significantly strengthen the paper by drawing connections between their proofs and others in the literature. I also think there should also be a discussion on the limitations of the example in Section 4, as the authors are not analyzing a standard kernel test (although, as they note, they are analyzing a statistic that is highly related to a common kernel test). I think this makes the results of Section 4 a little weaker. This, combined with XTTq's points about the relation to the existing literature, have led me to decrease my score somewhat. I'm still in favor of an accept because while the paper could be improved by restructuring, I think it still presents practically useful theory as-is. If the paper is accepted, I think the authors should at the very least include some discussion surrounding their discussions with XTTq.

---------------------
---------------------


This is a theoretical paper which aims to extend the reach of consistency results for the classical bootstrap. The authors prove some general theory about the bootstrap and then show that this theory can be specialized to prove consistency of the bootstrap for two sample kernel hypothesis tests and stacked ensemble estimators. To my knowledge, these results are novel in that previous bootstrap theory does not apply to them, and these results seem useful to the machine learning community. Besides solid results, I think this paper has a few nice strengths:

1.	The paper is well written
2.	The literature review is, to my knowledge, very thorough (which is not easy to do for a topic with as much background as the bootstrap).
3.	The authors are careful to call out the flaws of their theory throughout the paper.
4.	The authors try to explore the necessity of their assumptions with the use of counterexamples

A downside is that all of this takes a *lot* of room – the appendix is 45 pages of additional results, examples, counterexamples, and proofs. I’ve tried to go through this as carefully as possible, but I have not been able to carefully check all the proofs. Overall, I think this is a solid paper, and my criticisms listed below are not major.


**Limitations And Societal Impact:**

The authors do not discuss societal impact in this paper, but, given its subject, I think that is appropriate.

**Main Review:**

I have three larger comments that I think could improve the paper, listed as (A), (B) (C) below:

(A) I think it would be good to include some more discussion about what kinds of models in machine learning the paper does / does not extend the reach of bootstrap consistency proofs to. The authors give some generic theory and then show it applies to two specific examples (kernel hypothesis testing and stacked ensembled estimators). Is there some underlying theme between the proofs for these two examples? Can the authors conjecture about what kinds of other examples their proof techniques might work well for? Conversely, it seems that the authors do not cover *all* problems of interest in machine learning. My understanding is that for simple models like Poisson regression with covariates growing unbounded with n (that is, a GLM with unbounded link function with unbounded derivatives), existing common proof techniques for the consistency of the bootstrap (like linearization) are not known to apply. And it seems like the authors’ Assumption 1 would be violated in this case as well, so their theory will not apply to this case either. I don’t mean this as a criticism -- the authors definitely don’t try to sell their work as applying to everything. But it would be good to call out what’s still missing for the sake of future research.

(B) The authors put a lot of discussion up front about their use of particular proof techniques backing their new results (“Chaterjee’s generalized Lindenberg approach and a smart path interpolation technique”). This sounds interesting, and I was hoping to learn more in the main text! I wonder if the authors could summarize how these techniques fuel their results in the main text.

(C) I wonder if section 5, which is about bootstrapping ensembled estimators, could be reworked to include more discussion and fewer mathematical details, e.g. by making Prop 3 an informal statement and pointing to the appendix for a formal statement.  I say this because I think there are a few points in Section 5 that could use expanding on:
1. The authors analyze the bootstrapping properties of what seems like a non-standard ensemble estimator. In particular, they consider the ensemble of estimators indexed by 1,…,p, where the ith estimator is fit on datapoints x_1, …, x_i. This seems like a very particular estimator. To be fair, the authors do note “[this specific ensemble] allows us to show that the distribution of the ensemble’s risk can be estimated with the bootstrap”. But it would be great to understand what properties of this estimator are required for the authors’ analysis vs required for the bootstrap to converge (i.e. did the authors choose a somewhat non-standard ensemble because the bootstrap doesn’t converge otherwise or because their analysis won’t go through otherwise?).
2. What is \beta_n doing here? Is it a hyperparameter again? It seems like \beta_n is required to go to infinity, which in turn seems like it causes the ensembled estimator \hat\Theta_n to collapse to the best-performing \hat\theta^k_n. Is this correct? If so, why is this collapse a desirable thing?
3. Are the base estimators (the \hat\theta^k_n) not required to satisfy Assumption 1? This seemed like a pretty important part of previous theory, so how is it that we are stacking Assumption 1-violating estimators to get a bootstrap-consistent estimator?
4. Relatedly, it would be nice to have a conclusion at the very end summarizing the main results of the paper.

-------- Scattered smaller comments:
Line 263: what is d_n’?

Line 156: Notation for Lindenberg path interpolation is defined, but I don’t think this was used in the main text.

Line 167: Influence functions are mentioned here but not defined.

The title of Appendix N is “Further proofs from section 3”, but not all of the results discussed here seem to be from Section 3 (e.g. there is no Example 2 in Section 3)


**Time Spent Reviewing:**

8

---

> ### Author Response · Authors · 2021-08-10
> **Answer of reviewer iK1w**
>
> **Point A**
>
> *“Is there some underlying theme between the proofs for these two examples? Can the authors conjecture about what kinds of other examples their proof techniques might work well for?”*
>
> Yes, we can conjecture which examples will work and which ones will not. Heuristically for our theorems to apply we need that the importance of each random variable to not be too big in the sense that   $ \|g_n(X_1,\dots,X_n)-g_n(0,X_2,\dots,X_n)\|_{L_3}=o(n^{-⅓}) $ and we also need to for that difference to have a ``nice form” in the sense that it is approximable by $ h(X_1) g(X_2,\dots,X_n)$ where h and g are two measurable functions.
>
> **Point B:**
>
>
> We are happy to add more details about Chaterjee’s method and how we adapt it in the supplementary material section
>
> **Point C**
>
> *“In particular, they consider the ensemble of estimators indexed by $1,…,p$, where the ith estimator is fit on datapoints $x_1, …, x_i. $“*
>
> The ith estimator is built using the first m_n data points. We split the data set into two: the first part of the dataset is used to train the base estimators and the second part of the data is used to compute weights.
>
> *“What is \beta_n doing here? Is it a hyperparameter again? It seems like \beta_n is required to go to infinity, which in turn seems like it causes the ensembled estimator $\hat\Theta_n $ to collapse to the best-performing $\hat\theta^k_n$. Is this correct? If so, why is this collapse a desirable thing?”*
>
> $\beta_n$ is a hyperparameter that controls how concentrated around the estimator(s)  with the lowest empirical error our stacked estimator will be. But we do not require \beta_n to grow to infinity. Our theorem works if \beta_n grows but also works if  $\beta_n$ does not grow. The reason one might want \beta_n to grow to infinity is that one might hope the estimators that have the lowest empirical risk might also be the ones with the lowest expected loss (and so ``the best estimators”)
>
> *"Are the base estimators (the $\hat\theta^k_n$) not required to satisfy Assumption 1? This seemed like a pretty important part of the previous theory, so how is it that we are stacking Assumption 1-violating estimators to get a bootstrap-consistent estimator?"*
>
> No, they are not required to satisfy Assumption 1. Indeed, we want to build confidence intervals for the risk of the stacked  estimator. Therefore we only bootstrap the observations m_n+1 until n, which are the observations used to compute the empirical risk and the weights. This implies that we are not bootstrapping the basic estimators  \hat \theta_1,..., \hat \theta_p and that we do not need them to satisfy assumption 1. If one wanted to give a confidence interval for the stacked estimator (and not for its risk) then we would need to impose more conditions on the basic estimators.

---

### Official Review · Reviewer_iG7z · 2021-07-13

**Rating:** 7
**Confidence:** 1

**Summary:**

The authors derive general stability conditions under which the empirical bootstrap estimator is consistent and quantify the speed of convergence. These results are illustrated for two-sample kernel tests after kernel selection and the empirical risk of stacked estimators. These two applications are relevant due to the recent interest in kernel tests with learnt deep kernels and the popularity of ensemble models in practice.

**Ethical Concerns:**

This work does not reaise any ethical concerns.

**Limitations And Societal Impact:**

Limitations have not been fully explained (or maybe I missed them).
This contribution does not have potential negative societal impact.

**Main Review:**

The paper is highly technical and theoretical, and it still reads quite well. Although I am not very familiar with the statistical approaches and therefore not qualified to comment on its soundness, I am convinced that its originality and contribution is definitely very strong.
As mentioned above, the application of the general theorem to kernel tests with learnt kernels and model ensembles is very relevant to current machine learning challenges. While I am keen to see numerical results corroborating the theoretical ones, they are not a must for this paper, rather for a follow-up.
Finally, I would have appreciate a concluding section highlighting the limitations of the current approach and a recommendation on where to focus future research.

Other comments:
- Please use a spell-checker to review the paper as there are few typos here and there.
- Lines 82-84: p, p_n and n are used before being defined.
- The two equations after lines 123 and 127 are identical: should they?
- Page 4, please define C^{\alpha, n} and C^{\alpha, n}_\epsilon
- Page 8, Proposition 2: why is the result obtain using the L1-norm instead of just d_F \to 0 in probability, as d_F is a real number anyway?

**Time Spent Reviewing:**

2

---

> ### Author Response · Authors · 2021-08-10
> **Answer to reviewer iG7z**
>
> Thank you for your review and for your suggestions. We will fix the typos.
>
> *“ Finally, I would have appreciated a concluding section highlighting the limitations of the current approach and a recommendation on where to focus future research”*
>
> Thank you for your suggestion. We will incorporate your suggestion for the camera-ready version

---

### Official Review · Reviewer_FFbw · 2021-07-16

**Rating:** 8
**Confidence:** 3

**Summary:**

The paper discusses very general conditions on estimators under which the empirical bootstrap is asymptotically consistent. Explicit applications to two difficult inference problems, including two-sample tests using kernels with (approximate) kernel selection and on estimating the risk of ensemble learners.

**Limitations And Societal Impact:**

Limitations are not explicitly discussed. There is no apparent potential negative societal impact.

**Main Review:**

The conditions discussed in the paper under which consistency can be achieved are impressively general, and it is clearly advantageous to consider general conditions rather than specific problems as is common. The paper is well written, and for the most part clear, although there are places in which the notation becomes slightly challenging (see later).

I do not have any significant reservations about the paper, and my main comments are editorial suggestions:
- I recommend being explicit in the abstract that the examples covered are slightly modified; especially in the case of the kernel two-sample test, where a softmax variant is substantially different from a discrete selection, which is what (at least I) expected.
- I would recommend mentioning earlier on that the adjustment for the "centering" can actually be easily handled in practice. As I read through the first part of Section 3 I was bugged by the persistent presence of the centered processes, and it was only patently clear how this is handled when I got to line 214. This only bears mentioning earlier; it is not necessary that the details be given, but the ubiquity of the basic CLT as being the way around it will make less expert readers feel more at ease.
- Am I missing something in the final statement in Theorem 2? Should it not be P(E[g_n(Y^n)] \not \in [interval]) \leq \beta?
- The notation in Section 4 is challenging, since now the second subscript refers to the identity of the process and not the index of the component in the corresponding observation. While when the permutation test starts to be discussed, if I understand correctly, this becomes a little clearer since then the two processes (samples) are just stacked next to one another as one array of higher dimensional processes, but I think it is worth mentioning earlier on. Otherwise, the expression above line 224 is confusing if you interpret X_{1,1}^n as the first element of the first observation in X^n, and not as the entire first observation in (X^n_{i,1}).

To conclude, I have some minor comments/queries/typos, etc.
- The word "each" in line 17 seems out of place
- line 77: space between [14] and .
- line 83: Recommend either "A series of works has strengthened..." or "A series of work has strengthened..."
- lines 84 and 86: should not the dimension, p, explicitly depend on n as previously?
- line 86: "...conditions THAT log(..."
- line 94: "...specific STATISTIC or..."
- Although not ambiguous, the metric d_{\mathcal{F}} is defined between probability measures, but is evaluated on random variables.
- line 124: "same mean AS X^{n}_1
- Aren't lines 126-127 just repeating what lies above in line 123?
- In Proposition 1, recommend "Let \epsilon > 0 be constant, then..."
- line 164: missing subscript n in g(0X^n_{2:n})
- Although not ambiguous, I don't think the L_{p} spaces of RVs are not explicitly defined; only the norm is.
- this is purely cosmetic, but in the statement of assumption H_3, the parentheses in g(Y^n + x/sqrt{n}) would look better if larger
- line 224: should not "opposite" better be replaced with "converse"?
- What does (a) above equality mean? Should this be d?
-  What is the metric d_W?
- The font in Proposition 2 is inconsistent
- Why is sometimes max()/min() and other times join/meet used?

**Time Spent Reviewing:**

8

---

> ### Author Response · Authors · 2021-08-10
> **Answers to the comments from reviewer FFbw**
>
> **Editorial suggestions (point 1 and 2):**
>
> Thank you for the helpful suggestion. We will incorporate your suggestion in the camera ready version
>
> **Point 3**
>
> *“Am I missing something in the final statement in Theorem 2? Should it not be P(E[g_n(Y^n)] \not \in [interval]) \leq \beta?”*
>
> Sorry, this is a typo. Thank you for catching it. It will be fixed
>
> **Point 4**
>
> Thank you for noting this. We will clarify the notations for the camera ready version

---

### Official Review · Reviewer_XTTq · 2021-07-18

**Rating:** 3
**Confidence:** 4

**Summary:**

The authors put forward a set of high-level conditions that are sufficient for bootstrap consistency of nonlinear statistics and for constructing bootstrap confidence intervals that are approximately asymptotically correct. They illustrate their framework with two examples: a) a two-sample test to test whether two distributions are identical, and b) a procedure to construct confidence intervals for the risk of smooth stacked ensemble estimators.

**Limitations And Societal Impact:**

Somewhat. The authors provide several examples and counter-examples in the supplementary.

**Main Review:**

This paper is well-written and clearly structured. Obviously, the authors have put a lot effort in this paper. However, I am always wary of papers that aim to develop a very general theory. Often, the level of generality of such papers leads to weak implications. Sometimes such papers also impose (hidden) assumptions, that limit the range of applicability of the supposedly general results.

1) I believe that it is long clear that stability conditions such as Assumption 1 are part of the sufficient conditions for bootstrap consistency/ consistency of bootstrapped CIs. The difficulty lies in verifying these conditions for specific statistics and in deriving anti-concentration inequalities for specific statistics that are statistically meaningful. The examples provided in the supplementary materials (such as the square or p's power of the sample average) are not meaningful statistics. The two applications in the main text are essentially smoothed versions of a maximum/ supremum statistic and, therefore, it is easy to find equivalent maximum-statistics which could be analyzed via the approach developed Chetverikov, Chernozhukov and Kato in their sequence of papers from 2014-2020. What statistics other than statistics related to the maximum satisfy the stability properties?

2) I credit the authors with pointing out that anti-concentration inequalities are not needed if one is willing to settle for a weaker notion of CIs.

3) In the main text of the paper the authors write that their work relies on (or extends?) Chatterjee's (2006) generalized Lindeberg principle. I know that Chatterjee's results only apply to random vectors with i.i.d. or exchangeable entries. I therefore wonder how the authors modify Chatterjee's arguments to allow for arbitrary correlation between the entries of the random vectors. The proof of Theorem 1 appears to be incomplete as it references a Theorem 19 which is not in supplementary materials. Is the missing Theorem 19 is a version of Chatterjee's results for "triangular arrays of i.i.d. processes"? I could not locate an analogue of either Theorem 1.1 or 1.2 of Chatterjee (2006) in the supplementary materials.

4) I've never seen the notion of bootstrap consistency that the authors use (lines 113 - 118). It is weaker than convergence in Wasserstein distance and it does not seem to imply consistency in Levy-Prokhorov distance. Does it imply consistency in Kolmogorov distance? Since the authors' metric does not seem to metrize weak convergence I do not think that the bootstrap consistency result in Proposition 2 is particularly useful. It doesn't seem to imply that we can compute p-values needed for hypothesis tests.

5) The second main result of the paper (Theorem 2) holds only for nonlinear statistics which are also Lipschitz continuous with respect to the $\ell_\\infty$-norm (Assumption $H_3$). One can easily infer from the statement of the theorem that the larger the Lipschitz constant, the larger the proposed CI. This is problematic since the Lipschitz constant might grow with the dimension. Here are some not fully rigorous calculations to illustrate this point: For simplicity assume that X_i's are i.i.d. vectors with i.i.d. entries. Take $g(X_1, \\ldots X_n) = \\|n^{-1}\\sum_i X_i\\|_2$. Then $g$ is Lipschtiz continuous w.r.t. to $\\ell_\\infty$-norm and has Lipschitz constant $d_n^{1/2}$. Hence, by Theorem 2, as the sample size $n$ (and the dimension $d_n$) grow the diameter of the CI for $\\mathbb{E}\\|n^{-1}\\sum_i X_i\\|_2$ grows roughly at a rate of $d_n^{1/2}$ (b/c $t_\{g,n\}$ has to grow to cancel out the Lipschitz constant otherwise the tail bound on the maximum of the Gaussian random variable $N^n$ doesn't hold). Intuitively, this CI is too wide since we know the variance of $\\|n^{-1/2}\\sum_i X_i\\|_2$ is roughly constant (b/c $n^{-1/2}\\sum_i X_i$ is approximately Gaussian and the variance of the Euclidean norm of a centered Gaussian vector with identity covariance is dimension free).

**Time Spent Reviewing:**

5

---

> ### Author Response · Authors · 2021-08-10
> **Answer to the comments of reviewer XTTq**
>
> We would like to thank the reviewer for his time. Find below our answers to the reviewer's concerns:
>
>
> **Point 1:**
>
> *“I believe that it is long clear that stability conditions such as Assumption 1 are part of the sufficient conditions for bootstrap consistency”*
>
> To the best knowledge of the authors there is no previous  paper establishing those. It might be a prevalent heuristic. Many previous works established the consistency of the bootstrap method for specific statistics; however no previous paper established general and easily verifiable  conditions for the consistency of the bootstrap method. Our conditions, involving stability conditions, are easier to check for many machine learning applications.
>
>  *“The two applications in the main text are essentially smoothed versions of a maximum/ supremum statistic “*
>
> This is not true.  They are not soft approximations of a maximum or supremum. Instead, they are functions of a smooth argmax. Functions of that type are very common in machine learning, as estimators are often chosen to minimize a cost function (e.g they often are argmins), and are of great interest for machine learning applications.
>
> *“ and, therefore, it is easy to find equivalent maximum-statistics which could be analyzed via the approach developed Chetverikov, Chernozhukov and Kato in their sequence of papers from 2014-2020.”*
>
> Our proof and theorem also apply for the max of centered processes, that were analyzed by CCK. However, CCK doesn’t provide general conditions for consistency and uses a different style of proof than ours. Closest to our work on the max is the more recent work of Deng & Zhang (2020)  who also uses a Lindenberg style proof and our general theorem recovers the theorem in that work as a special case. So it is true that we can recover results from CCK as a special case of our general framework. However, the two examples we present in the main text do not fall into the setting of the max of centered processes.
>
> *" What statistics other than statistics related to the maximum satisfy the stability properties?"*
>
>  Once again, the examples are not smooth maximums but instead a function of the argmax (which is natural in ML). Moreover, our theorem can be used to study the consistency of the bootstrap method for other types of statistics. An example is the K-fold cross-validated risk for which deriving a confidence interval is of great interest. Our theorems apply to that context even when the limiting distribution of the K-fold cross-validated risk is unknown. We are happy to add this to the supplementary material for the camera-ready version. Another example is two step M-estimators.
>
> **Point 3**
>
> *“Is the missing Theorem 19 is a version of Chatterjee's results for "triangular arrays of i.i.d. processes"?*
>
> Sorry about this. Theorem 19 is lemma 19. It is a latex bug that will be fixed. Thank you for pointing it out. Lemma 19 is an intermediary result. We don’t exactly use Chatterjee’s method, we adapt it/expand it slightly. Indeed for a test function h: instead of doing a Taylor’s expansion of $ h\circ g_n$ around each observation $X_i$, we do a Taylor expansion on $h$ around $g_n(X_0,\dots,X_n)$ and $g_n(X_0,\dots, X_{i-1},0,X_{i+1},\dots,X_n)$. Lemma 19 is a result that allows us to adapt Chaterjee’s argument in that way.
> “Therefore I wonder how the authors modify Chatterjee's arguments to allow for arbitrary correlation between the entries of the random vectors. “
> We look at the impact of each vector X_i as a whole and not just at the impact of each entry. The correlation between the entries still plays a role: the weaker the correlations are the weaker the conditions are and notably the faster the number of entries can grow.
>
>  **Point 4**
>
> *“Since the authors' metric does not seem to metrize weak convergence I do not think that the bootstrap consistency result in Proposition 2 is particularly useful. It doesn't seem to imply that we can compute p-values needed for hypothesis tests.”*
>
> This is not true. It does metrize the weak convergence: any indicator function can be approximated by a function in $\mathcal{F}$. Therefore if $(X_n)$ and $(Y_n)$ are sequences of real valued random variables and if $d_{\mathcal{F}}(X_n,Y_n)$ then $X_n$ converges in distribution to $Y_n$.
>  Proposition 1 shows this and establishes the relation of d_{\mathcal{F}} to the Kolmogorov metric and confidence intervals.
>
> **Point 5**
>
> * “The second main result of the paper (Theorem 2) holds only for nonlinear statistics which are also Lipschitz continuous with respect to the ℓ∞-norm (Assumption H3). “*
>
> This is not true. We assume that the expectation is Lipchitz which is very different from assuming that the statistics itself is Lipchitz and a much weaker condition. It is notably the case if the estimator is unbiased.
>
>  One can easily infer from the statement of the theorem that the larger the Lipschitz constant, the larger the proposed CI.
> True this is a weakness, but we propose an alternative method in the appendix see appendix H.

---

> > ### Comment · Reviewer_XTTq · 2021-08-18
> > **Complete Rebuttal of the Authors' Answers**
> >
> > Reg 1: My criticism is that your Assumption 1.2 is very high level and needs to be verified for every specific statistic. Until now, such stability assumptions have only been shown to hold for statistics that can be approximated by the smooth-max function (Chernozhukov et al. (2014-2020), Koike (2019), Fang, Koike (2020), Deng, Zhang (2020),...), i.e. the statistics for which this assumption holds are essentially maximum-statistics (maximum, argmax, soft-max, smooth-argmax,...) Your examples in Sections 4 and 5 are clearly (related to) maximum statistics. There is really no difference between $\max\\{Y_1, \ldots, Y_N\\}$ and $\arg\max_{y \in \\{Y_1, \ldots, Y_N\\}} y$ even if the $Y_1, \ldots, Y_N$ are derived from the data $X_1, \ldots, X_n$, i.e. $Y_i = Y_i(X_1, \ldots, X_n)$ for all $1 \leq i \leq N$, since one can always work out everything conditionally on the data. Conceptually, your examples are not novel and I feel that you are overselling the level of generality of your results.
> >
> > Reg 3: Thank you for your clarification. Is there any way in which you could correct the Supplementary Material before the deadline and relay to the area chair so that he/ she can forward it to me to read it? I would love to take a closer look before I can make any final recommendation regarding this paper.
> >
> > Reg 4: [Note: I've updated this comment to reflect the authors response. They were right in pointing out that convergence w.r.t. to the (dual of the) Wasserstein distance is sufficient but not necessary for weak convergence.]
> >
> > The conventional metric that metrizes weak convergence of sequences of r.v. $(X_n)_{n \geq 1}$ to a limit r.v. $X$ is based on the class of bounded 1-Lipschitz continuous functions $\mathrm{BL}_1$, i.e. $X_n \leadsto X$
> > i̶f̶f̶  if
> >
> > $$\sup_{f \in \mathrm{BL}_1} |\mathrm{E}[f(X_n) - f(X)]| \rightarrow 0.$$
> >
> > Obviously, the class of thrice diff'able functions with bounded derivatives is a subset of $\mathrm{BL}_1$. Hence, convergence with respect to the distance defined in your paper i̶s̶ ̶w̶e̶a̶k̶e̶r̶ ̶t̶h̶a̶n̶ ̶w̶e̶a̶k̶ ̶c̶o̶n̶v̶e̶r̶g̶e̶n̶c̶e̶. may not immediately imply weak convergence in the same way as does convergence with respect to above distance based on functions $\mathrm{BL}_1$. T̶h̶e̶r̶e̶f̶o̶r̶e̶  It appears as if your metric $d_F$ does not metrize weak convergence, i.e. $d_F(X,Y| \mathcal{E}) = 0$ may not imply $X \overset{d}{=}Y$.
> >
> > It is correct that any function $f \in \mathrm{BL}_1(\mathrm{R})$ (the class of real-valued bounded 1-Lipschitz cont. functions) can be smoothly approximated by thrice diff'able functions: Just take the convolution $f_\epsilon : = f \star u_\epsilon \rightarrow f$ as $\epsilon \rightarrow 0$, where $u_\epsilon(x) = \epsilon^{-1} u(x/\epsilon)$ with $u \in C^3_b(\mathrm{R})$. However, the derivatives of $f_\epsilon$ satisfy $f'_\epsilon = O(1)$,  $f''_\epsilon = O(\epsilon^{-1})$ and $ f'''_\epsilon = O(\epsilon^{-2})$ and are thus unbounded as the approximation error vanishes. Therefore, the function class of thrice diff'able functions with bounded derivatives is not rich enough to recover weak convergence via convolutional smoothing even for real-valued functions. (In higher dimensions the rate at which the derivatives diverge will typically depend on the dimension and will be even worse.) In fact, I don't think that it is at all possible to smoothly approximate functions in $\mathrm{BL}_1$ with functions that have bounded derivatives. If you think otherwise, please provide an explicit construction; I'd love to learn something new.
> >
> > At the risk of repeating myself, more formally, we have
> >
> > $\\sup_{f \\in \\mathrm{BL}_1} | \\mathrm{E}[f(X_n) - f(X)]|$
> >
> > $\\leq \\sup_{f \\in \\mathrm{BL}_1}   |\\mathrm{E}[(f \\star u_\\epsilon) (X_n) - (f \\star u_\\epsilon)  (X)]|$
> >
> > $+ \\sup_{f \\in \\mathrm{BL}_1}   |\\mathrm{E}[(f \\star u_\\epsilon) (X_n) - f(X_n)]|$
> >
> > $+ \\sup_{f \\in \\mathrm{BL}_1}   |\\mathrm{E}[(f \\star u_\\epsilon) (X) - f(X)]|$
> >
> > $= A + B + C.$
> >
> > It is standard to show that $B, C \rightarrow 0$ as $\epsilon \rightarrow 0$. But as $\epsilon \rightarrow 0$, the derivatives of $f \star u_\\epsilon$ diverge and hence the term $A$ cannot be shown to be small by the tools that you have developed in your paper.
> >
> > In the response you mention indicator functions. That makes me wonder whether you were thinking of directly working with the Kolmogorov distance. Approximating indicator functions with thrice diff'able functions with bounded derivatives is again not possible if the approximation error is supposed to vanish. The problem is even more severe than in the case of Lipschitz continuous functions since indicator functions are not even in $\mathrm{BL}_1$. This is why already in the Gaussian approximation step (and not only in the bootstrap step) Chernozhukov et al. (2014-2020) need Gaussian anti-concentration inequalities. All these issues are the reason why Chernozhukov et al. (2014-2020) need to carefully choose the smoothing parameters (for smoothing the indicator function and the maximum) as functions of the dimension (or rather as functions of the expected size of the maximum-statistic). This is a crucial step and far from trivial. You entirely eschew this issue in your paper and as a result you obtain significantly weaker results.
> >
> > While your Proposition 1 is correct, it is not a statement about weak convergence and/ or the Kolmogorov distance. As I argued above, for weak convergence we need to let $\epsilon \rightarrow 0$. In this case, the inequalities in Proposition 1 are trivially true and void because their left-hand sides are always non-negative while their right-hand sides approach minus infinity (unless the $d_F(X,Y| \mathcal{E}) = 0$, which as argued above, still does not imply weak convergence).
> >
> > Therefore, the result of Proposition 2 does not imply (not alone and also not in combination with Proposition 1) that you can compute p-values for the test statistic. This contradicts your statement in lines 241-243. (Please correct me if I misinterpret what you write there.) At best your theory guarantees that one can construct a confidence interval with asymptotic coverage at least $1 - \alpha$. That's a relatively weak result. Similar comments apply to the implications of Proposition 3.
> >
> > Reg 5: Yes, I was sloppy. Thank you for pointing out that I omitted the expected value. However, with or without expected value, it doesn't change anything: This is an extra assumption that imposes restrictions on how non-linear the statistic can be and you do not provide examples beyond maximum type-statistics that satisfy this property. The main advantage of Lipschitz-continuity under expectation is that it allows one to deal with non-differentiable, discontinuous functions, e.g. $g_n(x) = \mathbf{1}\\{x < 0\\}$). However, non-differentiable, discontinuous functions cannot be approximated by thrice diff'able functions with bounded derivatives and thus violate Assumption 1.1 (note that the $L_1$-norm has absolute value inside the expected value and thus does not allow one to smooth non-differentiable, discontinuous functions). So, while the condition of Lipschitz-continuity under expectation is in principle quite interesting, you can't exploit this condition because all cases in which it could make a difference have to be excluded a priori from your analysis by Assumption 1.1.

---

> > > ### Author Response · Authors · 2021-08-19
> > > **Answer to reviewer XTTq**
> > >
> > > We would like to thank the reviewer for their response and for their time carefully reviewing our work. We provide further answers below to the points raised:
> > >
> > > 1) We note that indeed many of our examples are related to argmin and argmax, but the argmin and argmax that we deal with are more general than the form that the reviewer mentions in their response. In particular, we agree that $argmax_{y\in Y_1,\dots, Y_n} y$ and $\max Y_i$ are the same; but this is different from the type of argmin that we are studying which are more similar to $argmin_{\theta} f(\theta, X_1,\dots, X_n)$
> > > This is substantially very different. For example imagine that we observe $X_1,\dots,X_n$ and that our goal is to estimate $\hat \theta $ by minimizing an empirical loss over those training points: $\hat \theta:=argmin_{\theta} \sum_i \mathcal{L}(X_i, \theta).$ The distribution of  $\hat \theta$ and the distribution of the minimized loss can have nothing in common. For example, knowing that we obtain a zero training loss (for example for overparameterized models) tells you nothing about the distribution of the estimator itself. Another example is when the search space contains only two models that have the same risk. The minimized empirical loss will have the distribution of a maximum between two gaussians when the estimator itself will be a bernoulli random variable. Those distributions have little in common.
> > > Again we believe that estimators that come in the form of an argmin are natural in machine learning; but we also propose to add other examples such as the cross validated risk
> > >
> > > 2. *“Your metric does not metrize weak convergence, i.e. $d_{F}(X,Y)=0$  does not imply $X=d Y$. *
> > >
> > >   Even though our metric is weaker than the BL(1) metric, it still metricizes weak convergence. We are sorry if this was not clear in the current version of the paper. We will add more details for the camera ready version
> > > Here is our formal argument:
> > >
> > > Let $X$ and $Y$ be real valued random variables.  To show that $X\overset{d}{=}Y$, it is enough to show that $P(X\in [a,b])=P(Y\in [a,b])$ for all $a,b\in \mathbb{R}$. Now for all $\delta>0$, due to the monotone property of probability measures there is $\epsilon>0$ such that $$P\big(X\in (a-\epsilon,a)\big),P\big(X\in (b,b+\epsilon)\big)<\delta$$
> > >
> > >  Therefore $P\big(X\in (a-\epsilon, b+\epsilon)\big)\le P(X\in [a,b])+2\delta$. Moreover, using proposition 1 we know that $$P\big(X\in (a-\epsilon, b+\epsilon)\big)\ge P\big(Y\in [a,b]).$$ This implies that for all $\delta>0$ we have $$P(X\in [a,b])+2\delta\ge P(Y\in [a,b]).$$ As $\delta>0$ is arbitrary we have $P(X\in [a,b])\ge P(Y\in [a,b])$. By the exact same argument  we have $P(Y\in [a,b])\ge P(X\in [a,b])$ which implies that $$P(Y\in [a,b])=P(X\in [a,b]).$$
> > > QED.
> > >
> > > The reason, we do not need to make any anti concentration argument is that for any real valued random variable $X$ we have $\lim_{\epsilon \rightarrow 0} P\big(X\in (a-\epsilon,a)\big)=0$ and $lim_{\epsilon \rightarrow 0} P\big(X\in (a,a+\epsilon)\big)=0$. This is due to the monotone property of probability measures (see section 2.2 of “A course of probability” by Kai Lai Chung) that sets that if $A_i$ decreases to $A$ then $P(A_i)\rightarrow P(A)$. Indeed remark that  $(a-\epsilon,a)\overset{\epsilon\rightarrow 0}{\rightarrow} \emptyset$ therefore $P(X\in (a-\epsilon,a))\rightarrow 0$. This is true even if the measure is discrete. It is however interesting to note that as $(a-\epsilon,a+\epsilon)\searrow \{a\}$ then $lim_{\epsilon \rightarrow 0} P\big(X\in (a-\epsilon,a+\epsilon)\big)=P(X=a)$ which can be different from $0$. Things are more complex when dealing with sequences, see our answer below about those cases. We are sorry if our statement was not clear we will add more details and explanation in the camera ready version and happy to elaborate more on our argument in further responses, if the above proof seems inadequate or any points are inaccurate.
> > >
> > >
> > > *“While your Proposition 1 is correct, it is not a statement about weak convergence [...](unless the
> > > dF(X,Y|E)=0, which as argued above, still does not imply weak convergence)”*
> > >
> > >
> > > If $d_{\mathcal{F}}(X,Y)=0$ then $X\overset{d}{=}Y$. We present a formal argument above that we will add to the camera ready version as this point was not presented clearly. We apologize for this.
> > >
> > > It is true that our metric is weaker than Kolmogorov complexity. However, the metric $d_{\mathcal{F}}$ does metricize weak convergence:
> > >
> > > If $d_{\mathcal{F}}(X_i,Y)\rightarrow  0$ then $X_i$ converges in distribution to $Y$. We give a formal argument below that we will add it to the camera ready version. Let $(X_i)$ be a sequence of real valued random variables, to prove  that $(X_i)$ converges in distribution to $Y$ we only need to prove that for all $t$ at which the cdf $F_Y$ of $Y$ is continuous we have $\big|P(X_n\le t)-P(Y\le t)|\rightarrow 0$ (see theorem 4.3.1 of the textbook “A course in probability theory” by Kai Lai Chung). If $d_{\mathcal{F}}(X_n,Y)\rightarrow 0$, then using Proposition 1 we know that for all $\epsilon>0$ we have $\limsup_n P(X_n\le t)\le P(Y\le t+\epsilon)$ and $\liminf_n P(X_n\le t)\ge P(Y\le t-\epsilon)$. As $t$ is a point of continuity of $F_Y$ this directly implies that $\lim_n P(X_n\le t)=P(Y\le t)$ therefore we know that $(X_i)$ converges in distribution to $Y$.
> > >
> > > It is true that our metric is weaker than the one the reviewer suggests. However while convergence in distribution is implied by convergence in the BL(1) metric it is not necessary to have the BL(1) metric going to 0 to have convergence in distribution. This is notably the case if the random variables do not admit a $L_1$ moment. This is notably discussed in “An invitation to the Wasserstein Space” by V. Paranetos and Y. Zemel.  We present a formal argument below for why our metric still implies convergence in distribution. We are sorry for not having such formal claims in the current version and will add the formal argument to the camera ready version.
> > >
> > > *“All these issues are the reason why Chernozhukov et al. (2014-2020) need to carefully choose the smoothing parameters “*
> > >
> > > It is true that when dealing with triangular arrays $(X_n)$, $(Y_n)$, you might need to prove anti-concentration statements  to be able to state that $|P(X_n\in [a,b])-P(Y_n\in [a,b])|\rightarrow 0$ for all choices of $a$ and $b$. As the reviewer points out this is true regardless of the fact that we know that $d_{\mathcal{F}}(X_n,Y_n)\rightarrow 0$  or that we know the Wassertein distance goes to $0$ $d_W(X_n,Y_n)\rightarrow 0$. In the paper we comment on this and discuss what is achievable with and without anti concentration results. However we will expand on this to avoid any mis-conception and clarify the relation of our result with convergence under BL(1). The reason we do not add any anti concentration result is that these results typically require stronger conditions on the statistic and many times Gaussian limits, we omit this step in this work and note that a slightly weaker, albeit still practically useful, statement on coverage is achievable in a more general setup.
> > >
> > >
> > >
> > > 3. *“Therefore, the result of Proposition 2 does not imply (not alone and also not in combination with Proposition 1) that you can compute p-values for the test statistic. This contradicts your statement in lines 241-243. (Please correct me if I misinterpret what you write there.)”*
> > >
> > > This is true. We agree that the more accurate statement in our Kernel section is that we can compute an asymptotic upper bound to the p-value by slightly enlarging the rejection region (which is what is implied by Proposition 1). We will certainly make sure to revise this statement (stated before the main theorem of the kernel section) in the camera ready and to also add a comment about p-values, following our proposition 1, which only talks about confidence intervals. Thank you for pointing this out. We will revise this. However, while slightly weaker we believe that our result about p-values can still be very helpful. Indeed, being able to use the bootstrap method to give an upper bound on the p-value  is still helpful when the limiting distribution is unknown and no other methods to compute a p-value or an upper bound exist.
> > >
> > > 4) *“However, non-differentiable, discontinuous functions cannot be approximated by thrice diff'able functions with bounded derivatives and thus violate Assumption 1.1”*
> > >
> > > Certain sets of non-differentiable discontinuous functions can be approximated by three times differentiable functions. To cite an easy example: the max can be approximated by the softmax, as the reviewer pointed out, indicator functions can be approximated, the absolute value can,many differentiable by part functions can be approximated ect...
> > > *“So, while the condition of Lipschitz-continuity under expectation is in principle quite interesting, you can't exploit this condition because all cases in which it could make a difference have to be excluded a priori from your analysis by Assumption 1.1.”*
> > >
> > > We will make sure to clarify the point that the goal of our theorem is not to weaken assumption 1.1 but instead to relate $\mathbb{E}(g_n(Y_n))$ and $\mathbb{E}(g_n(\tilde Y_n))$. Indeed, Theorem 1 proves that the bootstrap estimator has the same asymptotic distribution than $g_n(\tilde Y_n)$ which might be different from the distribution of $g_n(Y_n)$. As we are interested in building confidence intervals for $\mathbb{E}(g_n(Y_n))$ and not for $\mathbb{E}(g_n(\tilde Y_n))$ this implies that the bootstrap estimator does not always provide with consistent confidence intervals for the former. Therefore the goal of theorem 2 is to show that under additional conditions of the bootstrap estimator one can correct the bootstrap confidence intervals to obtain intervals with a guaranteed minimum coverage for $\mathbb{E}(g_n(Y_n))$.
> > >
> > >
> > > We would like again to thank the reviewer for their time.

---

> > > > ### Comment · Reviewer_XTTq · 2021-08-19
> > > > **Rebuttal cont'**
> > > >
> > > > Thank you for your quick and comprehensive response.
> > > >
> > > > Reg 1: Let me just copy-paste part of a comment that I posted to my fellow reviewers and that was hidden from you. I'd love hear your thoughts on this, too:
> > > >
> > > > Do the authors really show how to apply their machinery to $\arg\max_\theta \sum_i \ell_i(\theta)$? Because the two examples in Section 4 and 5 are much simpler than this general case. (Admittedly, the examples in Section 4 and 5 are quite involved.)
> > > >
> > > > First, the statistic in Section 4 is a maximum statistics because as $\beta_n \rightarrow \infty$ the weights $w_k$  collapse to $w_k = \mathbf{1}\\{p_{\theta_k} = \max_j p_{\theta_j}\\}$. Therefore, to me a natural way of analyzing this statistic would be to interpret it as (yet) another approximation of the maximum function. This suggests that one should do all Gaussian and bootstrap approximation steps using the classical maximum statistic (by Chernohukov et al. 2013-2020) and in the end add an additional approximation error that arises from comparing said statistic to the maximum statistic. This way one could avoid most (if not everything) that authors have derived in this paper. On top of this, one would get a stronger result, namely consistency in Kolmogorov distance. This would allow one to compute critical values, p-values, etc. The theoretical results in the current paper are much weaker.
> > > >
> > > > Second, the statistic in Section 5 is a similar maximum-type statistic which simplifies considerably as $\beta_n \rightarrow \infty$ (but not as much as the example in Section 4). I'm convinced that as with the statistics of Section 4 its asymptotic behavior is fully described by a max-statistic plus negligible approximation errors. It should therefore be analyzed using the same reduction approach outlined above. (...) Quote end.
> > > >
> > > > Reg 2. Thank you for the proof that is supposed to demonstrate that $d_\mathcal{F}(X, Y) = 0$ implies $X \overset{d}{=} Y$.
> > > >
> > > > However, there is something I do not fully understand: As you take $\epsilon \rightarrow 0$ the derivatives of your smooth approximation of the indicator function $h_\epsilon$ (defined in supplement, lines 868-869) become unbounded and thus fall outside the class $\mathcal{F}$ (defined in main text, line 113). It seems to me that for every fixed $\epsilon > 0$ your argument works. But I do not see how you can make this work in the limit as $\varepsilon \rightarrow 0$.
> > > >
> > > > Since the boundedness of the derivatives is extremely helpful in your proofs of your main results (b/c it allows you to forgo the delicate choice of smoothing parameters by CCK and related literature), could you please clarify this? My hunch is, that you still cannot get weak convergence using your apparatus based on functions with bounded derivatives.
> > > >
> > > > Quoting again from my comment to the other reviewers:
> > > >
> > > > Instead, the authors should be using the standard metric based on bounded Lipschitz-continuous functions and, as reviewer iK1w suggested, work out the rate at which this distance goes to zero and pick  $\epsilon > 0$ accordingly. However, since this requires working with the supremum over all bounded Lipschitz continuous functions/ the Wasserstein distance, this is likely a very hard problem. It is much easier to show consistency only in Kolmogorov distance (and this is sufficient for all statistical purposes, e.g. computations of p-values). Consistency in Kolmogorov distance is precisely what Chernozhukov and Kato (2013-2020) show. In order to get a rate and to account for how the growing dimension interferes with approximation error, they use a Gaussian anti-concentration inequality and smooth approximation of indicator functions. So far, Gaussian anti-concentration inequalities only exist for the maximum-statistic. This and the quality of the smooth approximation of the indicator function are the two bottlenecks of this field. Obviously, the current paper makes zero contribution to these two important problems. (...) Quote end.
> > > >
> > > > Regarding your comment on anti-concentration in main text in lines 146-149: What stronger assumptions are you referring to? When do you require a Gaussian limit? To me the genius of the work of CCK is to clarify that we can approximate certain statistics with sequences of Gaussian proxy-statistics that usually have no limit or no Gaussian limit. The only conditions the statistics need to satisfy are certain stability conditions of the derivatives of (smooth approximations of) the statistic and a Gaussian anti-concentration inequality. The contribution of your paper seems to be that you (implicitly) show that we do not need Gaussian anti-concentration inequalities if we are satisfied with much weaker results, i.e. some vague notion of confidence sets with an asymptotic lower bound on the coverage probability. I gave you credit for this already in my first round of reviews. This is certainly a nice observation, but I do not think that it is terribly useful result (even though it requires 60 pages of proofs, some of which are interesting in its own right!).
> > > >
> > > > Some quotes about non Gaussian limits and Gaussian approximation:
> > > >
> > > > Quote from CCK, 2013, AOS top of page 2787: "When p is fixed, this distribution can be approximated by the classical Central limit theorem (CLT) applied to X. However, in modern applications (cf. [8]), p is often comparable or even larger than n, and and the classical $\mathrm{\textbf{CLT\\:does\\:not\\:apply}}$ in such cases."
> > > >
> > > > Quote from CCK, 2014, AOS, Abstract: "We study applications of this approximation theorem to local and series empirical processes arising in nonparametric estimation via kernel and series methods, where the classes of functions change with the sample size and are $\mathrm{\textbf{non-Donsker}}$".
> > > >
> > > > Reg 4. Apparently I wasn't clear enough: Yes, you can approximate non-differentiable and discontinuous functions with thrice diff'able functions. But I do not know how to construct approximating functions that have bounded derivatives independent of the goodness of the approximation. In particular, I do not know how to approximate the max and the indicator function such that the approximation error goes to zero while the derivatives stay bounded. (While I do not have a formal proof of this, i find this also very intuitive.) Therefore, I believe that your answer to this point misses the mark. I still feel that your assumption $H_\infty$ combined with Assumption 1.1. essentially requires Lipschitz-continuity with respect to the $\ell_\infty$-norm and hence severely restricts the classes of functions that can be handled and/or results in excessively large CIs (see my first round review).

---

> > > > > ### Author Response · Authors · 2021-08-20
> > > > > **Answer**
> > > > >
> > > > > 1.*"First, the statistic in Section 4 is a maximum statistics because as $\beta_n \rightarrow \infty$ the weights $w_k$  collapse to $w_k = \mathbf{1}\{p_{\theta_k} = \max_j p_{\theta_j}}$. "*
> > > > >
> > > > > Imagine that indeed we took $\beta$ to go to infinity and that all the weights concentrate on one Kernel then this kernel would be the one that has the highest estimated power: $argmax_{\theta} \hat{P}_{\theta}.$
> > > > >
> > > > > The statistics we study is the empirical average of $1/n^2\sum_{i,j} H_{i,j}^{\theta_k}$ which is a (different) function of the chosen Kernel. However, it is not a maximum: we choose the kernel by maximizing the power, and then we use that chosen kernel to compute another statistic that is completely different from the power. Note that $p_{\theta}$ and $H^{\theta}$ are very different functions. Therefore our statistics is of the form $h(\hat \theta,X_1,\dots,X_n)$ with $\hat\theta:=argmin_{\theta} g(X_1,\dots,X_n)$ where $h$ and $g$ are different statistics.
> > > > >
> > > > >
> > > > > *“Second, the statistic in Section 5 is a similar maximum-type statistic which simplifies considerably  “ *
> > > > >
> > > > > No, it will not be a maximum type of statistics. We will definitely elaborate more in the text on why these statistics do not fall (or are approximable by) the maximum type of statistics captured by CCK. It is key to note that in proposition 3 the statistics is the empirical loss of $\hat \Theta_n$ computed on a different sample than $\hat \Theta_n$ was computed on: either a bootstrap sample or an independent copy. We should have maybe made this more explicit but in proposition 3 the smooth stacked estimator has not been resampled by the bootstrap estimator. Therefore even if you would have $\hat\Theta_n$ collapse to the estimator with the lowest empirical risk on $X$, then the statistics is still not a min as it is computed on a different sample than the estimator $\hat \Theta_n$ was sampled on. Note that this out-of-sample structure is crucial for our result to work without further restrictions on the estimator types.
> > > > >
> > > > >
> > > > > 2.
> > > > > *“However, there is something I do not fully understand: As you take $\epsilon \rightarrow 0$ the derivatives of your smooth approximation of the indicator function $h_\epsilon$ (defined in supplement, lines 868-869) become unbounded and thus fall outside the class $\mathcal{F}$ (defined in main text, line 113). It seems to me that for every fixed $\epsilon > 0$ your argument works. But I do not see how you can make this work in the limit as $\varepsilon \rightarrow 0$.
> > > > > “*
> > > > >
> > > > >
> > > > > The key is to first take the limit in $i\rightarrow \infty$ and then take the limit in $\epsilon\rightarrow 0$.
> > > > >
> > > > >
> > > > > Yes if we took first $\epsilon$ to $0$ then as the reviewer pointed out the upper and lower bound of proposition 1 would diverge if $d_{\mathcal{F}}(X_i,Y)$ is not 0. However this is not what we are arguing to do. As the definition of convergence in distribution only requires that $P(X_i\le t)\rightarrow P(Y\le t)$ for all points at which $t$ is continuous one can first take the limit in $i\rightarrow \infty$ and then take the limit in $\epsilon\rightarrow 0$. Indeed let $t\in \mathbb{R}$  be such that $t\rightarrow P(Y\le t)$ is continuous. As $P(Y\le t)$ is continuous at that point for all $\delta>0$ there is an $\epsilon>0$ such that $$\big|P(Y\le t)-P(Y\le t-\epsilon)\Big|\le \delta, \qquad \big|P(Y\le t)-P(Y\le t+\epsilon)\big|\le \delta.$$
> > > > > Now for that fixed $\epsilon>0$ proposition 1 (which the reviewer agrees is true we have )
> > > > > $$\liminf_i P(X_i\le t)\ge P(Y\le t-\epsilon)\ge P(Y\le t)-\delta,\qquad \limsup_i P(X_i\le t)\le P(Y\le t+\epsilon)\le P(Y\le t)+\delta.$$
> > > > > This directly implies that $$\limsup_i \big|P(X_i\le t)- P(Y\le t)\big|\le 2\delta.$$
> > > > > Now this can be done for all $\delta>0$ therefore $$\limsup_i \big|P(X_i\le t)- P(Y\le t)\big|=0.$$
> > > > >  The key is to note that we can first take the limit in $i\rightarrow \infty$ and then only take $\epsilon\rightarrow 0$ because we only to prove converges of the cdf at contuinity points.. We will make this more clear in the camera ready version
> > > > >
> > > > > This implies that if $d_{\mathcal{F}}(X_i,Y)\rightarrow 0$ then $X_i\overset{d}{\rightarrow}Y$.
> > > > >
> > > > > *“vague notion of confidence sets with an asymptotic lower bound on the coverage probability. “*
> > > > >
> > > > > Proposition 1 gives a concrete notion of a confidence set that, while being slightly weaker, is still useful in practice as $\epsilon>0$ can be taken to be infinitesimally small.
> > > > >
> > > > >
> > > > > 3) *“the contribution of your paper seems to be that you (implicitly) show that we do not need Gaussian anti-concentration inequalities “*
> > > > >
> > > > > No this is not what we claim to be the contribution of our paper. The contribution of our paper is to give general conditions under which you can study the asymptotic distribution of the bootstrap estimator. We understand that the bootstrap estimator was already studied for statistics that come in the form of maximums but we give general conditions that go further than simply the maximums. We would like to note that even for the maximum (see the appendix) we obtain a faster rate of convergence than in CCK and a rate that is instead matching the one of Deng (2020)
> > > > >
> > > > >
> > > > > 4) Thank you for the clarification. This is very helpful. It is true that the better our approximation will be, the bigger the derivatives will often be. However we can solve this problem by carefully choosing the function $f_n$ chosen to approximate $g_n$. We will want to choose them to be such that ultimately as n grows the function $f_n$ become better and better estimators of the functions $g_n$ but also want to choose them to be such that the derivatives of $f_n$ grow very slowly with $n$. It is a careful tradeoff and this is what we do in the appendix in the proof of proposition 5 (which is in this case a maximum statistics).
> > > > >
> > > > >  To make this clear let’s give a toy example:
> > > > >  Imagine that our observations $X_i$ are i.i.d uniform [0,1], and definite $h:x\rightarrow \mathbb{I}(x\le 0.5)$ to be an indicator function. This is indeed a non-smooth function. As we have shown in the proof of proposition $1$ we can find functions $(f_{\epsilon})$ that are such that:
> > > > >
> > > > > (i) $\big|f_{\epsilon}(x)-h(x)\big|\le 3\epsilon$ and such that (ii) we have the derivatives bounded by $\max_{i\le 3}|h^{(i)}(x)|\le \frac{1}{\epsilon^3}$.
> > > > >
> > > > > Imagine that our statistics is the indicator function $g_n(x_1,\dots,x_n)=h(\frac{1}{\sqrt{n}}\sum_i X_i)$. Then choose $\epsilon_n=1/\log(n)$ ( a very slow choice)  and set $f_n(x_1,\dots,x_n)=f_{\epsilon_n}(\frac{1}{\sqrt{n}}\sum_i X_i)$.
> > > > >
> > > > > Then as $\epsilon_n$ goes to $0$ asymptotically we have
> > > > >
> > > > > $$||f_n(X_1,\dots,X_n)-g_n(X_1,\dots,X_n)||_{\infty}\rightarrow 0$$
> > > > >
> > > > >  but as $\epsilon_n$ is chosen to goes to 0 very slowly the derivatives would grow only as $\log(n)^3$.
> > > > >
> > > > > So while the derivatives grow they grow at such a slow pace that we can still say that $f_n(Z_1,\dots,Z_n)$ has the same asymptotic distribution than $f_n(\tilde Y_1,\dots,\tilde Y_n)$  and that it does so at a rate of
> > > > >
> > > > > $\frac{log(n)^3}{\sqrt{n})$.
> > > > >
> > > > >  Note that if we did not make $\epsilon$ go to $0$  but instead studied $f_{\epsilon}(Z_1,\dots,Z_n)$ for  a fixed $\epsilon$ then the convergence would happen at a speed of $\frac{1}{\sqrt{n}}$

---

> > > > > > ### Comment · Reviewer_XTTq · 2021-08-20
> > > > > > **Cont'**
> > > > > >
> > > > > > Regarding 2:
> > > > > >
> > > > > > a) I finally understand one of my misunderstandings. In the supplementary material line 870 the function class $\mathcal{F}$ can indeed be chosen to contain only functions with bounded derivatives. Clearly, $\epsilon^3 h_\epsilon$ lies in this function class. So, your argument in the proof is completely sound. I always thought you assumed $h_\epsilon \in \mathcal{F}$ and for fixed $\epsilon > 0$ is was willing to accept this as an irrelevant mistake (because a fixed $\varepsilon > 0$ doesn't faze me).
> > > > > >
> > > > > > b) Does your argument also work if the sequence $(X_n)_{n\geq 1} $ has no weak limit (as in high-dim. cases with diverging dimension)? This is unclear to me because we might not be able to talk about continuity points of the CDF.
> > > > > >
> > > > > > It seems as if invoking the continuity of the limiting CDF allows you to avoid having to develop an anti-concentration inequality for your complicated statistics. That's interesting. If a similar argument works also in high-dim. (when there is no weak convergence), you could establish consistency in your metric $d_\mathcal{F}$ and consistency in Kolmogorov distance would follow - the only price you'd pay would be that you do not know the rate at which the Kolmogorov distance goes to zero.
> > > > > >
> > > > > > Thank your answers on my other questions. I will study them carefully and if I have further questions, I'll let you know.

---

> > > > > > > ### Comment · Reviewer_XTTq · 2021-08-20
> > > > > > > **Cont'**
> > > > > > >
> > > > > > > Reg 1: Ok. Of course, this is absolutely correct what you write.
> > > > > > >
> > > > > > > But you will certainly agree that both examples in Section 4 and 5 involve some sort of weighting reminiscent to the smooth max. Why do you do this? Well, I suspect that you do this because this the only way in which you can satisfy the strong stability condition (Assumption 1.2), which requires that the sum over the partial derivatives are of order $o(n^{-1/3})$ and smaller. This is an extremely strong assumption because the sum over the first partial derivatives comprises $d_n$ summands, the sum over the second partial derivatives $d_n^2$ summands, etc.. Thus, if $d_n \gg n$ this is extremely difficult to satisfy. Notably, three out of the four examples in the supplementary materials are in fact low-dimensional with $d_n = 1$  (examples 3,4,5) while the fourth example (example 6) has $d_n = n$ which also does not qualify as high-dimensional. Moreover, example 6 is again some sort of smooth max function. Besides, at least in the case of example 3, we could use CLT and D̶e̶l̶t̶a̶-̶M̶e̶t̶h̶o̶d̶  Slutzky's Theorem to obtain [an implicit characterization of] the asymptotic distribution under weaker moment conditions.
> > > > > > >
> > > > > > > This seems to suggest that your theory will most likely only be applicable either to low-dimensional settings (which to a large extend could also be dealt with by existing bootstrap theory) or to some sort of maximum statistics that involve some version of the smooth max (which may or may not be dealt with by existing theory for the max-statistic). All in all, your general framework seems to have a rather narrow scope with regard to applications. (See also reviewer iK1w for a similar concern.)
> > > > > > >
> > > > > > > Question: Do your assumptions actually allow the analysis of the max-statistic $\max_{1 \leq k \leq d_n}  n^{-1/2}\sum_{i=1}^n X_{ik}$? I don't know whether the smooth max approximation of this statistic satisfy your assumption 1.2 (see also Lemma A.2 in CCK 2013, AOS).
> > > > > > >
> > > > > > > Reg 3: I think that if your argument under 2 (i.e. convergence in $d_F$ implies convergence in Kolmogorov distance) extends to the case in which the limit distribution does not exist, then this is your most interesting contribution. If this was true, Theorem 1 would immediately allow you to compute p-values and construct asymptotically exact CIs and thus sharpen all theoretical results in your paper. However, note that even if the argument under 2 can be extended, it will not give a rate on the Kolmogorov distance. Therefore you cannot compare your rate on $d_F$ with the rates of CCK (2013-2020) and Deng & Zhang (2020) on the Kolmogorov distance.

---

> > > > > > > > ### Author Response · Authors · 2021-08-20
> > > > > > > > **Answer**
> > > > > > > >
> > > > > > > > We would like to thank the reviewer for his time and effort.
> > > > > > > >
> > > > > > > > *“Does your argument also work if the sequence $(X_n)_{n\geq 1} $ has no weak limit (as in high-dim. cases with diverging dimensions)? “*
> > > > > > > >
> > > > > > > > The distribution that we are interested in studying is the distribution of the bootstrap estimator $g_n(Z_1,\dots,Z_n)$. While the observations $(X_i),(Z_i)$ can take value in high dimensional spaces the function $g_n$ is assumed to take value in $\mathbb{R}$, an example of choice for $(g_n)$ can be the risk of a high dimensional estimator.
> > > > > > > > If $d_{\mathcal{F}}(X_n,Y)\rightarrow 0$ then $(X_n)$ converges in distribution to $Y$. If $d_{\mathcal{F}}(X_n,Y_n)\rightarrow 0$ and $(Y_n)$ does not have a limit then using proposition 1 we can construct infinitesimal wider CI with asymptotically nominal coverage. If $(Y_n)$ is tight then on all the subsequences that admit a limit, the corresponding sequence of $(X_n)$ will converge to the same limit.
> > > > > > > >
> > > > > > > >
> > > > > > > > *“But you will certainly agree that both examples in Section 4 and 5 involve some sort of weighting reminiscent to the smooth max. Why do you do this?”*
> > > > > > > >
> > > > > > > > The reason why it is reminiscent of a smooth max is because we are considering smooth argmax, which is natural in machine learning as most quantities of interest can be written as a function of an estimator that is chosen by minimizing an empirical risk. Our examples  are  functions of estimators/kernels chosen by minimizing/maximizing some objective functions. Moreover our methods can be extended to other types of quantities such as the cross-validated risk.
> > > > > > > >
> > > > > > > > “only way in which you can satisfy the strong stability condition (Assumption 1.2), which requires that the sum over the partial derivatives are of order $o(n^{-1/3})$ and smaller. This is an extremely strong assumption “
> > > > > > > >
> > > > > > > > The condition that the partial derivatives are of order $o(n^{-⅓})$ is not such a strong assumption and is there to guarantee that no single observation is too influential in determining the value of $g_n$. Notably we need that
> > > > > > > >
> > > > > > > > $$||g_n(X_1,\dots,X_n)-g_n(0,X_2,\dots,X_n)||_{L_3}=o(n^{-1/3}).$$
> > > > > > > >
> > > > > > > > This condition is tight, in the appendix we propose an example of $g_n$ that is such that $||g_n(X_1,\dots,X_n)-g_n(0,X_2,\dots,X_n)||_{L_3}\propto n^{-1/3},$ and prove that in that case the bootstrap is not consistent. This goes to show that this condition is not “extremely strong” but instead close to being tight.
> > > > > > > >
> > > > > > > > *“Notably, three out of the four examples in the supplementary materials are in fact low-dimensional with $d_n = 1$  (examples 3,4,5) while the fourth example (example 6) has $d_n = n$ which also does not qualify as high-dimensional. Moreover, example 6 is again some sort of smooth max function. Besides, at least in the case of example 3, we could use CLT and Delta-Method to obtain the asymptotic distribution under weaker moment conditions.”*
> > > > > > > >
> > > > > > > > In the appendix some examples are meant to be illustrative examples. They are therefore low dimensional examples that are easy to understand. We would like to argue that what can be considered as  high-dimensional depends a lot on the type of estimators considered. For example in compressed sensing when the dimension is growing linearly with $n$ it is considered as  a high-dimensional setting. Moreover, the example in the appendix that is studying the maximum as well as both of our examples in the main text have the dimension scaling exponentially with the number of observations $n$.
> > > > > > > >
> > > > > > > > *“Moreover, example 6 is again some sort of smooth max function. Besides, at least in the case of example 3, we could use CLT and Delta-Method to obtain the asymptotic distribution under weaker moment conditions.”*
> > > > > > > >
> > > > > > > > We agree example 6 is a smooth max function and the only smooth max function that we consider. Example 3 is an illustrative example. In that example the estimators are not asymptotically normal (but instead are for example asymptotically chi-squared). The CLT plus delta method always produces asymptotic normal limits. Hence, even for there illustrative examples, the approach proposed is not applicable. Though again, these are illustrative examples to show the breadth of the applicability of our result and should not be seen as new results. Most probably for these illustrative examples, simpler proofs could apply to show consistency of the bootstrap.
> > > > > > > >
> > > > > > > >
> > > > > > > > *Do your assumptions actually allow the analysis of the max-statistic $\max_{1 \leq k \leq d_n}  n^{-1/2}\sum_{i=1}^n X_{ik}$? I don't know whether the smooth max approximation of this statistic satisfy your assumption 1.2 *
> > > > > > > >
> > > > > > > > Yes we also treat the maximum (see section J (example 6)) and we assume that $\log(d_n)=o(n^{1/4})$.  To do so we use an extension of our theorem that you can find in Appendix B.
> > > > > > > >
> > > > > > > > *“This seems to suggest that your theory will most likely only be applicable either to low-dimensional settings (which to a large extend could also be dealt with by existing bootstrap theory) or to some sort of maximum statistics that involve some version of the smooth max (which may or may not be dealt with by existing theory for the max-statistic).”*
> > > > > > > >
> > > > > > > > Both of our examples in the main text have the dimensions growing exponentially with $n$ which is a high-dimensional setting. Moreover, our examples cannot be seen as maximums but instead are functions of estimators/kernels chosen by minimizing/maximizing some objective functions. There are indeed connections between soft max and soft argmax functions but those still behave substantially differently.
> > > > > > > >
> > > > > > > > *“Therefore you cannot compare your rate on $d_F$ with the rates of CCK (2013-2020) and Deng & Zhang (2020) on the Kolmogorov distance.”*
> > > > > > > >
> > > > > > > > This is a fair point but we do not compare the rate. Instead we compare the dimensions of our problem. We allow the dimension to grow as $\log(p)=o(n^{¼})$. This is faster than in CKK.

---

### Decision · Program_Chairs · 2021-09-28

**Decision:**

Accept (Poster)

**Comment:**

There has been a tremendous amount of discussion about this paper. Several points of criticism have been raised by Reviewer XTTq, mostly concerning the used stability condition, consistency in high dimensions and the generality of the specific kernel used and the examples in sections 4/5. On the other hand, all other reviewers had a much more positive impression of this paper, which is reflected in their high scores that they were willing to defend in the discussion. In my opinion the author rebuttal was very convincing and successfully addressed most major points of criticism in a detailed, transparent and meaningful way. So I recommend acceptance of this paper.

**Consistency Experiment:**

NeurIPS has a long history of experimentation. In 2014, NeurIPS ran an experiment in which 10% of submissions were reviewed by two independent committees to quantify the randomness in the review process. This year, we repeated a variant of this experiment to see how the quality of the review process has changed over time.  This paper was part of the experiment and was therefore assigned to two committees (consisting of reviewers, an Area Chair, and a Senior Area Chair) that reached independent decisions.  If both committees made the same recommendation, this recommendation was followed. If a single committee recommended acceptance, the paper was accepted (with the exception of a few cases in which the other committee identified what we considered a fatal flaw, e.g., an error in a key result).

This copy’s committee reached the following decision: **Accept (Poster)**

The other committee assigned to the paper recommended **Reject**.  You can find the other set of reviews, along with any follow up discussion with the authors here:
https://openreview.net/forum?id=5JPPOluv-bp